# Circumventing Backdoor Space via Weight Symmetry

**Jie Peng**[1]  **Hongwei Yang**[1]  **Jing Zhao**[1]  **Hengji Dong**[1]  **Hui He**[1]  **Weizhe Zhang**[1 2]  **Haoyu He**[3]

## Abstract

Deep neural networks are vulnerable to backdoor attacks, where malicious behaviors are implanted during training. While existing defenses can effectively purify compromised models, they typically require labeled data or specific training procedures, making them difficult to apply beyond supervised learning settings. Notably, recent studies have shown successful backdoor attacks across various learning paradigms, highlighting a critical security concern. To address this gap, we propose *Two-stage Symmetry Connectivity* (TSC), a novel backdoor purification defense that operates independently of data format and requires only a small fraction of clean samples. Through theoretical analysis, we prove that by leveraging permutation invariance in neural networks and quadratic mode connectivity, TSC amplifies the loss on poisoned samples while maintaining bounded clean accuracy. Experiments demonstrate that TSC achieves robust performance comparable to state-of-the-art methods in supervised learning scenarios. Furthermore, TSC generalizes to self-supervised learning frameworks, such as SimCLR and CLIP, maintaining its strong defense capabilities. Our code is available at https://github.com/JiePeng104/TSC.

## 1. Introduction

Modern classifiers require substantial data and computational resources to achieve high accuracy, providing adversaries with opportunities to implant backdoors into deep neural networks (Gu et al., 2017; Chen et al., 2017). This vulnerability arises from either injecting poisoned data (Gu et al., 2017; Turner et al., 2019) (*i.e., data-poisoning attack*), or manipulating the training process (Pang et al., 2019;

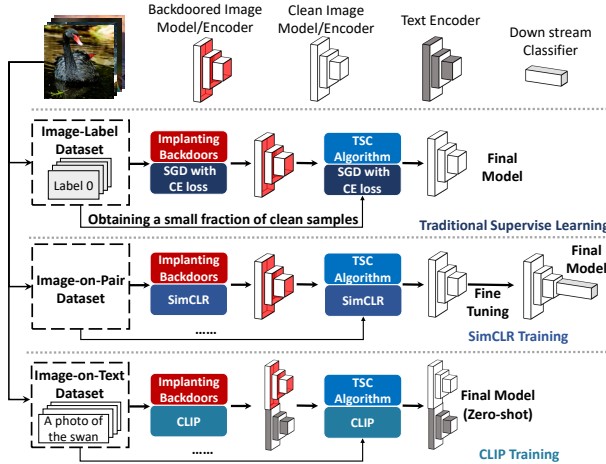

*Figure 1.* Illustration of the application of TSC in three popular learning settings. We assume adversaries can perform either data-poisoning attack or training-manipulation attack to implant a backdoor into the weights of a classification model or an encoder. Unlike most existing defenses requiring specific training procedures, TSC provides a framework to remove backdoors using only a small fraction of clean samples and the original training process. For instance, in SimCLR (Chen et al., 2020) setting, a TSC defender can directly remove the backdoor hidden in the image encoder by combining TSC with the SimCLR training procedure. This allows for training the downstream classifier without the adversarial influence inherited from upstream.

Nguyen & Tran, 2020) (*i.e., training-manipulation attack*). For instance, recent advancements in self-supervised learning strategies often rely on large volumes of training data, which, while circumventing the need for curated or labeled datasets, can be time-intensive and incur high computational costs (Radford et al., 2021; He et al., 2022; Chen et al., 2020). Therefore, many users prefer to delegate model training to third-party providers or fine-tune publicly available models on downstream tasks. This practice exposes models to backdoor attacks. For example, malicious providers can manipulate the training process to implant backdoors and then force downstream classifiers to output adversarial labels (Yao et al., 2019; Saha et al., 2022; Li et al., 2023).

Consequently, many defenses have been developed to eliminate backdoors hidden within models. One of the most prominent mechanisms is *post-purification defenses*, which remove backdoors through post-training processes (Liu

[1]School of Cyberspace Science, Harbin Institute of Technology, Harbin, China. [2]Pengcheng Laboratory, Shenzhen, China. [3]Department of Data Science and AI, Faculty of IT, Monash University, Melbourne, Australia. Correspondence to: Hui He <hehui@hit.edu.cn>.

*Proceedings of the 42nd International Conference on Machine Learning*, Vancouver, Canada. PMLR 267, 2025. Copyright 2025 by the author(s).

et al., 2018; Wang et al., 2019; Wu & Wang, 2021; Zeng et al., 2022). Usually, post-purification defenses require only a small amount of clean data and are effective against both data-poisoning and training-manipulation attacks. However, existing approaches focus primarily on supervised learning scenarios and rely on training methods requiring labeled data (Wang et al., 2019; Wu & Wang, 2021; Li et al., 2021b; Chai & Chen, 2022) (*e.g.,* methods working as analogues to adversarial training). Thus, they are not directly applicable to learning regimes like self-supervised or unsupervised learning. Moreover, some studies have shown that most current defense mechanisms, whether categorized as post-purification defenses or not, are vulnerable to attacks using small poisoning rates or adaptively designed triggers (Hayase et al., 2021; Qi et al., 2023; Min et al., 2023).

Previous studies have intentionally or unintentionally tried to address the challenges of robustness and transferability in backdoor defense. Recent work by Min et al. (2023) re-investigated fine-tuning (FT) based methods, proposing to enhance robustness under low poisoning rates. While achieving promising performance, their methods remain confined to supervised learning settings. The mode connectivity repair (MCR) (Zhao et al., 2020), which leverages quadratic mode connectivity (Garipov et al., 2018), offers a data-format agnostic procedure but shows limited robustness against various attack methods. To address all these challenges, in this paper, we explore *a robust post-purification defense that operates independently of the data format.*

We propose Two-stage Symmetry Connectivity (TSC), a novel defense mechanism that leverages weight symmetry in neural networks to guide compromised models away from backdoor behaviors without directly unlearning backdoor patterns (*i.e.,* circumventing the backdoor space). Our approach builds upon two key properties: permutation invariance, which allows equivalent model representations through weight layer permutations (Entezari et al., 2022); and quadratic mode connectivity (Garipov et al., 2018), which connects model states through low-loss quadratic paths.

Our defense process begins by projecting a copy of the compromised model into a distinct yet symmetrical loss basin, leveraging the permutation invariance property of neural networks. These two models serve as endpoints for a Bézier curve that is trained using a small clean dataset. We then pick a point on this trained curve as the purified model, which completes the first stage of our defense. As the endpoints are in different loss basins and this curve training utilizes only clean samples, we show that the loss of poisoned samples along the curve is amplified. Subsequently, we merge the purified model and the backdoored model in the original loss basin to maintain clean accuracy, which completes the second stage of our defense. Figure 1 illus-

trates how TSC operates across different learning settings, using only a small fraction of clean samples and a procedure aligned with the original training process. Overall, our contributions are as follows:

- We discover that the property of weight symmetry enable effective backdoor space circumvention. Based on this insight, TSC provides a unified framework for backdoor defense across various learning settings.

- Through analysis of the mechanisms behind permutation invariance and mode connectivity, we provide theoretical guarantees that TSC can amplify the upper bound of loss values on poisoned samples while maintaining accuracy on the initial task.

- Experiments on CIFAR10 (Krizhevsky, 2009), GTSRB (Houben et al., 2013), and ImageNet100 (Deng et al., 2009) under supervised learning demonstrate that TSC achieves performance comparable to state-of-the-art methods while maintaining robustness against various attack settings, including small poisoning rates and adaptively designed attacks. Moreover, TSC successfully counters attacks on image encoders across different learning frameworks.

## 2. Related Work

**Mode Connectivity.** Merging two models with different initializations usually involves the concept of *mode connectivity*, which lies at the heart of finding a low-loss linear or nonlinear path connecting two models (Frankle et al., 2020; Garipov et al., 2018; Gotmare et al., 2018; Draxler et al., 2018). Empirical studies have shown that aligning two models through permutation before merging can greatly enhance the generalization of models along a linear path (Entezari et al., 2022; Simsek et al., 2021), or a quadratic curve (Tatro et al., 2020). Notably, recent research demonstrates that models aligned within the same loss basin can be merged effectively by averaging their weights (Ainsworth et al., 2023; Singh & Jaggi, 2020; Jordan et al., 2023).

**Backdoor Attacks.** A backdoor adversary aims to make the victim model maintain accuracy on clean inputs while assigning target labels to trigger-embedded inputs (Gu et al., 2017; Chen et al., 2017). Based on the attacker's capabilities, backdoor attacks can be categorized as *data-poisoning attack* and *training-manipulation attack*. In *data-poisoning attacks*, the adversary poisons a portion of the training set by injecting pre-designed triggers and modifying their labels. The trigger patterns can range from pixel squares to real-world objects (Wenger et al., 2022) or invisible patterns (Li et al., 2021a; Saha et al., 2020; Li et al., 2021c). To avoid detection of mislabeled samples as outliers, some studies have developed backdoor attacks that maintain original labels (Turner et al., 2019; Barni et al., 2019; Shafahi et al., 2018; Zhu et al., 2019). *Training-manipulation* attackers

have full access to the training process, enabling them to effectively inject backdoors through specific training-based methods (Nguyen & Tran, 2020; Pang et al., 2019).

Most backdoor attack studies focus on supervised learning scenarios. However, recent work has indicated that models in self-supervised learning settings are vulnerable to backdoor attacks, which may involve the adversary controlling the training process (Jia et al., 2022), or merely injecting poisoned samples (Saha et al., 2022; Carlini & Terzis, 2022; Li et al., 2023; Carlini et al., 2023).

**Backdoor Defenses.** Current backdoor defenses can be divided into three categories: *training-time defenses*, *test-time defenses*, and *post-purification defenses*. Training-time defenses require training clean models on a polluted dataset to counteract data-poisoning attacks (Chen et al., 2018; Tran et al., 2018; Hayase et al., 2021; Li et al., 2021d; Khaddaj et al., 2023). A recent approach, ASSET (Pan et al., 2023), extends this concept to handle various attacks and learning settings. Test-time defenses aim to filter out malicious inputs during inference rather than directly eliminating backdoor threats (Gao et al., 2019; Guo et al., 2023; Hou et al., 2024).

In this study, we focus on *post-purification defenses* (Liu et al., 2018; Wu & Wang, 2021; Zeng et al., 2022; Wang et al., 2019; Min et al., 2023) aiming to remove the backdoor injected into the weights of a model. Existing visual *post-purification defenses* mostly require labels accompanied with the training images, such as adversarial training based methods (Wu & Wang, 2021; Zeng et al., 2022), limiting their applicability to other training frameworks without labels. While Feng et al. (2023) proposed a method to identify backdoors in self-supervised learning settings, it focuses on detection rather than removal. Recently, Zheng et al. (2024) extended unlearning methods to self-supervised settings. Instead of unlearning backdoor patterns, our method leverages weight symmetry to purify models while preserving performance, generalizing robustly across various learning scenarios and attack settings.

## 3. Preliminaries

### 3.1. Minimum Loss Path

Here, we consider the connecting method with respect to quadratic mode connectivity (Garipov et al., 2018; Tatro et al., 2020). Let $\boldsymbol{\theta}_A$ and $\boldsymbol{\theta}_B$ be the weights of two trained models, and let $\boldsymbol{\gamma}_{\boldsymbol{\theta}_{A,B}}(t)$ denote a parametric curve connecting $\boldsymbol{\theta}_A$ and $\boldsymbol{\theta}_B$ such that $\boldsymbol{\gamma}_{\boldsymbol{\theta}_{A,B}}(0) = \boldsymbol{\theta}_A$ and $\boldsymbol{\gamma}_{\boldsymbol{\theta}_{A,B}}(1) = \boldsymbol{\theta}_B$. To train $\boldsymbol{\gamma}_{\boldsymbol{\theta}_{A,B}}(t)$, Garipov et al. (2018) proposed finding the set of parameters $\boldsymbol{\theta}_{A,B}$ that minimizes the expectation of the loss $\mathcal{L}(\boldsymbol{\gamma}_{\boldsymbol{\theta}_{A,B}}(t))$ over the distribution $p_{\boldsymbol{\theta}_{A,B}}(\cdot)$ on the curve,

$$\ell(\boldsymbol{\theta}_{A,B}) = \int_0^1 \mathcal{L}(\boldsymbol{\gamma}_{\boldsymbol{\theta}_{A,B}}(t)) p_{\boldsymbol{\theta}_{A,B}}(t) \, \mathrm{d}t, \quad (1)$$

where the $p_{\boldsymbol{\theta}_{A,B}}(t)$ is the distribution for sampling the models on the curve indexed by $t$. For simplicity in computation, the uniform distribution $U(0, 1)$ is typically chosen as $p_{\boldsymbol{\theta}_{A,B}}(\cdot)$. To characterize the parametric curve $\boldsymbol{\gamma}_{\boldsymbol{\theta}_{A,B}}(t)$ for $t \in [0, 1]$, the *Bézier curve* with 3 bends is commonly employed and is defined as follows:

$$\boldsymbol{\gamma}_{\boldsymbol{\theta}_{A,B}}(t) = (1 - t)^2 \boldsymbol{\theta}_A + 2t(1 - t)\boldsymbol{\theta}_{A,B} + t^2 \boldsymbol{\theta}_B. \quad (2)$$

More details about mode connectivity can be found in Appendix B.

### 3.2. Permutation Invariance

For simplicity, we consider an $L$-layer feedforward neural network with an element-wise activation function $\sigma$ and weights $\boldsymbol{\theta}$. We use $\boldsymbol{W}_l$ to denote the weight of the $l^{\text{th}}$ layer, $\boldsymbol{x}_0 \in \mathbb{R}^{d_0}$ to represent the input data and $\boldsymbol{y} \in \mathbb{R}^{d_L}$ to indicate the output logits (or features). The $L$-layer feedforward network can be expressed as:

$$\boldsymbol{f}(\boldsymbol{x}_0, \boldsymbol{\theta}) = \boldsymbol{y} = \boldsymbol{W}_L \circ \sigma \circ \boldsymbol{W}_{L-1} \circ ... \circ \sigma \circ \boldsymbol{W}_1 \boldsymbol{x}_0, \quad (3)$$

Moreover, we denote the $l^{\text{th}}$ intermediate feature as $\boldsymbol{x}_l \in \mathbb{R}^{d_l}$ and $\boldsymbol{x}_l = \boldsymbol{W}_l \circ \sigma \circ \boldsymbol{x}_{l-1}$.

One of the key techniques central to our defense method is *permutation invariance* of neural networks (Entezari et al., 2022; Tatro et al., 2020; Ainsworth et al., 2023). Let $\boldsymbol{P}_l \in \Pi_{d_l}$ be a permutation matrix that permutes output feature $\boldsymbol{x}_l$ of the $l^{\text{th}}$ layer, where $\Pi_{d_l}$ is the set of all possible $d_l \times d_l$ permutation matrices. Since $\boldsymbol{P}_l^\top \boldsymbol{P}_l = \boldsymbol{I}$, without changing the final output $\boldsymbol{y}$, we can permute $\boldsymbol{W}_l$ to $\boldsymbol{P}_l \boldsymbol{W}_l \boldsymbol{P}_{l-1}^\top$. We denote this operation as $\pi(\boldsymbol{\theta}, S(\boldsymbol{P}))$, where $S(\boldsymbol{P}) = \{\boldsymbol{P}_1, \boldsymbol{P}_2, ..., \boldsymbol{P}_{L-1}\}$ is the set of permutation matrices. Consequently, we obtain a new network with weights $\boldsymbol{\theta}^{S(\boldsymbol{P})} = \pi(\boldsymbol{\theta}, S(\boldsymbol{P}))$ defined as:

$$\boldsymbol{W}_1^{S(\boldsymbol{P})} = \boldsymbol{P}_1 \boldsymbol{W}_1; \qquad \boldsymbol{W}_L^{S(\boldsymbol{P})} = \boldsymbol{W}_L \boldsymbol{P}_{L-1}^\top;$$
$$\boldsymbol{W}_l^{S(\boldsymbol{P})} = \boldsymbol{P}_l \boldsymbol{W}_l \boldsymbol{P}_{l-1}^\top, \ \forall \, l \in \{2, 3, ..., L - 1\}. \quad (4)$$

The property of a neural network that allows it to be transformed by such permutation, resulting in $f(\boldsymbol{x}_0, \boldsymbol{\theta}^{S(\boldsymbol{P})}) = f(\boldsymbol{x}_0, \boldsymbol{\theta})$, is known as *permutation invariance*. This property implies that two networks can be functionally identical even if the arrangement of their neurons within each layer differs. For example, if two parameter sets, $\boldsymbol{\theta}_A$ and $\boldsymbol{\theta}_B$, satisfy $f(\boldsymbol{x}_0, \boldsymbol{\theta}_A) = f(\boldsymbol{x}_0, \boldsymbol{\theta}_B)$ for any input $\boldsymbol{x}_0 \in \mathbb{R}^{d_0}$, permutation invariance implies that their corresponding layer weights, $\boldsymbol{W}_i^A$ and $\boldsymbol{W}_i^B$, can differ. This observation motivates the concept of *neuron alignment*.

### 3.3. Neuron Alignment

Previous studies have found that two functionally identical networks, $\boldsymbol{\theta}_A$ and $\boldsymbol{\theta}_B$, when trained independently with the same architecture but using different random initializations or yielding different SGD solutions, can be misaligned (*i.e.,*

their parameters, $\boldsymbol{\theta}_A$ and $\boldsymbol{\theta}_B$, correspond to different neuron arrangements). As a result, the loss obtained from a linear interpolation of their parameters can be quite large.

Recently, research (Entezari et al., 2022; Singh & Jaggi, 2020; Ainsworth et al., 2023; Jordan et al., 2023) has shown that such misaligned networks could be projected to the same loss basin using a specific set of permutation matrices $S(\hat{\boldsymbol{P}})$. To identify such permutation set $S(\hat{\boldsymbol{P}})$ and achieve alignment, we can minimize a cost function $c_l : \mathbb{R}^{d_l} \times \mathbb{R}^{d_l} \to \mathbb{R}^+$ with respect to the intermediate features $\boldsymbol{x}_l^A$ and $\boldsymbol{x}_l^B$ to get the $\hat{\boldsymbol{P}}_l$ for each layer. In practice, given a dataset $D$ containing $n$ samples, we minimize the sum of $c_l$ across $D$. The optimal $\hat{\boldsymbol{P}}_l$ can then be found by solving:

$$\hat{\boldsymbol{P}}_l = \arg\min_{\boldsymbol{P}_l \in \Pi_{d_l}} \sum_i^n c_l(\boldsymbol{x}_{i,\,l}^A,\ \boldsymbol{P}_l\,\boldsymbol{x}_{i,\,l}^B). \tag{5}$$

This problem is a classic example in the field of optimal transport (Villani et al., 2009; Singh & Jaggi, 2020), which could be solved via the Hungarian algorithm (Kuhn, 1955). Following previous work, we adopt the alignment method proposed by Li et al. (2015), where $c_l = 1 - \mathrm{corr}(\boldsymbol{v},\ \boldsymbol{z})$ and $\mathrm{corr}(\cdot,\ \cdot)$ denotes the correlation between two vectors. This makes the minimization procedure is equivalent to ordinary least squares constrained to the solution space $\Pi_{d_l}$ (Ainsworth et al., 2023; Tatro et al., 2020). Thus, problem (5) can be specified as:

$$\hat{\boldsymbol{P}}_l = \arg\min_{\boldsymbol{P}_l \in \Pi_{d_l}} \sum_i^n \left\| \boldsymbol{x}_{i,\,l}^A - \boldsymbol{P}_l\,\boldsymbol{x}_{i,\,l}^B \right\|^2. \tag{6}$$

Throughout the paper, we define $M_l(\boldsymbol{\theta}_A, \boldsymbol{\theta}_B; D)$ to measure the $L_2$ norm feature distance of the $l^{\text{th}}$ layer between $\boldsymbol{\theta}_A$ and $\boldsymbol{\theta}_B$ given dataset $D$:

$$M_l(\boldsymbol{\theta}_A, \boldsymbol{\theta}_B; D) = \sum_i^n \left\| \boldsymbol{x}_{i,\,l}^A - \boldsymbol{x}_{i,\,l}^B \right\|^2. \tag{7}$$

Therefore, $\hat{\boldsymbol{P}}_l$ can be regarded as the optimal solution which has the smallest $M_l(\boldsymbol{\theta}_A, \pi(\boldsymbol{\theta}_B, S(\boldsymbol{P})); D)$ among all $\boldsymbol{P}_l \in \Pi_l$. Additionally, if we denote $\boldsymbol{x}_{i,\,l}^A$ and $\boldsymbol{x}_{i,\,l}^B$ as samples from two distributions $\mathbb{P}_l^A$ and $\mathbb{P}_l^B$, the minimal distance in (6) corresponds to the 2-Wasserstein distance $W_2(\mathbb{P}_l^A,\ \mathbb{P}_l^B; D)$ between $\mathbb{P}_l^A$ and $\mathbb{P}_l^B$. Thus, we have $W_2(\mathbb{P}_l^A,\ \mathbb{P}_l^B; D) = M_l(\boldsymbol{\theta}_A, \pi(\boldsymbol{\theta}_B, S(\hat{\boldsymbol{P}})); D)$. A brief explanation can be found in Appendix D.

### 3.4. Threat Model and Evaluation Metrics

We consider a scenario where an adversary can manipulate a portion of the training data or access the model training procedure. The backdoor is assumed to be implanted into the parameters of a standard neural architecture rather than a model with specific malicious modules (Bober-Irizar et al., 2023). A TSC defender requires access to a small fraction of clean training samples and and the ability to retrain the

model using the original training procedure. A discussion about the applicability of the proposed defense setting can be found in Appendix A.

To evaluate the performance of various backdoor defenses, we consider two primary metrics: *Attack Success Rate* (ASR), the proportion of attack samples misclassified as the target label, and *Clean Accuracy* (ACC), the prediction accuracy on benign samples. An effective defense should achieve a high ACC while maintaining a low ASR.

## 4. Method

Before formally describing our approach, we first provide the intuition behind TSC. Removing the implanted backdoor in a model is equivalent to inducing the model to have a high loss value for the poisoned samples (*i.e., adversarial loss*). However, performing a procedure similar to anti-backdoor learning (Li et al., 2021d) to increase the loss of the poisoned samples could be challenging for a post-time defender, as the trojan method is unknown, and the adversary could inject various trigger patterns that are difficult to recover (Li et al., 2021c).

To remove the backdoor, we propose a repairing method consisting of two stages of mode connectivity. In the first stage, we *amplify the adversarial loss* by un-aligning the malicious model $\boldsymbol{\theta}_{adv}$ with its own copy $\boldsymbol{\theta}_{adv'}$ on the loss landscape and then training a curve $\boldsymbol{\gamma}_1$ connecting $\boldsymbol{\theta}_{adv}$ and $\boldsymbol{\theta}_{adv'}$ with the given benign samples. Since this unalignment process is designed to place the endpoints ($\boldsymbol{\theta}_{adv}$ and $\boldsymbol{\theta}_{adv'}$) in different loss basins, and the curve $\boldsymbol{\gamma}_1$ is trained exclusively with benign samples, models $\boldsymbol{\theta}_t$ along this curve are expected to exhibit high adversarial loss but low loss on benign samples. This outcome is consistent with established properties of quadratic mode connectivity (Garipov et al., 2018; Tatro et al., 2020). In the second stage, to *recover the clean accuracy*, we train another curve that connects the aligned $\boldsymbol{\theta}_{adv}$ and $\boldsymbol{\theta}_t$. This procedure aims to find a curve $\boldsymbol{\gamma}_2$ with descending adversarial loss along the curve but a much lower loss for benign samples compared to $\boldsymbol{\gamma}_1$.

### 4.1. Adversarial Loss Amplification

We present a case in the left of Figure 2, where the model is attacked by SSBA (Li et al., 2021c), and the defender simply trains a Bézier curve connecting the initial backdoored model $\boldsymbol{\theta}_{adv}$ and its slightly fine-tuned version, $\boldsymbol{\theta}_{ft}$ (Zhao et al., 2020). As shown, if the poisoned model $\boldsymbol{\theta}_{adv}$ and $\boldsymbol{\theta}_{ft}$ lie in the same loss basin, eliminating the backdoor through model connection or fusion is difficult, as the adversarial loss along the curve remains low. To amplify the adversarial loss, we utilize the permutation invariance property to project $\boldsymbol{\theta}_{adv}$ to a distinct loss basin to obtain the other endpoint, $\boldsymbol{\theta}_{adv'}$, rather than fine-tuning $\boldsymbol{\theta}_{adv}$. This can be achieved by finding a set of permutation matrices $S(\boldsymbol{P}')$ that maximize the cost function $c_l$ for each layer. In con-

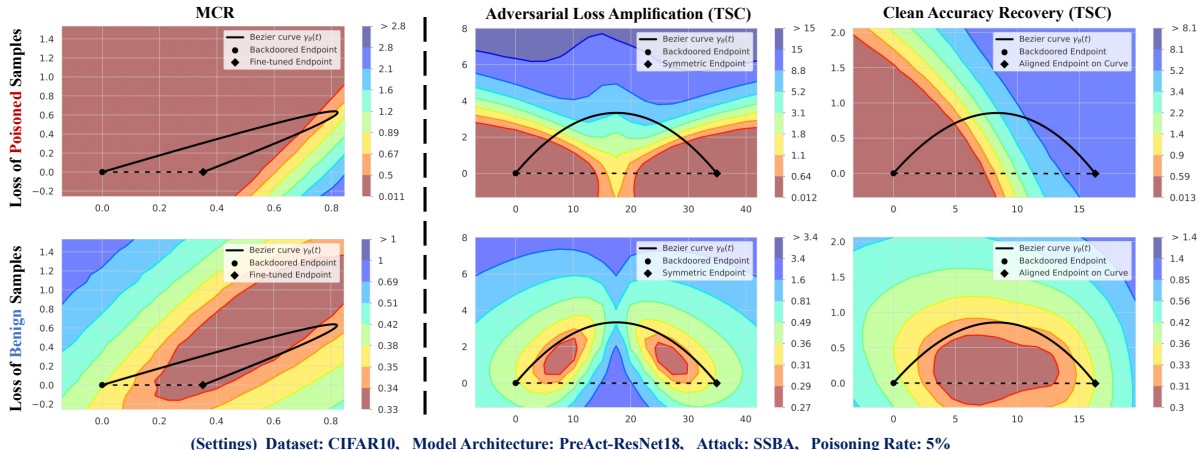

(Settings) Dataset: CIFAR10, Model Architecture: PreAct-ResNet18, Attack: SSBA, Poisoning Rate: 5%

*Figure 2.* Loss landscape for poisoned and benign samples, along with trained quadratic curves connecting distinct models. The backdoored model is a PreAct-ResNet18 trained on CIFAR-10, which contains 5% SSBA poisoned samples (Li et al., 2021c). **Left**: the curve identified by MCR. **Middle**: the curve identified by the first stage of TSC. **Right**: the curve identified by the second stage of TSC.

trast to neuron alignment in problem (5), our goal is to find the permutation $\boldsymbol{P}'_l$ for all layers to project the backdoored model into a distinct loss basin. Specifically, we formulate the following optimization problem to obtain $S(\boldsymbol{P}')$:

$$\boldsymbol{P}'_l = \arg\max_{\boldsymbol{P}_l \in \Pi_{d_l}} \sum_{i}^{n} \left\| \boldsymbol{x}^{adv}_{i,\,l} - \boldsymbol{P}_l\, \boldsymbol{x}^{adv}_{i,\,l} \right\|^2. \quad (8)$$

Then, we get the updated model $\boldsymbol{\theta}_{adv'} = \pi(\boldsymbol{\theta}_{adv}, S(\boldsymbol{P}'))$. It's important to note that this procedure does not alter the output of the backdoored model for any input; consequently, it does not increase the loss for any sample, though here we are dealing with a maximization problem. To amplify the adversarial loss, we then train a Bézier curve, $\boldsymbol{\gamma}_1$, connecting $\boldsymbol{\theta}_{adv}$ and $\boldsymbol{\theta}_{adv'}$, and subsequently select a model corresponding to a middle point on this curve.

As shown in the middle of Figure 2, when connecting $\boldsymbol{\theta}_{adv}$ and $\boldsymbol{\theta}_{adv'}$, the loss of poisoned samples increases significantly along the Bézier curve $\boldsymbol{\gamma}_1$ (*i.e.,* circumventing the backdoor space). As expected, the loss landscape becomes symmetric for both poisoned and benign samples. However, compared to MCR, we observe an increase in the loss of benign samples, implying that the models selected from the curve connecting $\boldsymbol{\theta}_{adv}$ and $\boldsymbol{\theta}_{adv'}$ would perform poorly on the initial task.

### 4.2. Clean Accuracy Recovery

To ensure that the purified model maintains better performance on benign data, we re-align the model $\boldsymbol{\theta}_t$ found on $\boldsymbol{\gamma}_1$ to the benign loss basin of the backdoored model $\boldsymbol{\theta}_{adv}$. We denote the corresponding permutation set as $\boldsymbol{P}^*$. Furthermore, empirically, when setting $t = 0.4$, the model $\boldsymbol{\theta}_t$ typically exhibits higher loss on both poisoned and benign samples. We then train another curve $\boldsymbol{\gamma}_2$ connecting $\boldsymbol{\theta}_{adv}$ and $\boldsymbol{\theta}_{t^*=0.4} = \pi(\boldsymbol{\theta}_{t=0.4}, S(\boldsymbol{P}^*))$. The right side of Figure 2 shows that the loss of poisoned samples gradually increases

along the curve from $\boldsymbol{\theta}_{adv}$ to $\boldsymbol{\theta}_{t^*=0.4}$, and the model point along this curve could also attain a lower loss on benign samples than $\boldsymbol{\theta}_{t=0.4}$. A model along the curve with a high loss for poisoned samples and a low loss for benign samples can be selected as the ideal purified model.

Additionally, as shown in Figure 3, we find that performing one round of TSC is insufficient to remove the backdoor. In practice, we slightly fine-tune the model obtained from the second stage of TSC, then use this fine-tuned model as the input for the next round to mitigate the backdoor threat step by step. It is evident that as the global epoch $E_{TSC}$ increases, the ASR of model points along the second-stage curve of TSC decreases. Empirically, setting $E_{TSC} = 3$ can effectively eliminate the backdoor while maintaining good performance on the benign task.

We give the pseudocode in Algorithm 1. Since we assume the defender only has access to a small fraction of clean samples $D_c$, both the computation of the permutation and the training of the curve are conducted over $D_c$. The function $\text{PERMUTELAYERS}(\boldsymbol{\theta}_A, \boldsymbol{\theta}_B, D, \text{OPT})$ in the pseudocode returns a model by permuting the layers of $\boldsymbol{\theta}_B$, aligning or un-aligning it with $\boldsymbol{\theta}_A$. The function $\text{FITQUADCURVE}(\boldsymbol{\theta}_A, \boldsymbol{\theta}_B, \mathcal{F}, D, e)$ returns the quadratic Bézier curve connecting $\boldsymbol{\theta}_A$ and $\boldsymbol{\theta}_B$, trained using method $\mathcal{F}$ over $D$ for $e$ epochs. Moreover, $\mathcal{F}$ can be various training methods tailored to the corresponding data format, as shown in Figure 1. The function $\text{RETRIEVEPOINT}(\boldsymbol{\gamma}_{A,B}, t)$ returns the model point along $\boldsymbol{\gamma}_{A,B}$ at index $t$, as described in Equation (2). See more details in Appendix F.

### 4.3. Theoretical Analysis of TSC

We consider two feedforward neural networks, as defined in Equation (3), with weights $\boldsymbol{\theta}_0$ and $\boldsymbol{\theta}_1$. We say that $\boldsymbol{\theta}_0$ and $\boldsymbol{\theta}_1$ are $L_2$-*norm-consisten* if they satisfy the following condition:

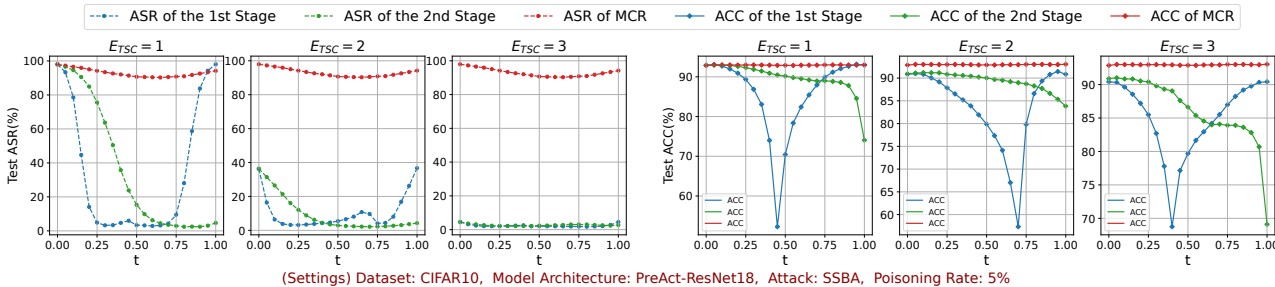

(Settings) Dataset: CIFAR10, Model Architecture: PreAct-ResNet18, Attack: SSBA, Poisoning Rate: 5%

*Figure 3.* Test attack success rate (ASR) and accuracy (ACC) on benign samples are evaluated as functions of the points along the Bézier curve found by MCR and TSC. We assess the performance against SSBA on CIFAR-10 with 5% poison rate using PreAct-ResNet18. We select model points along the curve at $t = 0.4$ for each stage and round. Since MCR only trains a single curve for model purification, we plot the results of MCR at each round of TSC for better comparison.

---

**Algorithm 1** Two-stage Symmetry Connectivity

**Require:** backdoored model $\boldsymbol{\theta}_{adv}$, clean dataset $D_c$, global epoch $E_{TSC}$, curve index $t$, curve training epoch $e$, training method $\mathcal{F}$;

**Ensure:** purified model $\boldsymbol{\theta}_p$;

    Initialize  $\boldsymbol{\theta}_p \leftarrow \boldsymbol{\theta}_{adv}$;

    **for** $i = 1$ **to** $E_{TSC}$ **do**

        ▷ Adversarial Loss Amplification

        $\boldsymbol{\theta}_{p'} \leftarrow$ PermuteLayers$(\boldsymbol{\theta}_p, \boldsymbol{\theta}_p, D_c, \text{MAX})$;

        $\boldsymbol{\gamma}_{\boldsymbol{\theta}_{p,p'}} \leftarrow$ FitQuadCurve$(\boldsymbol{\theta}_p, \boldsymbol{\theta}_{p'}, \mathcal{F}, D_c, e)$;

        $\boldsymbol{\theta}_t \leftarrow$ RetrievePoint$(\boldsymbol{\gamma}_{\boldsymbol{\theta}_{p,p'}}, t)$;

        ▷ Clean Accuracy Recovery

        $\boldsymbol{\theta}_{t^*} \leftarrow$ PermuteLayers$(\boldsymbol{\theta}_p, \boldsymbol{\theta}_t, D_c, \text{MIN})$;

        $\boldsymbol{\gamma}_{\boldsymbol{\theta}_{p,t^*}} \leftarrow$ FitQuadCurve$(\boldsymbol{\theta}_p, \boldsymbol{\theta}_{t^*}, \mathcal{F}, D_c, e)$;

        $\boldsymbol{\theta}_p \leftarrow$ RetrievePoint$(\boldsymbol{\gamma}_{\boldsymbol{\theta}_{p,t^*}}, t)$;

    **end for**

---

**Definition 4.1** (Weight $L_2$-norm consistency condition). Let $\boldsymbol{W}_l^0$ and $\boldsymbol{W}_l^1$ be the parameters of the $l^{\text{th}}$ layer of $\boldsymbol{\theta}_0$ and $\boldsymbol{\theta}_1$, respectively. We define that the parameters satisfy the condition that the $L_2$ norms of the corresponding layer weights are equal: $\|\boldsymbol{W}_l^0\|_2 = \|\boldsymbol{W}_l^1\|_2, \forall l \in \{1, 2, ..., L\}$, where the $L_2$ norm of the matrix refers to the element-wise $L_2$ norm (*i.e.,* Frobenius norm).

This condition implies that the magnitude of the weights in each layer is preserved between the two models. Consequently, both models effectively operate at the same scale, which can be critical for their comparative performance.

Next, considering three optimal independent feedforward networks with weights $\boldsymbol{\theta}_A$, $\boldsymbol{\theta}_B$ and $\boldsymbol{\theta}_C$, we can present a theorem regarding the upper bounds of the loss $\ell(\cdot)$ in Equation (1) over the quadratic curves $\boldsymbol{\gamma}_{\boldsymbol{\theta}_{A,B}}(t)$ and $\boldsymbol{\gamma}_{\boldsymbol{\theta}_{A,C}}(t)$ across dataset $D$. Before stating the theorem, we first reformulate the parametric Bézier curve $\boldsymbol{\gamma}_{\boldsymbol{\theta}_{A,B}}(t)$ by replacing $\boldsymbol{\theta}_{A,B}$ with its deviation $\tilde{\boldsymbol{\theta}}_{A,B}$:

$$\boldsymbol{\gamma}_{\tilde{\boldsymbol{\theta}}_{A,B}}(t) = (1-t)\boldsymbol{\theta}_A + t\boldsymbol{\theta}_B + 2t(1-t)\tilde{\boldsymbol{\theta}}_{A,B}. \quad (9)$$

Similarly, we can reformulate $\boldsymbol{\gamma}_{\boldsymbol{\theta}_{A,C}}(t)$ to $\boldsymbol{\gamma}_{\tilde{\boldsymbol{\theta}}_{A,C}}(t)$. Such reformulation allows us to express the Bézier curves in a considerably simpler form, thus facilitating theoretical analysis. The detailed derivation is provided in Equation (14). Moreover, we refer to the quantity $M_l(\boldsymbol{\theta}_A, \boldsymbol{\theta}_B; D)$, as defined in Equation (6), which measures the $L_2$ norm of the feature distance at the $l^{\text{th}}$ layer between models $\boldsymbol{\theta}_A$ and $\boldsymbol{\theta}_B$ for a given dataset $D$. We can then derive the following theorem:

**Theorem 4.2.** *We assume that the activation function $\sigma$ in Equation (3) and the loss function $\mathcal{L}$ in Equation (1) are Lipschitz continuous. Let $\boldsymbol{\gamma}_{\tilde{\boldsymbol{\theta}}_{A,B}}(t)$ and $\boldsymbol{\gamma}_{\tilde{\boldsymbol{\theta}}_{A,C}}(t)$ be two Bézier curves defined in Equation (9). Then, under the following conditions:*

*(1) $M_l(\boldsymbol{\theta}_A, \boldsymbol{\theta}_B; D) \leq M_l(\boldsymbol{\theta}_A, \boldsymbol{\theta}_C; D)$, $\forall l \in \{1, 2, \ldots, L\}$; (2) $\boldsymbol{\theta}_B$ and $\boldsymbol{\theta}_C$ are $L_2$-norm-consistent; (3) $\tilde{\boldsymbol{\theta}}_{A,B}$ and $\tilde{\boldsymbol{\theta}}_{A,C}$ are $L_2$-norm-consistent; there exists upper bounds $U_{A,B}$ and $U_{A,C}$ for $\ell(\cdot)$ such that: $\ell(\tilde{\boldsymbol{\theta}}_{A,B}) \leq U_{A,B}$, $\ell(\tilde{\boldsymbol{\theta}}_{A,C}) \leq U_{A,C}$, where $U_{A,B} \leq U_{A,C}$.*

*Proof.* See Appendix E.1. $\qquad\square$

A similar theorem can be found in (Tatro et al., 2020). It is important to note that the theorem demonstrated in (Tatro et al., 2020) provides only the upper bound relations between linear paths connecting aligned and unaligned models, despite Tatro et al. (2020) claiming that their theorem pertains to quadratic Bézier curves. Moreover, their theorem represents only the left inequality of Corollary 4.3, which is a special case of Theorem 4.2 and is insufficient for theoretical analysis of TSC.

**Increasing the Adversarial Loss.** Now, we redirect our focus to our method and compare the scenarios where one endpoint is fixed by model $\boldsymbol{\theta}_A$ while the other is settled by $\boldsymbol{\theta}_B$, $\boldsymbol{\theta}_{\hat{B}} = \pi(\boldsymbol{\theta}_B, S(\hat{\boldsymbol{P}}))$ or $\boldsymbol{\theta}_{B'} = \pi(\boldsymbol{\theta}_B, S(\boldsymbol{P}'))$. $S(\hat{\boldsymbol{P}})$ and $S(\boldsymbol{P}')$ correspond to the solutions in problems (6) and (8), respectively. Then, we have the following corollary:

**Corollary 4.3.** *We assume that the activation function $\sigma$ and the loss function $\mathcal{L}$ are Lipschitz continuous. Let $\gamma_{\tilde{\theta}_{A,B}}(t)$, $\gamma_{\tilde{\theta}_{A,\hat{B}}}(t)$ and $\gamma_{\tilde{\theta}_{A,B'}}(t)$ be three Bézier curves defined in Equation (9). We also assume that $\tilde{\theta}_{A,B}$, $\tilde{\theta}_{A,\hat{B}}$ and $\tilde{\theta}_{A,B'}$ are $L_2$-norm-consistent with each other. Then, there exists upper bounds $U_{A,B}$, $U_{A,\hat{B}}$ and $U_{A,B'}$ for $\ell(\cdot)$ such that: $U_{A,\hat{B}} \leq U_{A,B} \leq U_{A,B'}$.*

*Proof.* See Appendix E.2. □

Intuitively, to increase the loss with respect to poison samples along the curve, we can project the copy of backdoored model $\theta_{adv}$ to loss basin different from that of the original. Theoretically, Corollary 4.3 implies that applying the permutation by $S(P')$ can enlarge the upper bound of the backdoor loss over the curve, resulting in a more robust model on the curve when faced with attack samples.

However, Corollary 4.3 also indicates that permuting the layers via $S(P')$, leads to a looser upper bound for the loss on benign samples compared to models that reside in the same loss basin. Therefore, an extended method is needed to ensure that the purified model maintains strong performance on clean samples.

**Reducing the Clean Loss.** In the second stage, we train another curve connecting the origianl backdoored model and re-aligned model found previously. As demonstrated in Corollary 4.3, re-aligning the model found on the curve to the original loss basin of the backdoored model can reduce the upper bound of loss value along the curve. When aligning the model with benign samples, the curve is more likey to be trained in a loss basin of benign data rather than poisoned ones. Thus, the second stage successfully improves the performance on benign task.

Moreover, let $\theta_{adv*}$ be a model satisfies the following three conditions: (1) it is aligned with $\theta_{adv}$; (2) it is $L_2$-norm-consistent with the model $\theta_{t*=0.4}$; and (3) it achieves higher accuracy on the poisoned dataset $D_{adv}$ than $\theta_{t*=0.4}$. Also, consider a curve $\gamma_{adv}$ that connects $\theta_{adv*}$ and $\theta_{adv}$. We can say that the 2-Wasserstein distance $W_2(\mathbb{P}_l^{adv}, \mathbb{P}_l^{adv*}; D_{adv})$ is smaller than $W_2(\mathbb{P}_l^{adv}, \mathbb{P}_l^{t=0.4}; D_{adv})$ for each layer, as $\theta_{adv}$ is more functionally identical to $\theta_{adv}$ over $D_{adv}$. Thus, we have $M_l(\theta_{adv}, \theta_{adv*}; D_{adv}) \leq M_l(\theta_{adv}, \theta_{t*=0.4}; D_{adv})$. According to Theorem 4.2, if the parameters of $\gamma_{adv}$ and $\gamma_2$ are $L_2$-norm-consistent, we conclude that the upper bound of $\gamma_2$ is higher than that of $\gamma_{adv}$ over the poisoned dataset. This finding implies that a high loss can be maintained for poisoned samples along the curve $\gamma_2$.

Figure 3 compares model performance along curves identified by MCR and TSC. For each epoch of TSC, as $t$ approaches 0.5, both ACC and ASR of model points from the first stage decrease more rapidly than those of MCR and TSC's second stage. Besides, model points from the sec-

ond stage exhibit significantly lower ASR than MCR while maintaining moderate ACC decline compared to the first stage. These observations support our theoretical analysis.

## 5. Experiments

### 5.1. Experimental Settings

**Attack Setup.** (1) **Supervised Learning.** We consider eleven typical backdoor attacks, including eight label-flipping attacks (BadNet (Gu et al., 2017), Blended (Chen et al., 2017), SSBA (Li et al., 2021c), LF (Zeng et al., 2021), WaNet (Nguyen & Tran, 2021), Inputaware (Nguyen & Tran, 2020), SBL (Pham et al., 2024) and SAPA (He et al., 2024)) and three clean label attacks (LC (Turner et al., 2019), SIG (Barni et al., 2019) and Narcissus (Zeng et al., 2023)). These attacks are conducted on CIFAR10 (Krizhevsky, 2009) using PreAct-ResNet18 (He et al., 2016a) and ImageNet100 (Deng et al., 2009) using ResNet50 (He et al., 2016b) with various poisoning rates. (2) **Self-Supervised Learning.** We evaluate two self-supervised learning attacks: BadEncoder (Jia et al., 2022) and CTRL (Li et al., 2023). BadEncoder is conducted using two typical SSL training methods, SimCLR (Chen et al., 2020) and CLIP (Radford et al., 2021). For SimCLR, we utilize publicly available backdoored ResNet18 and ResNet50 encoders on CIFAR10 and ImageNet, respectively, evaluating ACC and ASR through linear probing (Alain & Bengio, 2017) on downstream datasets STL10 (Coates et al., 2011), GTSRB, and SVHN (Netzer et al., 2011). For CLIP, following (Jia et al., 2022), we fine-tune a pre-trained CLIP ResNet50[1] on ImageNet100 using Sim-CLR to inject backdoors, and assess performance through linear probing and zero-shot evaluation on STL10, Food101 (Bossard et al., 2014), and VOC2007 (Everingham et al.). The CTRL attack is evaluated using the same SimCLR settings. Implementation details are provided in Appendix L.

**Defense Setup.** In our experiments, we focus on post-purification defenses methods and provide all defenses with 5% of the clean training dataset, except for the defenses for the CLIP model. To address the backdoored CLIP visual model, we employ the entire MS-COCO dataset [2] (Lin et al., 2014). (1) **Supervised Learning.** We evaluate six post-purification defenses methods as baselines: FP (Liu et al., 2018), NC (Wang et al., 2019), MCR (Zhao et al., 2020), ANP (Wu & Wang, 2021), FT-SAM (Zhu et al., 2023), I-BAU (Zeng et al., 2022), and SAU (Wei et al., 2023). For TSC, we set the global epoch $E_{TSC} = 3$, curve index $t = 0.4$, and curve training epoch $e = 200$. (2) **Self-Supervised Learning.** We consider MCR (Zhao et al., 2020) and SSL-

---

[1]https://github.com/openai/CLIP

[2]As the dataset used to train CLIP (Radford et al., 2021) is not publicly available and involves 400M images, we conduct defenses with the much smaller MS-COCO-2017 dataset (Lin et al., 2014), which contains approximately 120K images with 5 captions each.

*Table 1.* Results on CIFAR10 and ImageNet100 under supervised learning scenarios. Attack Success Rates (ASRs) below 15% are highlighted in blue to indicate a successful defense, while ASRs above 15% are denoted in red as failed defenses.

| Dataset | Attacks | Poison Rate | No Defense ACC(↑) | ASR(↓) | FP ACC(↑) | ASR(↓) | NC ACC(↑) | ASR(↓) | MCR ACC(↑) | ASR(↓) | ANP ACC(↑) | ASR(↓) | FT-SAM ACC(↑) | ASR(↓) | I-BAU ACC(↑) | ASR(↓) | SAU ACC(↑) | ASR(↓) | TSC (ours) ACC(↑) | ASR(↓) |
|---|---|---|---|---|---|---|---|---|---|---|---|---|---|---|---|---|---|---|---|---|
| CIFAR10 | BadNet | 5% | 92.64 | 88.74 | 92.26 | 1.17 | 90.53 | 1.01 | 92.17 | 7.62 | 86.45 | 0.02 | 92.19 | 3.50 | 88.66 | 0.92 | 89.32 | 1.74 | 89.19 | 1.90 |
| | | 1% | 93.14 | 74.73 | 92.59 | 2.29 | 92.07 | 0.77 | 92.90 | 18.06 | 85.82 | 0.04 | 92.39 | 1.57 | 87.80 | 2.29 | 65.38 | 2.06 | 90.71 | 1.26 |
| | Blended | 5% | 93.66 | 99.61 | 92.70 | 49.47 | 93.67 | 99.61 | 93.23 | 99.01 | 88.95 | 18.76 | 93.00 | 29.59 | 88.07 | 34.86 | 90.69 | 7.74 | 90.14 | 10.53 |
| | | 1% | 93.76 | 94.88 | 92.92 | 69.74 | 93.76 | 94.88 | 93.62 | 93.10 | 89.69 | 60.52 | 93.00 | 49.36 | 89.62 | 25.74 | 90.02 | 36.16 | 91.12 | 12.46 |
| | LF | 5% | 93.36 | 98.03 | 92.84 | 59.12 | 90.98 | 2.43 | 93.07 | 97.32 | 84.20 | 2.46 | 92.89 | 7.44 | 88.64 | 45.66 | 90.60 | 1.71 | 88.50 | 3.78 |
| | | 1% | 93.56 | 86.44 | 92.45 | 65.80 | 93.56 | 86.46 | 93.09 | 84.11 | 86.27 | 11.28 | 93.47 | 11.71 | 90.53 | 69.28 | 91.58 | 18.12 | 90.68 | 11.67 |
| | SSBA | 5% | 93.27 | 94.91 | 92.55 | 16.27 | 93.27 | 94.91 | 92.94 | 92.06 | 88.72 | 0.13 | 92.71 | 2.87 | 89.65 | 1.54 | 91.30 | 2.06 | 89.43 | 2.18 |
| | | 1% | 93.43 | 73.44 | 93.01 | 7.68 | 91.60 | 0.46 | 93.33 | 65.88 | 85.33 | 0.31 | 93.02 | 1.49 | 89.56 | 4.87 | 91.38 | 0.99 | 91.18 | 2.18 |
| | SBL-BadNet | 5% | 90.79 | 93.48 | 92.59 | 1.13 | 92.26 | 0.59 | 92.26 | 91.82 | 82.82 | 51.63 | 92.16 | 60.03 | 90.67 | 27.06 | 91.31 | 0.60 | 91.02 | 1.12 |
| | | 1% | 91.71 | 88.64 | 93.10 | 31.77 | 91.82 | 0.72 | 93.23 | 86.11 | 82.71 | 81.48 | 92.77 | 59.58 | 90.63 | 2.00 | 92.32 | 1.01 | 91.54 | 1.93 |
| | SAPA | 1% | 94.01 | 99.97 | 92.34 | 92.22 | 92.80 | 2.14 | 93.83 | 100.00 | 86.06 | 92.68 | 93.06 | 79.80 | 86.69 | 15.17 | 91.83 | 1.96 | 90.37 | 7.41 |
| | | 0.5% | 93.77 | 84.80 | 88.82 | 82.76 | 92.74 | 1.52 | 93.78 | 80.83 | 87.99 | 81.52 | 93.23 | 82.02 | 90.16 | 26.48 | 91.75 | 0.68 | 90.98 | 7.32 |
| | LC | 5% | 93.31 | 98.33 | 92.19 | 72.99 | 92.32 | 0.64 | 92.94 | 99.94 | 88.15 | 13.83 | 92.59 | 57.18 | 90.15 | 1.99 | 91.53 | 1.50 | 90.04 | 2.38 |
| | | 1% | 93.79 | 75.93 | 92.86 | 29.86 | 92.31 | 0.68 | 93.67 | 82.54 | 86.58 | 31.46 | 92.83 | 39.40 | 89.78 | 0.71 | 92.16 | 3.77 | 90.08 | 5.78 |
| | Narcissus | 1% | 93.68 | 82.87 | 92.29 | 44.88 | 93.68 | 47.87 | 93.61 | 49.79 | 92.01 | 27.01 | 93.05 | 26.80 | 90.21 | 18.67 | 91.36 | 3.24 | 90.65 | 7.88 |
| | | 0.5% | 93.68 | 80.58 | 92.94 | 29.59 | 93.67 | 32.57 | 93.69 | 32.96 | 89.35 | 16.78 | 93.06 | 14.08 | 89.16 | 21.09 | 91.74 | 5.81 | 91.71 | 8.02 |
| ImageNet100 | BadNet | 0.5% | 84.30 | 99.78 | 83.36 | 9.80 | 81.92 | 0.52 | 85.24 | 99.66 | 78.44 | 94.18 | 83.70 | 9.45 | 73.70 | 8.34 | 73.86 | 0.28 | 80.20 | 0.22 |
| | | 1% | 84.56 | 99.86 | 83.10 | 9.58 | 85.08 | 99.86 | 85.08 | 99.86 | 79.48 | 93.64 | 83.88 | 24.14 | 71.46 | 43.66 | 72.84 | 0.26 | 78.06 | 0.14 |
| | Blended | 0.5% | 84.44 | 94.32 | 82.80 | 63.25 | 84.44 | 94.32 | 85.58 | 94.97 | 84.56 | 93.27 | 83.40 | 75.43 | 74.22 | 62.34 | 73.84 | 3.72 | 76.58 | 12.63 |
| | | 1% | 84.90 | 98.04 | 83.36 | 69.21 | 80.21 | 70.21 | 85.04 | 97.58 | 84.54 | 97.70 | 83.86 | 82.00 | 73.10 | 61.25 | 69.24 | 0.53 | 75.88 | 6.35 |
| | LF | 0.5% | 84.24 | 98.87 | 83.10 | 50.26 | 84.24 | 98.87 | 85.70 | 97.70 | 81.32 | 86.20 | 83.80 | 70.48 | 74.36 | 74.97 | 75.22 | 0.18 | 78.78 | 5.39 |
| | | 1% | 83.92 | 99.56 | 83.00 | 35.82 | 76.76 | 49.87 | 85.30 | 99.03 | 81.10 | 88.53 | 83.40 | 70.69 | 71.06 | 22.32 | 67.38 | 2.93 | 78.58 | 5.41 |
| | SSBA | 0.5% | 84.30 | 95.31 | 83.34 | 46.75 | 84.30 | 95.31 | 85.04 | 95.13 | 76.96 | 6.18 | 83.16 | 15.70 | 71.52 | 1.19 | 76.12 | 0.89 | 79.56 | 1.45 |
| | | 1% | 84.02 | 99.43 | 83.34 | 59.68 | 78.47 | 70.78 | 85.14 | 97.72 | 80.22 | 22.77 | 83.50 | 20.30 | 72.58 | 7.45 | 73.94 | 0.36 | 79.88 | 4.91 |
| | SBL-Blended | 0.5% | 72.52 | 97.56 | 85.10 | 89.29 | 72.52 | 97.56 | 82.54 | 92.63 | 68.28 | 97.05 | 83.84 | 89.07 | 70.78 | 37.35 | 73.66 | 14.87 | 77.18 | 7.18 |
| | | 1% | 72.68 | 99.17 | 83.41 | 68.24 | 71.42 | 70.14 | 82.82 | 95.78 | 72.72 | 99.15 | 83.92 | 92.69 | 73.52 | 20.30 | 76.78 | 39.29 | 77.16 | 8.87 |
| | SAPA | 0.5% | 85.04 | 98.83 | 83.44 | 20.53 | 78.57 | 9.20 | 85.60 | 93.05 | 80.42 | 96.59 | 83.82 | 30.04 | 69.32 | 18.34 | 73.34 | 1.07 | 79.00 | 1.74 |
| | | 1% | 85.50 | 98.83 | 83.34 | 27.86 | 77.42 | 3.52 | 85.42 | 95.80 | 83.12 | 93.76 | 83.96 | 45.88 | 69.12 | 41.64 | 75.42 | 1.23 | 78.14 | 1.41 |
| | LC | 0.5% | 84.22 | 0.61 | 83.34 | 0.20 | 84.22 | 0.61 | 85.04 | 0.93 | 84.48 | 0.57 | 83.86 | 0.24 | 69.72 | 0.36 | 76.20 | 0.16 | 80.36 | 0.22 |
| | | 1% | 84.10 | 32.97 | 83.48 | 4.28 | 81.13 | 0.42 | 85.48 | 76.75 | 84.22 | 32.42 | 83.58 | 8.06 | 73.52 | 1.76 | 70.50 | 0.79 | 80.18 | 0.57 |
| | SIG | 0.5% | 84.20 | 16.22 | 83.26 | 2.22 | 78.48 | 0.43 | 85.18 | 18.34 | 84.00 | 15.86 | 83.88 | 4.51 | 70.76 | 3.43 | 75.22 | 0.12 | 77.20 | 0.55 |
| | | 1% | 84.16 | 70.08 | 83.40 | 20.48 | 79.58 | 0.89 | 85.02 | 77.84 | 80.40 | 65.68 | 83.36 | 44.81 | 70.98 | 19.98 | 73.76 | 0.30 | 80.22 | 9.98 |

Cleanse (Zheng et al., 2024) as baselines. For TSC, we set the global epoch $E_{TSC} = 2$, curve index $t = 0.25$, and curve training epoch $e = 200$ for both SimCLR and CLIP. Comprehensive settings for all defenses are provided in Appendix L. For the following experiments, we consider a defense against an attack successful if the ASR is reduced to below 15%.

## 5.2. Results for Supervised Learning

Table 1 compares the performance of TSC with existing defenses on CIFAR10 and ImageNet100 under supervised learning scenarios. Comprehensive results for GTSRB and of other attack settings are provided in Appendix I.

It's clear that TSC successfully reduces the ASR to below 15% for all attacks on CIFAR10 GTSRB and ImageNet100, demonstrating its robustness and effectiveness. Among all attacks, Blended attack with small poisoning rates proves most challenging to defend against. While other defenses struggle to contain ASRs below 25%, TSC reduces the ASR to 12.46% on CIFAR10 (1% poisoning rate) and 12.63% on ImageNet100 (0.5% poisoning rate). SAU shows strong backdoor removal capabilities against most attacks but suffers from ACC instability. For example, under BadNet Attack with 1% poisoning rate on CIFAR10, SAU's accuracy falls below 70%. We attribute this behavior to SAU's aggressive unlearning strategy, which leads to catastrophic decreases in both ACC and ASR. ANP, FT-SAM, and I-BAU also demonstrate effectiveness, reducing ASRs to below 15% for most attacks on CIFAR10 and GTSRB. However,

their performance diminishes on ImageNet100, particularly against Blended, Inputaware and LF attacks. While TSC occasionally yields lower initial task accuracy than FT-SAM, it outperforms ANP and I-BAU in most cases.

To further explore the impact of TSC on ACC, we provide results for ACC drops on non-backdoored models in Appendix K. These results indicate that the ACC drop for TSC is acceptable in scenarios without data poisoning.

## 5.3. Results for Self-supervised Learning

Tables 2 and 3 show the results against BadEncoder attacks under SimCLR and CLIP training scenarios, respectively. We employ different settings for SimCLR and CLIP to showcase the flexibility of TSC. For SimCLR, encoders are trained and backdoored independently with specific target labels for each downstream task. For CLIP, we backdoor a single visual encoder using 'truck' as the target label/caption and evaluate performance before and after defenses across downstream tasks. Since Food101 and VOC2007 lack the 'truck' label, we augment their training sets with truck images from STL10 for evaluation of linear probing. Results indicate the attack remains effective even without targeting specific downstream datasets. More results for CTRL attack and SSL-Cleanse are provided in Appendix J.

Under SimCLR, while MCR successfully removes backdoors targeting STL10 and GTSRB downstream tasks, it fails against attacks targeting the SVHN dataset. In contrast, TSC effectively reduces the ASR to below 11% across

*Table 2.* Defense results under SimCLR training scenario, where linear probing is used to evaluate the downstream tasks.

| Pre-training Dataset | Downstream Dataset | No Defense | | MCR | | TSC (ours) | |
|---|---|---|---|---|---|---|---|
| | | ACC(↑) | ASR(↓) | ACC(↑) | ASR(↓) | ACC(↑) | ASR(↓) |
| CIFAR10 | STL10 | 76.74 | 99.65 | 74.93 | 7.92 | 71.11 | 4.44 |
| | GTSRB | 81.12 | 98.79 | 75.51 | 0.54 | 77.57 | 1.68 |
| | SVHN | 63.12 | 98.71 | 57.35 | 65.58 | 64.13 | 10.26 |
| ImageNet | STL10 | 94.93 | 98.99 | 90.20 | 2.08 | 86.99 | 3.11 |
| | GTSRB | 75.94 | 99.76 | 72.38 | 0.13 | 69.47 | 6.47 |
| | SVHN | 72.64 | 99.21 | 71.27 | 34.15 | 66.44 | 3.64 |

*Table 3.* Defense results under CLIP training scenario, where linear probing and zero-shot learning are used to evaluate the downstream tasks.

| Pre-training Dataset | Downstream Dataset | No Defense | | MCR | | TSC (ours) | |
|---|---|---|---|---|---|---|---|
| | | ACC(↑) | ASR(↓) | ACC(↑) | ASR(↓) | ACC(↑) | ASR(↓) |
| CLIP (linear probe) | STL10 | 97.07 | 99.33 | 96.43 | 99.86 | 94.15 | 0.67 |
| | Food101 | 72.58 | 97.91 | 72.36 | 96.62 | 69.33 | 1.04 |
| | VOC 2007 | 76.07 | 99.83 | 75.47 | 99.92 | 78.42 | 0.34 |
| CLIP (zero-shot) | STL10 | 94.06 | 99.86 | 91.51 | 99.85 | 90.25 | 0.88 |
| | Food101 | 67.72 | 99.96 | 66.51 | 99.56 | 61.69 | 0.28 |
| | VOC 2007 | 71.22 | 99.92 | 70.09 | 99.12 | 75.08 | 1.45 |

all downstream tasks. For CLIP, TSC achieves remarkable performance, reducing ASR below 2%, while MCR proves ineffective against BadEncoder attacks.

### 5.4. Resistance to Potential Adaptive Attacks

The previous experiments demonstrate the effectiveness of TSC against existing backdoor attacks. However, it is essential to consider adpative attacks against TSC. The core defense mechanism of TSC relies on increasing adversarial loss along the quadratic Bézier curve by projecting model $\theta_{adv}$ to a distinct loss basin to find $\theta_{adv'}$ during the first stage. An adaptive attack would attempt to train and backdoor a model that maintains low backdoor loss along the curve identified by TSC.

To design such an adaptive attack, we build upon the neural network subspace learning approach proposed in (Wortsman et al., 2021), originally developed for improving accuracy and calibration. Our strategy involves simultaneously training a curve and updating its endpoints $\theta_{adv}$ and $\theta_{adv'}$ using the mixture of benign and poisoned data (*i.e.*, learning a backdoored subspace). After training, we select one endpoint as the final model. We convert all previously evaluated attacks into adaptive versions under both supervised and self-supervised learning settings. Experimental results reveal that TSC remains robust against such attack. The implementation details and results are provided in Appendix G.

To further validate the robustness of TSC against adaptive attacks, we present corresponding loss landscape visualizations in Appendix G.3. The analysis reveals that the combination of the permutation mechanism and training exclusively with benign samples contributes to amplifying the loss on poisoned samples, even for adaptive attacks. Moreover, given that our defense involves the concept of loss landscapes, we also conduct experiments against more

advanced attacks, including SBL (Pham et al., 2024), Narcissus (Zeng et al., 2023), and SAPA (He et al., 2024). These modern backdoor attacks aim to optimize flatter loss landscapes or entangle benign and backdoor features. The results demonstrate that TSC effectively reduces the ASR to below 15% even against these sophisticated attacks.

### 5.5. Ablation Studies

We conduct comprehensive ablation studies to analyze the impact of TSC's key hyperparameters: the number of global epochs $E_{TSC}$ and the curve index $t$. Through extensive experiments, we find that increasing $t$ (up to 0.5) and $E_{TSC}$ improves backdoor removal performance but meanwhile reducing accuracy on benign samples. Moreover, a larger $E_{TSC}$ leads to more stable performance. Based on empirical results, we recommend $t = 0.4$ and $E_{TSC} = 3$ for supervised learning scenarios, and $t = 0.25$ and $E_{TSC} = 2$ for self-supervised learning. Detailed analysis and additional experimental results can be found in Appendix H.1.

To validate the stability of TSC, we conduct experiments with VGG19-BN (Simonyan & Zisserman, 2014) and InceptionV3 (Szegedy et al., 2016) on CIFAR10. Results in Appendix H.2 show the robustness of TSC across different model architectures.

We opt to use the same $t$ for both stages to maintain a simpler parameter design. Employing distinct $t$ values for each stage would lead to numerous parameter combinations, potentially complicating the algorithm's overall structure. Moreover, as shown in Figure 3, the ACC/ASR values in the first stage exhibit a roughly symmetric pattern with respect to $t$, whereas in the second stage, they decrease as $t$ increases. Although the overall trends for the two stages differ across $t \in [0, 1]$, they both demonstrate a decreasing tendency within $t \in [0, 0.5]$. Notably, in Stage 2, ASR decreases effectively while ACC remains high for $t$ values near 0.5. Considering this observation from Stage 2, the symmetry in Stage 1, and the consistent trend within $t \in [0, 0.5]$, we suggest selecting $t$ from this range for both stages.

### 6. Conclusion and Limitation

In this paper, we propose TSC, a novel defense mechanism leveraging permutation invariance. Unlike previous post-purification defenses, TSC utilizes weight symmetry to remove backdoors and is applicable to both supervised and self-supervised learning scenarios, with potential extensions to other learning paradigms. Our experiments demonstrate the robustness of TSC under diverse attack settings, achieving comparable or superior performance to existing defenses. However, TSC occasionally trades off accuracy on benign samples for backdoor removal. Future work could focus on optimizing such trade-off to maintain high ASR reduction while improving ACC.

## Acknowledgements

This work was supported in part by the National Key Research and Development Program of China (Grant No. 2024YFB31NL00101), the National Natural Science Foundation of China (Grant No. U22A2036), and the National Natural Science Foundation of China (Grant No. 62472122).

## Impact Statement

This paper aims to advance the field of Machine Learning Security by proposing a novel purification method for removing implanted backdoors. The potential societal benefits include providing a framework for eliminating backdoor behavior across various machine learning scenarios, thereby enhancing model security. However, new attacks targeting our method may emerge in the future. Further work is needed to validate the effectiveness of our approach on a broader scale.

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

# A. Applicability of the Defense Setting of TSC

As stated in Section 3.4, this paper considers two attack scenarios: one where an adversary can only poison a portion of the training data, and another where the adversary gains control over the training procedure. In our defense setting, we assume that the defender has knowledge of the original basic training methods such as Stochastic Gradient Descent (SGD) using a Cross-Entropy loss function. To address potential concerns about the practical relevance of this defense setting, we provide here a detailed discussion of how our proposed defense approach applies in real-world scenarios.

## A.1. Data-poisoning Attacks

For adversaries employing data poisoning (who have no control over the training process), defenders may repair the backdoored model post-training using defenses such as TSC or other post-purification techniques. In this scenario, as the defender has control of the training it's natural the defender has the knowledge of the training procedure. Thus, the setting in our paper is applicable in such data poison scenario.

## A.2. Training-manipulation Attack

For adversaries have control over the training, we provides two common examples here:

- Public Pre-trained Models: Public repositories or research papers release pre-trained models that may contain backdoors. Since these sources typically provide detailed descriptions of the model-training procedure, defenders can leverage this information to apply TSC effectively. Using public large-scale image encoders for downstream tasks is increasingly common, making our setting practically relevant. Advanced zero-shot deployment models (*e.g.*, CLIP) further exemplify this applicability.

- Internal Adversary in Organizations: Consider an internal adversary scenario within an organization where malicious attackers backdoor a model without others' awareness. Typically, benign team members possess knowledge of the basic training process but lack insight into the malicious manipulations. In this context, defenders within the organization can deploy TSC to purify the model without taking retraining from scratch.

Moreover, beyond traditional learning scenarios, our method shows potential for application in other settings. For example, in federated learning, models are trained collaboratively across many distributed devices, with participants computing and sending their local gradients for global aggregation. Malicious participants could inject poisoned updates into this system, thereby introducing backdoors, even without direct access to the overall training procedure. Since a common underlying training methodology is typically employed by both clients and the server in such federated architectures, defenders in this setting could apply our method to remove backdoors using only a small amount of data.

# B. Quadratic Mode Connectivity

In Section 3.1, we can find a path connecting $\boldsymbol{\theta}_A$ and $\boldsymbol{\theta}_B$ using Equation (1). However, since $p_{\boldsymbol{\theta}_{A,B}}(t)$ depends on $\boldsymbol{\theta}_{A,B}$, it is intractable to compute the stochastic gradients of $\ell(\boldsymbol{\theta}_{A,B})$ in Equation (1). To address this, Garipov Garipov et al. (2018) choose the uniform distribution $U(0,1)$ over the interval $[0,1]$ to replace $p_{\boldsymbol{\theta}_{A,B}}(t)$, leading to the following loss:

$$\ell'(\boldsymbol{\theta}_{A,B}) = \int_0^1 \mathcal{L}(\boldsymbol{\gamma}_{\boldsymbol{\theta}_{A,B}}(t))\,\mathrm{d}t = \mathbb{E}_{t\sim U(0,1)}\mathcal{L}(\boldsymbol{\gamma}_{\boldsymbol{\theta}_{A,B}}(t)), \tag{10}$$

The primary contrast between (1) and (10) is that the former calculates the average loss $\mathcal{L}(\boldsymbol{\gamma}_{\boldsymbol{\theta}_{A,B}}(t))$ over a uniform distribution along the curve, while the latter calculates the average loss over a uniform distribution within the interval $[0,1]$ for the variable $t$. To minimize $\ell'(\boldsymbol{\theta}_{A,B})$, at each step one can randomly select a sample $\hat{t}$ from the uniform distribution over the interval $[0,1]$ and update the value of $\boldsymbol{\theta}_{A,B}$ based on the gradient of the loss function $\mathcal{L}(\boldsymbol{\gamma}_{\boldsymbol{\theta}_{A,B}}(\hat{t}))$. This implies that we can use $\nabla_{\boldsymbol{\theta}_{A,B}}\mathcal{L}(\boldsymbol{\gamma}_{\boldsymbol{\theta}_{A,B}}(\hat{t}))$ to estimate the actual gradient of $\ell'(\boldsymbol{\theta}_{A,B})$,

$$\nabla_{\boldsymbol{\theta}_{A,B}}\ell'(\theta) = \nabla_{\boldsymbol{\theta}_{A,B}}\mathbb{E}_{t\sim U(0,1)}\mathcal{L}(\boldsymbol{\gamma}_{\boldsymbol{\theta}_{A,B}}(t)) \tag{11}$$

$$= \mathbb{E}_{t\sim U(0,1)}\nabla_{\boldsymbol{\theta}_{A,B}}\mathcal{L}(\boldsymbol{\gamma}_{\boldsymbol{\theta}_{A,B}}(t)) \tag{12}$$

$$\simeq \nabla_{\boldsymbol{\theta}_{A,B}}\mathcal{L}(\boldsymbol{\gamma}_{\boldsymbol{\theta}_{A,B}}(\hat{t})). \tag{13}$$

We can choose the **Bézier curve** as the basic parametric function to characterize the parametric curve $\gamma_{\theta_{A,B}}(t)$. And we could initialize $\theta_{A,B}$ with $\frac{1}{2}(\theta_A + \theta_B)$. A Bézier curve provides a convenient parametrization of smooth paths with given endpoints. We can reform the parametric Bézier curve $\gamma_{\theta_{A,B}}$ in Equation (2) by replacing $\theta_{A,B}$ with its deviation $\tilde{\theta}_{A,B}$:

$$
\begin{aligned}
\gamma_{\theta_{A,B}}(t) &= (1-t)^2 \theta_A + 2t(1-t)\theta_{A,B} + t^2 \theta_B \\
&= (1-t)^2 \theta_A + 2t(1-t)(\frac{\theta_A + \theta_B}{2} + \tilde{\theta}_{A,B}) + t^2 \theta_B \\
&= (1-t)\theta_A + t\theta_B + 2t(1-t)\tilde{\theta}_{A,B} = \gamma_{\tilde{\theta}_{A,B}}(t).
\end{aligned}
\tag{14}
$$

## C. Permutation Invariance and Neuron Alignment

After applying the permutation operation $\pi(\theta, S(P))$, the feedforward neural network defined in Equation (3) is transformed to:

$$
\begin{aligned}
y &= W_L \circ \sigma \circ P_{L-1}^\top P_{L-1} W_{L-1} \circ ... \circ \sigma \circ P_1^\top P_1 W_1 x_0 \\
&= W_L P_{L-1}^\top \circ \sigma \circ P_{L-1} W_{L-1} P_{L-2}^\top \circ ... \circ \sigma \circ P_1 W_1 x_0.
\end{aligned}
\tag{15}
$$

The second equation in (15) follows from the fact that $\sigma$ is an element-wise function. The weights of the permuted network can then be obtained as defined in Equation (4).

Previous studies have found that two functionally identical networks $\theta_A$ and $\theta_B$, trained independently with the same architecture but different random initializations or SGD solutions, could be misaligned. And the loss of their linearly interpolated network, represented by $\theta_t = t\theta_A + (1-t)\theta_B$ (where $0 \leq t \leq 1$), could be quite large (Entezari et al., 2022; Frankle et al., 2020). However, Entezari et al. (2022) conjecture that if the permutation invariance of neural networks is taken into account, then networks obtained by all SGD solutions could be linearly connected.

Recently, reserach (Singh & Jaggi, 2020; Ainsworth et al., 2023; Jordan et al., 2023) has shown that such misaligned networks $\theta_A$ and $\theta_B$ could be projected to the same loss basin using a specific set of permutation matrices $S(\hat{P})$. Subsequently, these networks can be fused through a linear path, *i.e.,* they are linear mode connected. For example, one can let $\theta_B$ re-aligned with $\theta_A$ by permuting $\theta_B$ to $\theta_B^{S(\hat{P})}$, enabling the linearly interpolated network of $\theta_A$ and $\theta_B^{S(\hat{P})}$ to exhibit performance similar to both $\theta_A$ and $\theta_B$.

To solve the problem defined in Equation (5), various cost functions $c_l$ can be employed to compute $\hat{P}_l$ (Li et al., 2015; Singh & Jaggi, 2020; Ainsworth et al., 2023). One commonly used $c_l$ is defined as $c_l = 1 - \text{corr}(v, z)$ in (Li et al., 2015), where the corr compute the correlation between $v \in \mathbb{R}^{d_l}$ and $z \in \mathbb{R}^{d_l}$. Singh & Jaggi (2020) utilized optimal transport to soft-align neurons before model fusion. Ainsworth et al. (2023) introduced two novel alignment algorithm and compared them with the method in (Li et al., 2015). Jordan et al. (2023) proposed enhancing the linear mode connectivity after alignment by renormalizing the activations. While these studies aimed to apply neuron alignment for linear interpolated networks, Tatro et al. (2020) found that alignment could improve both robustness and accuracy along the quadratic curve connecting adversarially robust models. Following previous work, we continue to use this alignment method proposed by Li et al. (2015).

## D. Wasserstein distance

Wasserstein distance, a key concept in optimal transport (OT) theory, measures the distance between probability distributions by considering the cost of transforming one distribution into another (Villani et al., 2009). It provides a geometric perspective on comparing distributions. Formally, let $\mathbb{P}_r$ and $\mathbb{P}_q$ be two probability distributions, and $\Omega(\mathbb{P}_r, \mathbb{P}_r)$ denote the set of all joint distributions $\omega$ that have $\mathbb{P}_r$ and $\mathbb{P}_q$ as their marginal distributions. Then the $p$-Wasserstein distance can be expressed as:

$$
W_p(\mathbb{P}_r, \mathbb{P}_q) = \left( \inf_{\omega \in \Omega(\mathbb{P}_r, \mathbb{P}_q)} \mathbb{E}_{(x,y) \sim \omega} \|x - y\|^p \right)^{1/p}.
\tag{16}
$$

The joint distribution $\omega$ can be regarded as the optimal transport solution (*i.e.,* the minimal distance) between $\mathbb{P}_r$ and $\mathbb{P}_q$. When we solve the problem outlined in Equation (6) to find the optimal permutation $\hat{P}_l$, we are essentially seeking the optimal transport $\omega \in \Pi_{d_l}$ that corresponds to the 2-Wasserstein distance between $\mathbb{P}_l^A$ and $\mathbb{P}_l^B$.

# E. Omitted Proofs

### E.1. Proof of Theorem 4.2

*Proof.* For clarity, we first introduce some notation about $\tilde{\boldsymbol{\theta}}_{A,B}(t)$. These notations can also be applied to $\tilde{\boldsymbol{\theta}}_{A,C}(t)$ by substituting appropriate symbols.

We use $\boldsymbol{\theta}_{A,B}(t)$ to denote the point sampled from the curve $\boldsymbol{\gamma}_{\tilde{\boldsymbol{\theta}}_{A,B}}(t)$. Meanwhile, we consider a dataset containing a single sample $\boldsymbol{x}_0$. Then, $\forall l \in \{1, 2, \ldots, L\}$, the condition can be reformulated as:

$$M_l(\boldsymbol{\theta}_A, \boldsymbol{\theta}_B; \boldsymbol{x}_0) \leq M_l(\boldsymbol{\theta}_A, \boldsymbol{\theta}_C; \boldsymbol{x}_0). \tag{17}$$

Moreover, we use $\tilde{\boldsymbol{W}}_l^{A,B}$ to indicate the weight of the $l^{\text{th}}$ layer of $\tilde{\boldsymbol{\theta}}_{A,B}$. This allows us to construct a feedforward network respect to $\boldsymbol{\theta}_{A,B}(t)$:

$$\boldsymbol{f}(\boldsymbol{x}_0, \boldsymbol{\theta}_{A,B}(t)) = \left((1-t)\boldsymbol{W}_L^A + t\boldsymbol{W}_L^B + 2t(1-t)\tilde{\boldsymbol{W}}_L^{A,B}\right) \sigma \left((1-t)\boldsymbol{W}_{L-1}^A + t\boldsymbol{W}_{L-1}^B + 2t(1-t)\tilde{\boldsymbol{W}}_{L-1}^{A,B}\right)$$
$$\ldots \sigma \left((1-t)\boldsymbol{W}_1^A + t\boldsymbol{W}_1^B + 2t(1-t)\tilde{\boldsymbol{W}}_1^{A,B}\right) \boldsymbol{x}_0. \tag{18}$$

Then, we define the pre-activation and post-activation for each layer as follows:

$$\boldsymbol{x}_1^{A,B}(t) = \left((1-t)\boldsymbol{W}_1^A + t\boldsymbol{W}_1^B + 2t(1-t)\tilde{\boldsymbol{W}}_1^{A,B}\right) \boldsymbol{x}_0, \tag{19}$$

$$\boldsymbol{x}_l^{A,B}(t) = \left((1-t)\boldsymbol{W}_{l-1}^A + t\boldsymbol{W}_{l-1}^B + 2t(1-t)\tilde{\boldsymbol{W}}_{l-1}^{A,B}\right) \sigma \, \boldsymbol{x}_{l-1}^{A,B}(t). \tag{20}$$

Now, we consider the $L_2$ norm distance between $\boldsymbol{x}_1^{A,B}(t)$ and the endpoints defining the following distances:

$$d_1^{A,B}(t, 0) = \|\boldsymbol{x}_1^{A,B}(t) - \boldsymbol{x}_1^{A,B}(0)\|$$
$$= \|t(\boldsymbol{x}_1^{A,B}(1) - \boldsymbol{x}_1^{A,B}(0)) + 2t(1-t)\tilde{\boldsymbol{W}}_1^{A,B}\boldsymbol{x}_0\|, \tag{21}$$

$$d_1^{A,B}(t, 1) = \|\boldsymbol{x}_1^{A,B}(t) - \boldsymbol{x}_1^{A,B}(1)\|$$
$$= \|(1-t)(\boldsymbol{x}_1^{A,B}(1) - \boldsymbol{x}_1^{A,B}(0)) + 2t(1-t)\tilde{\boldsymbol{W}}_1^{A,B}\boldsymbol{x}_0\|, \tag{22}$$

Then, we can use Triangle Inequality and Cauchy-Schwarz Inequality to get an upper bound $U(d_1^{A,B}(t, 0))$ for $d_1^{A,B}(t, 0)$:

$$d_1^{A,B}(t, 0) \leq t\|\boldsymbol{x}_1^{A,B}(1) - \boldsymbol{x}_1^{A,B}(0)\| + 2t(1-t)\|\tilde{\boldsymbol{W}}_1^{A,B}\boldsymbol{x}_0\| \qquad \text{(Triangle Inequality)}$$
$$\leq t\|\boldsymbol{x}_1^{A,B}(1) - \boldsymbol{x}_1^{A,B}(0)\| + 2t(1-t)\|\tilde{\boldsymbol{W}}_1^{A,B}\| \cdot \|\boldsymbol{x}_0\| \qquad \text{(Cauchy-Schwarz Inequality)}$$
$$= tM_1(\boldsymbol{\theta}_A, \boldsymbol{\theta}_B; \boldsymbol{x}_0) + 2t(1-t)\|\tilde{\boldsymbol{W}}_1^{A,B}\| \cdot \|\boldsymbol{x}_0\|$$
$$= U(d_1^{A,B}(t, 0)). \tag{23}$$

Similarly, we can get

$$U(d_1^{A,B}(t, 1)) = (1-t)M_1(\boldsymbol{\theta}_A, \boldsymbol{\theta}_B; \boldsymbol{x}_0) + 2t(1-t)\|\tilde{\boldsymbol{W}}_1^{A,B}\| \cdot \|\boldsymbol{x}_0\|, \tag{24}$$

$$U(d_1^{A,C}(t, 0)) = tM_1(\boldsymbol{\theta}_A, \boldsymbol{\theta}_C; \boldsymbol{x}_0) + 2t(1-t)\|\tilde{\boldsymbol{W}}_1^{A,C}\| \cdot \|\boldsymbol{x}_0\|, \tag{25}$$

$$U(d_1^{A,C}(t, 1)) = (1-t)M_1(\boldsymbol{\theta}_A, \boldsymbol{\theta}_C; \boldsymbol{x}_0) + 2t(1-t)\|\tilde{\boldsymbol{W}}_1^{A,C}\| \cdot \|\boldsymbol{x}_0\|, \tag{26}$$

Since $\tilde{\boldsymbol{\theta}}_{A,B}$ and $\tilde{\boldsymbol{\theta}}_{A,C}$ are $L_2$-norm-consistent, and given that $\boldsymbol{\theta}_A$, $\boldsymbol{\theta}_B$ and $\boldsymbol{\theta}_C$ satisfy the condition (17), we can conclude that:

$$U(d_1^{A,B}(t, 0)) \leq U(d_1^{A,C}(t, 0)), \tag{27}$$

$$U(d_1^{A,B}(t,1)) \leq U(d_1^{A,C}(t,1)). \tag{28}$$

We also consider the $L_2$ norm distance $b_1^{A,B}(t,0)$ between the post-activation $\sigma x_1^{A,B}(t)$ and the endpoints. Assuming that $\sigma$ is Lipschitz continuous, we have:

$$b_1^{A,B}(t,0) = \|\sigma x_1^{A,B}(t) - \sigma x_1^{A,B}(0)\|$$

$$\leq L_\sigma \, U(d_1^{A,B}(t,0)), \tag{29}$$

where the $L_\sigma$ is the Lipschitz constant of $\sigma$. This property holds for other post-activations as well. Thus, both pre-activation distance $d_1^{A,B}(t,0)$ and post-activation distance $b_1^{A,B}(t,0)$ for the first layer of $\boldsymbol{\theta}_{A,B}(t)$ have tighter upper bounds than those for $\boldsymbol{\theta}_{A,C}(t)$.

Let $z_1^{A,B}(t) = \sigma x_l^{A,B}(t)$ be the post-activation for the $l^{\text{th}}$ layer of $\boldsymbol{\theta}_{A,B}(t)$. We can also derive the the upper bound for the pre-activation distance $d_l^{A,B}(t,0)$ for deep layer:

$$d_l^{A,B}(t,0) = \|x_l^{A,B}(t) - x_l^{A,B}(0)\|$$

$$= \|(1-t)W_l^A(z_{l-1}^{A,B}(t) - z_{l-1}^{A,B}(0)) + tW_l^B(z_{l-1}^{A,B}(t) - z_{l-1}^{A,B}(1))$$

$$+ t(W_l^B z_{l-1}^{A,B}(1) - W_l^A z_{l-1}^{A,B}(0)) + 2t(1-t)\tilde{W}_l^{A,B}(z_{l-1}^{A,B}(t) - z_{l-1}^{A,B}(0))$$

$$+ 2t(1-t)\tilde{W}_l^{A,B} z_{l-1}^{A,B}(0)\| \tag{30}$$

$$\leq (1-t)\|W_l^A\| b_{l-1}^{A,B}(t,0) + t\|W_l^B\| b_{l-1}^{A,B}(t,1) + t\, d_l^{A,B}(1,0)$$

$$+ 2t(1-t)\|\tilde{W}_l^{A,B}\| b_{l-1}^{A,B}(t,0) + 2t(1-t)\|\tilde{W}_l^{A,B}\| z_{l-1}^A$$

$$\leq (1-t)L_\sigma\|W_l^A\| U(d_{l-1}^{A,B}(t,0)) + tL_\sigma\|W_l^B\| U(d_{l-1}^{A,B}(t,1)) + tM_l(\boldsymbol{\theta}_A, \boldsymbol{\theta}_B; \boldsymbol{x}_0)$$

$$+ 2t(1-t)L_\sigma\|\tilde{W}_l^{A,B}\| U(d_{l-1}^{A,B}(t,0)) + 2t(1-t)\|\tilde{W}_l^{A,B}\| z_{l-1}^A$$

$$= U(d_l^{A,B}(t,0)) \tag{31}$$

To streamline the proof, we ignore the other upper bounds here. Similar to our demonstration for the first layer, we have:

$$U(d_l^{A,B}(t,0)) \leq U(d_l^{A,C}(t,0)), \tag{32}$$

$$U(d_l^{A,B}(t,1)) \leq U(d_l^{A,C}(t,1)). \tag{33}$$

Finally, since we assume that $\mathcal{L}$ is Lipschitz continuous, there exists a constant $L_\mathcal{L}$ such that:

$$\mathcal{L}(\boldsymbol{f}(\boldsymbol{x}_0, \boldsymbol{\theta}_{A,B}(t)) - \boldsymbol{y}) \leq L_\mathcal{L}\|\boldsymbol{f}(\boldsymbol{x}_0, \boldsymbol{\theta}_{A,B}(t)) - \boldsymbol{y}\|$$

$$= L_\mathcal{L}\|\boldsymbol{x}_L^{A,B}(t) - \boldsymbol{y}\|$$

$$\leq L_\mathcal{L} U(\|\boldsymbol{x}_L^{A,B}(t) - \boldsymbol{y}\|)$$

$$= U_{A,B}(t), \tag{34}$$

where $\boldsymbol{y}$ is the ground truth label (or feature). As $\boldsymbol{\theta}_A$, $\boldsymbol{\theta}_B$, and $\boldsymbol{\theta}_C$ are three optimal networks, we can find a constant $\epsilon$ and specify them as $\epsilon$ optimal networks such that $\|\boldsymbol{f}(\boldsymbol{x}_0, \boldsymbol{\theta}_\alpha(t)) - \boldsymbol{y}\|$, for $\alpha \in \{A, B, C\}$. Then, we have:

$$\|\boldsymbol{x}_L^{A,B}(t) - \boldsymbol{y}\| = \|(1-t)W_L^A(z_{L-1}^{A,B}(t) - z_{L-1}^{A,B}(0)) + tW_L^B(z_{L-1}^{A,B}(t) - z_{L-1}^{A,B}(1))$$

$$+ (1-t)W_L^A(z_{L-1}^{A,B}(0) - \boldsymbol{y}) + tW_L^B(z_{L-1}^{A,B}(1) - \boldsymbol{y})$$

$$+ 2t(1-t)\tilde{W}_L^{A,B} z_{L-1}^{A,B}(t)\| \tag{35}$$

$$\leq (1-t)L_\sigma\|W_L^A\| U(d_{L-1}^{A,B}(t,0)) + tL_\sigma\|W_L^B\| U(d_{L-1}^{A,B}(t,1)) + \epsilon$$

$$+ 2t(1-t)L_\sigma\|\tilde{W}_L^{A,B}\| U(d_{L-1}^{A,B}(t,0)) + 2t(1-t)\|\tilde{W}_L^{A,B}\| z_{L-1}^A$$

$$= U(\|\boldsymbol{x}_L^{A,B}(t) - \boldsymbol{y}\|). \tag{36}$$

Following the above scaling, we can also get $U_{A,C}(t)$ and derive $U_{A,B}(t) \leq U_{A,C}(t)$. Thus, for the upper bounds of loss value over the curves $\gamma_{\tilde{\theta}_{A,B}}(t)$ and $\gamma_{\tilde{\theta}_{A,C}}(t)$, we have $\ell(\tilde{\theta}_{A,B}) \leq U_{A,B}$, $\ell(\tilde{\theta}_{A,C}) \leq U_{A,C}$, where $U_{A,B} \leq U_{A,C}$, finishing the proof. $\qquad\square$

### E.2. Proof of Corollary 4.3

*Proof.* Since $S(\boldsymbol{P}')$ and $S(\hat{\boldsymbol{P}})$ are the sets consisting of permutation matrices, the Frobenius norms of the weights in each layer of $\theta_{\hat{B}}$ and $\theta_{B'}$ are identical to those of $\theta_B$. Thus, $\theta_{\hat{B}}$, $\theta_{B'}$, and $\theta_B$ are $L_2$-norm-consistent with each other, according to Definition 4.1.

As $S(\hat{\boldsymbol{P}})$ and $S(\boldsymbol{P}')$ are solutions to the optimization problems (6) and (8), respectively, we derive the following relations for all $l \in \{1, 2, \ldots, L\}$:

$$M_l(\theta_A, \theta_{\hat{B}}; D) \leq M_l(\theta_A, \theta_B; D) \leq M_l(\theta_A, \theta_{B'}; D).$$

By Theorem 4.2, we conclude that $U_{A,\hat{B}} \leq U_{A,B} \leq U_{A,B'}$, completing the proof.

$\qquad\square$

## F. Practical Algorithm

### F.1. Computing the Permutation for Each Model Layer

PERMUTELAYERS($\theta_A, \theta_B, D, \text{OPT}$) in Algorithm 1 returns a model by permuting the layers of $\theta_B$, aligning or un-aligning it with $\theta_A$. To compute the permutation of the $l^{\text{th}}$ layer of $\theta_A$ for alignment with $\theta_B$, we first obtain the corresponding activations $\boldsymbol{x}_{i,l}^A$ and $\boldsymbol{x}_{i,l}^B$ for each sample $x^{(i)} \in D$. We then employ the cost function $c_l = 1 - \text{corr}(\boldsymbol{v}, \boldsymbol{z})$ and compute the sum of the cross-correlation matrices $R_{i,l}$ of the normalized $\boldsymbol{x}_{i,l}^A$ and $\boldsymbol{x}_{i,l}^B$ as follows:

$$R_l = \sum_{i=1}^{|D|} R_{i,l} = \sum_{i=1}^{|D|} \frac{\boldsymbol{x}_{i,l}^A - \boldsymbol{\mu}_{\boldsymbol{x}_{i,l}^A}}{\boldsymbol{\Sigma}_{\boldsymbol{x}_{i,l}^A}} \frac{\boldsymbol{x}_{i,l}^B - \boldsymbol{\mu}_{\boldsymbol{x}_{i,l}^B}}{\boldsymbol{\Sigma}_{\boldsymbol{x}_{i,l}^B}}. \tag{37}$$

Finally, we use the Hungarian algorithm (Kuhn, 1955) to solve the maximization problem (or equivalently the minimization problem) according to OPT. It is noteworthy that, to stay consistent with previous work, we compute the cross-correlation matrix here rather than the cost function defined in Equation (6) or Equation (8). This minimization or maximization procedure is still equivalent to ordinary least squares constrained to the solution space $\Pi_{d_l}$ (Ainsworth et al., 2023; Tatro et al., 2020). We give the pseudocode of computing the permutation matrices for feedforward networks in Algorithm 2. For more complex model architectures, the same principles apply, but the implementation details may vary. We refer readers to Ainsworth et al. (2023) and Jordan et al. (2023) for more details on computing the permutation matrices for other architectures, such as ResNet-18 (He et al., 2016b) and VGG19-BN (Simonyan & Zisserman, 2014).

---

**Algorithm 2** PERMUTELAYERS (Compute Permutation Matrices for Layer Alignment/Un-alignment)

---

**Require:** model $\theta_A$, model $\theta_B$, dataset $D$, optimization OPT;

  Set assignment problem $\boldsymbol{O}$ with minimization/maximization objective according to OPT;

  **for** each layer $l$ in $\{1, 2, \ldots, L-1\}$ **do**

    **for** each sample $\boldsymbol{x}_{i,0}$ in dataset $D$ **do**

      Compute $\boldsymbol{x}_{i,l}^A \leftarrow \sigma \circ \boldsymbol{W}_l^A \circ \ldots \circ \sigma \circ \boldsymbol{W}_1^A \boldsymbol{x}_{i,0}$,    $\boldsymbol{x}_{i,l}^B \leftarrow \sigma \circ \boldsymbol{W}_l^B \circ \ldots \circ \sigma \circ \boldsymbol{W}_1^B \boldsymbol{x}_{i,0}$;

    **end for**

    Compute the correlation matrix $R_l$ using Equation (37);

    Compute $P_l$ by solving the $\boldsymbol{O}$ with $R_l$ using Hungarian algorithm;

    Update $\boldsymbol{W}_l^B \leftarrow P_l \boldsymbol{W}_l^B$,    $\boldsymbol{W}_{l+1}^B \leftarrow \boldsymbol{W}_{l+1}^B P_l^\top$;

  **end for**

**Ensure:** permuted model $\theta_B$;

---

### F.2. Training the Quadratic Bézier Curve

FITQUADCURVE($\theta_A, \theta_B, \mathcal{F}, D, e$) returns the quadratic Bézier curve connecting $\theta_A$ and $\theta_B$, trained using method $\mathcal{F}$ over $D$ for $e$ epochs. As mentioned in Appendix B, at each step, we randomly select a sample $\hat{t}$ from the uniform distribution

over the interval $[0, 1]$ and update the value of $\boldsymbol{\theta}_{A,B}$ based on the gradient of the loss computed using the training method $\mathcal{F}$. This is achieved by computing the loss of $\boldsymbol{\theta}_{\hat{t}}$ over the dataset $D$ using $\mathcal{F}$, and then calculating the gradient of $\boldsymbol{\theta}_{A,B}$ (*i.e.*, the weights of the curve $\boldsymbol{\gamma}_{\boldsymbol{\theta}_{A,B}}(t)$) via the chain rule. The pseudocode is provided in Algorithm 3.

---

**Algorithm 3** FITQUADCURVE (Train Quadratic Bézier Curve)

---

**Require:** model $\boldsymbol{\theta}_A$, model $\boldsymbol{\theta}_B$, training method $\mathcal{F}$, dataset $D$, curve training epoch $e$;
   Initialize  $\boldsymbol{\theta}_{A,B} \leftarrow \frac{1}{2}(\boldsymbol{\theta}_A + \boldsymbol{\theta}_B)$;
   Initialize parametric curve  $\boldsymbol{\gamma}_{\boldsymbol{\theta}_{A,B}}(t) \leftarrow (1-t)^2\boldsymbol{\theta}_A + 2t(1-t)\boldsymbol{\theta}_{A,B} + t^2\boldsymbol{\theta}_B$;
   **for** $i = 1$ **to** $e$ **do**
      Sample $\hat{t}$ from the distribution $U(0, 1)$;
      $\boldsymbol{\theta}_{\hat{t}} \leftarrow$ RETRIEVEPOINT$(\boldsymbol{\gamma}_{\boldsymbol{\theta}_{A,B}}, \hat{t})$;
      Update $\boldsymbol{\theta}_{A,B}$, the weights of the curve $\boldsymbol{\gamma}_{\boldsymbol{\theta}_{A,B}}(t)$, using the loss computed by $\mathcal{F}$ with respect to $\boldsymbol{\theta}_{\hat{t}}$ over the dataset $D$;
   **end for**
**Ensure:** quadratic Bézier curve $\boldsymbol{\gamma}_{\boldsymbol{\theta}_{A,B}}$;

---

# G. Evaluating TSC's Robustness Against Adaptive Attacks

## G.1. Apdative Attack Design

The key defense mechanism of TSC relies on increasing adversarial loss along the quadratic Bézier curve by projecting model $\boldsymbol{\theta}_{adv}$ to a distinct loss basin to find $\boldsymbol{\theta}_{adv'}$ during its first stage. An adaptive attack would aim to create a backdoored model maintaining low backdoor loss along this defensive curve.

Here, we assume the adversary has access to the model training procedure. Building upon the neural network subspace learning approach from (Wortsman et al., 2021), originally developed for accuracy and calibration improvements, we design an adaptive attack strategy. The key insight is to place $\boldsymbol{\theta}_{adv}$ and its symmetric point $\boldsymbol{\theta}_{adv'}$ in a subspace that minimizes loss on poisoned samples. Our approach simultaneously trains a curve and updates its endpoints $\boldsymbol{\theta}_{adv}$ and $\boldsymbol{\theta}_{adv'}$ using mixed benign and poisoned data (*i.e.*, learning a backdoored subspace). After training, we select one endpoint as the final model.

The implementation details are provided in Algorithm 4. We first project the backdoored model $\boldsymbol{\theta}_{adv}$ to a symmetric loss basin to obtain $\boldsymbol{\theta}_{adv'}$ by solving Equation (8), similar to TSC's initial process. We then train the curve connecting $\boldsymbol{\theta}_{adv}$ and $\boldsymbol{\theta}_{adv'}$ over $D_{adv}$ without fixed endpoints, ensuring the curve lies within the backdoored subspace. Finally, we return $\boldsymbol{\theta}_{adv}$ as the output model, which is expected to achieves lower loss along the curve found by TSC compared to the original model.

---

**Algorithm 4** Adaptive Attack against TSC

---

**Require:** backdoored model $\boldsymbol{\theta}_{adv}$, training method $\mathcal{F}$, poisoned dataset $D_{adv}$, curve training epoch $e$;
   ▷  Projecting backdoored model to symmetric subspace
   $\boldsymbol{\theta}_{adv'} \leftarrow$ PERMUTELAYERS$(\boldsymbol{\theta}_{adv}, \boldsymbol{\theta}_{adv}, D_{adv}, \text{MAX})$;
   Initialize  $\boldsymbol{\theta}_{m-adv} \leftarrow \frac{1}{2}(\boldsymbol{\theta}_{adv} + \boldsymbol{\theta}_{adv'})$;
   Initialize parametric curve  $\boldsymbol{\gamma}_{\boldsymbol{\theta}_{adv}}(t) \leftarrow (1-t)^2\boldsymbol{\theta}_{adv} + 2t(1-t)\boldsymbol{\theta}_{m-adv} + t^2\boldsymbol{\theta}_{adv'}$;
   ▷  Learning Symmetric Backdoored Subspace
   **for** $i = 1$ **to** $e$ **do**
      Sample $\hat{t}$ from the distribution $U(0, 1)$;
      $\boldsymbol{\theta}_{\hat{t},adv} \leftarrow$ RETRIEVEPOINT$(\boldsymbol{\gamma}_{\boldsymbol{\theta}_{adv}}, \hat{t})$;
      Update the weights of $\boldsymbol{\theta}_{adv}$, $\boldsymbol{\theta}_{m-adv}$ and $\boldsymbol{\theta}_{adv'}$ simultaneously, using the loss computed by $\mathcal{F}$ with respect to $\boldsymbol{\theta}_{\hat{t},adv}$ over the dataset $D_{adv}$;
   **end for**
**Ensure:** backdoored model $\boldsymbol{\theta}_{adv}$;

---

## G.2. Empirical Evaluation Against Adaptive Attack

To evaluate TSC's robustness against adaptive attacks, we apply Algorithm 4 to convert models backdoored by BadNet (Gu et al., 2017), Blended (Chen et al., 2017), SSBA (Li et al., 2021c), LF (Zeng et al., 2021), WaNet (Nguyen & Tran, 2021), Input-aware (Nguyen & Tran, 2020), LC (Turner et al., 2019), and SIG (Barni et al., 2019) under supervised learning. We

*Table 4.* Performance of TSC against adaptive attacks on **CIFAR-10** under **supervised learning**.

| CIFAR10 | No Defense | | TSC | |
|---|---|---|---|---|
| (Poisoning rate-5%) | ACC($\uparrow$) | ASR($\downarrow$) | ACC($\uparrow$) | ASR($\downarrow$) |
| Adap-BadNet | 92.43 | 91.26 | 91.47 | 2.07 |
| Adap-Blended | 93.12 | 99.23 | 90.88 | 7.26 |
| Adap-Inputaware | 92.11 | 90.18 | 90.21 | 4.85 |
| Adap-LC | 93.27 | 98.14 | 89.87 | 2.92 |
| Adap-LF | 92.46 | 97.91 | 88.66 | 3.39 |
| Adap-SIG | 92.33 | 94.99 | 90.34 | 1.82 |
| Adap-SSBA | 92.39 | 94.87 | 89.33 | 2.68 |
| Adap-WaNet | 92.61 | 87.29 | 90.19 | 1.59 |

*Table 5.* Performance of TSC against adaptive attacks on **Im-ageNet100** under **supervised learning**.

| ImageNet100 | No Defense | | TSC | |
|---|---|---|---|---|
| (Poisoning rate-1%) | ACC($\uparrow$) | ASR($\downarrow$) | ACC($\uparrow$) | ASR($\downarrow$) |
| Adap-BadNet | 83.64 | 99.47 | 78.22 | 0.23 |
| Adap-Blended | 84.42 | 98.15 | 78.85 | 9.61 |
| Adap-Inputaware | 79.42 | 70.44 | 77.04 | 0.82 |
| Adap-LC | 84.22 | 42.67 | 80.44 | 0.36 |
| Adap-LF | 83.61 | 99.66 | 78.18 | 3.26 |
| Adap-SIG | 83.39 | 69.86 | 79.55 | 5.61 |
| Adap-SSBA | 83.13 | 99.32 | 80.19 | 7.67 |
| Adap-WaNet | 81.75 | 88.04 | 80.31 | 0.26 |

*Table 6.* Performance of TSC against adaptive attacks using **SimCLR** with CIFAR-10 and ImageNet100 pretraining (**self-supervised learning**).

| Pre-training Dataset | Downstream Dataset | No Defense | | TSC (ours) | |
|---|---|---|---|---|---|
| | | ACC($\uparrow$) | ASR($\downarrow$) | ACC($\uparrow$) | ASR($\downarrow$) |
| CIFAR10 | STL10 | 76.51 | 99.80 | 72.62 | 4.83 |
| | GTSRB | 82.47 | 98.94 | 77.77 | 1.90 |
| | SVHN | 64.90 | 98.74 | 64.41 | 5.13 |
| ImageNet | STL10 | 94.94 | 98.87 | 88.24 | 3.42 |
| | GTSRB | 75.81 | 99.90 | 70.94 | 6.13 |
| | SVHN | 73.62 | 99.32 | 68.18 | 3.42 |

conduct experiments on CIFAR10 using PreAct-ResNet with 5% poisoning rate and ImageNet using ResNet50 with 1% poisoning rate.

For self-supervised learning, we adapt BadEncoder (Jia et al., 2022) into our adaptive attack framework. Using SimCLR, we utilize publicly available backdoored ResNet18 and ResNet50 encoders on CIFAR10 and ImageNet, respectively, evaluating ASR and ACC through linear probing on downstream datasets STL10, GTSRB, and SVHN.

Since our adaptive attack is designed to exlpore the robustness of TSC, we do not consider other defenses here. For fair evaluation, we use the default settings of TSC: global epoch $E_{TSC} = 3$, curve index $t = 0.4$, and curve training epoch $e = 200$ for supervised learning; $E_{TSC} = 2$, $t = 0.25$, and $e = 200$ for SimCLR. We provide defenders with 5% clean samples. Additional experimental settings follow Appendix L.

The defense results are presented in Tables 4 to 6. After retraining the backdoored models using Algorithm 4, we observe slight improvements in both ACC and ASR compared to the original models. We suspect that such improvements are due to the benefits of the subspace learning approach (Wortsman et al., 2021). Importantly, TSC maintains its robustness against these adaptive attacks across both supervised and self-supervised learning settings.

### G.3. Loss Landscape Visualization of Adaptive Attack

To further explore the effectiveness of our defense against adaptive attacks, we present a loss landscape visualization for Adap-BadNet before and after applying TSC, as shown in Figure 4. The left plot shows that the adaptive attack's curve training procedure successfully places its backdoored model, $\boldsymbol{\theta}_{adv}$, and its corresponding symmetric point, $\boldsymbol{\theta}_{adv'}$, in a subspace that minimizes loss on poisoned samples. However, as depicted in the middle plot, the curve identified by the first stage of TSC traverses a loss basin characterized by considerably higher loss on poisoned samples, rather than the basin exploited by the adaptive attack. We attribute this to TSC's training procedure, which exclusively uses benign samples, thereby guiding the curve to circumvent the loss basin optimized by the adaptive attack. Moreover, the permutation operation in the first stage of TSC could also find an endpoint in a different loss basin from both $\boldsymbol{\theta}_{adv}$ and $\boldsymbol{\theta}_{adv'}$, as there are multiple permutations matrices that satisfy the maximization objective in Equation (8) (Entezari et al., 2022).

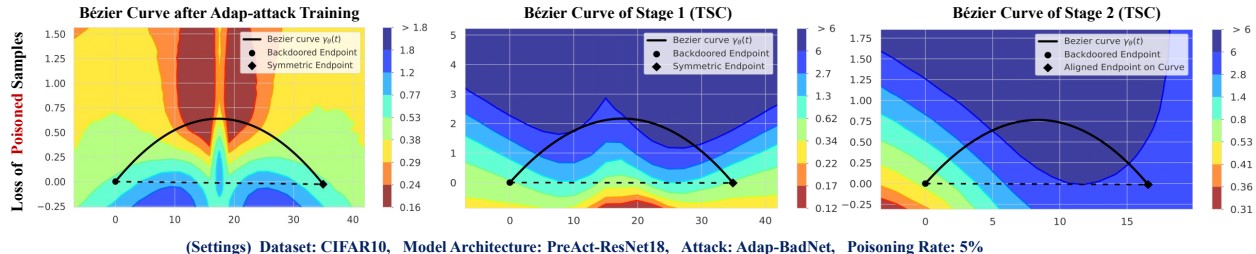

(Settings) Dataset: CIFAR10, Model Architecture: PreAct-ResNet18, Attack: Adap-BadNet, Poisoning Rate: 5%

*Figure 4.* Loss landscape for poisoned samples of Adap-Badnet, along with trained quadratic curves connecting distinct models. The backdoored model is a PreAct-ResNet18 trained on CIFAR-10, which contains 5% poisoned samples. **Left**: the curve identified by our adaptive attack. **Middle**: the curve identified by the first stage of TSC. **Right**: the curve identified by the second stage of TSC.

Overall, the combination of the permutation mechanism and training exclusively with benign samples contributes to amplifying the loss on poisoned samples.

## H. Ablation Studies

### H.1. Sensitivity Analysis on Global Epochs $E_{TSC}$ and the Curve Index $t$

TSC has two key hyperparameters: the number of global epochs $E_{TSC}$ and the curve index $t$.

For supervised learning, we investigate their impact through ablation studies on CIFAR10 using PreAct-ResNet18 with a 5% poisoning rate under supervised learning. Figure 5 illustrates the purification performance of TSC under varying $E_{TSC}$ and $t$. Each data point represents averaged results against twelve attacks: BadNet (Gu et al., 2017), Blended (Chen et al., 2017), SSBA (Li et al., 2021c), LF (Zeng et al., 2021), WaNet (Nguyen & Tran, 2021), Inputaware (Nguyen & Tran, 2020), LC (Turner et al., 2019), SIG (Barni et al., 2019), SBL-BadNet, SBL-Blended (Pham et al., 2024), Narcissus (Zeng et al., 2023) and SAPA (He et al., 2024).

For self-supervised learning, we conduct experiments on pre-training dataset CIFAR10 and downstream dataset STL10 using ResNet18 to evaluate the performance of TSC against BadEncoder (Jia et al., 2022). Figure 6 shows the results of TSC under different $E_{TSC}$ and $t$ values.

**Importantly**, each point in Figures 5 and 6 shows the final ACC and ASR values of TSC using different combinations of $E_{TSC}$ and $t$. This differs from Figure 3, Figure 7, and similar figures, which display ACC and ASR values evaluated along the curve at each round of $E_{TSC}$ with fixed $t$.

The results in Figure 5 demonstrate that increasing the curve index $t$ to 0.5 enhances purification performance at the cost of slightly reduced clean accuracy (ACC) per epoch. Similarly, as $t$ increases from 0.5 to 0.95, ACC improves but purification effectiveness decreases. As previously noted, when $t \geq 0.4$ (or $t \leq 0.6$), a single epoch of TSC proves insufficient for backdoor removal. Furthermore, increasing the number of global epochs $E_{TSC}$ could enhance the stability of the defense performance against various attacks. Our empirical analysis suggests $t = 0.4$ and $E_{TSC} = 3$ as optimal parameters for supervised learning settings.

We can observe that the performance of TSC on self-supervised learning is consistent with that of supervised learning. The key difference is that a smaller curve index $t$ suffices for backdoor removal in self-supervised learning. Though Figure 6 (left) demonstrates that setting $t = 0.2$ and $E_{TSC} = 1$ achieves effective purification, we opt for more conservative hyperparameters ($t = 0.25$ and $E_{TSC} = 2$) to ensure robust performance across diverse attack scenarios.

Additional results for ACC and ASR along the curve with fixed $t = 0.4$ are presented in Figures 7, 8, 11 and 12. While $E_{TSC} = 1$ or $E_{TSC} = 2$ can occasionally reduce ASR to near zero, $E_{TSC} = 3$ provides more consistent robustness across different attacks. For conservative defenders, we recommend larger values of $E_{TSC}$ and $t$ as the hyperparameters.

### H.2. Analysis on Model Architecture

To evaluate the stability of TSC across different architectures, we conduct experiments using VGG19-BN (Simonyan & Zisserman, 2014) and InceptionV3 (Szegedy et al., 2016) on the CIFAR10 dataset under supervised learning settings. We

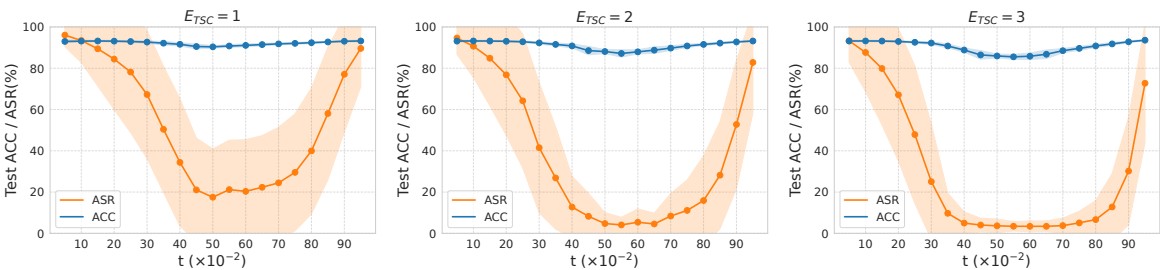

*Figure 5.* (**Supervised Learning**) Effect of global epoch $E_{TSC}$ and the curve index $t$ on TSC defense. We evaluate the performance of TSC on CIFAR10 with 5% poisoning rate using PreAct-ResNet18. Each point is averaged over the results of TSC against 12 attacks.

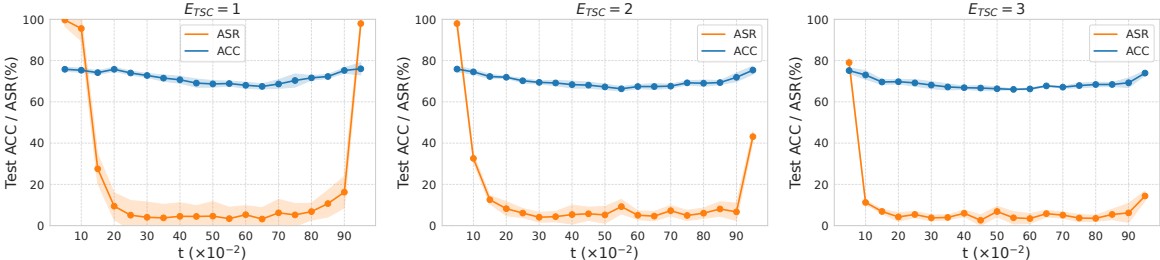

*Figure 6.* (**Self-Supervised Learning, SimCLR**) Effect of global epoch $E_{TSC}$ and the curve index $t$ on TSC defense. We evaluate the performance of TSC against BadEncoder on pre-training dataset CIFAR10 and downstream dataset STL10 with ResNet18. We performed 3 defense runs for each point and averaged the results in the figure.

*Table 7.* (**Supervised Learning**) Performance of MCR and TSC on CIFAR-10 with 5% poisoning rate using **VGG19-BN**.

| CIFAR10 | No Defense | | MCR | | TSC | |
|---|---|---|---|---|---|---|
| (Poisoning rate-5%) | ACC(↑) | ASR(↓) | ACC(↑) | ASR(↓) | ACC(↑) | ASR(↓) |
| BadNet | 91.19 | 93.92 | 90.44 | 55.21 | 90.69 | 1.76 |
| Blended | 92.24 | 99.43 | 91.29 | 96.56 | 90.46 | 9.09 |
| Inputaware | 89.22 | 93.34 | 90.96 | 5.44 | 90.58 | 6.32 |
| LC | 91.78 | 99.21 | 91.05 | 100.00 | 89.82 | 3.23 |
| LF | 89.27 | 96.29 | 90.22 | 1.02 | 88.97 | 1.38 |
| SIG | 91.91 | 97.23 | 91.09 | 99.87 | 89.25 | 8.22 |
| SSBA | 91.53 | 90.39 | 91.05 | 81.84 | 91.03 | 7.92 |
| WaNet | 87.42 | 94.32 | 92.04 | 2.45 | 90.36 | 1.91 |

*Table 8.* (**Supervised Learning**) Performance of MCR and TSC on CIFAR-10 with 5% poisoning rate using **Inception-v3**.

| CIFAR10 | No Defense | | MCR | | TSC | |
|---|---|---|---|---|---|---|
| (Poisoning rate-5%) | ACC(↑) | ASR(↓) | ACC(↑) | ASR(↓) | ACC(↑) | ASR(↓) |
| BadNet | 90.23 | 95.82 | 89.74 | 70.16 | 90.78 | 1.88 |
| Blended | 90.16 | 99.40 | 90.69 | 91.57 | 90.36 | 6.93 |
| Inputaware | 89.01 | 93.50 | 90.94 | 1.42 | 90.71 | 2.27 |
| LC | 92.19 | 99.31 | 90.06 | 96.57 | 90.76 | 3.09 |
| LF | 91.43 | 93.12 | 90.28 | 22.22 | 91.00 | 1.31 |
| SIG | 90.89 | 91.29 | 91.49 | 92.04 | 89.44 | 2.36 |
| SSBA | 90.53 | 89.24 | 90.86 | 62.65 | 90.81 | 3.85 |
| WaNet | 89.34 | 91.26 | 90.13 | 5.44 | 90.51 | 1.87 |

set the poisoning rate to 5% for all attacks and employ the default hyperparameters for MCR and TSC. The results are presented in Table 7 and Table 8 respectively.

The results demonstrate that TSC maintains robust performance across different model architectures. While TSC occasionally shows marginally lower ACC compared to MCR, it consistently demonstrates superior robustness against all considered attacks.

# I. Additional Results for Supervised Learning

In this section, we provide comprehensive results for supervised learning settings.

Figures 7 to 10 illustrates the performance of TSC and MCR on CIFAR10 with 10%, 5%, and 1% poisoning rates, respectively. Figures 11 and 12 present the results on ImageNet100 with 1% and 0.5% poisoning rates, respectively. Taking CIFAR10 as an example, we observe that backdoors are more effectively eliminated by TSC when the poisoning rate is relatively high. When the poisoning rate is 10%, a small number of global epochs $E_{TSC}$ is sufficient to remove backdoors implanted by attacks such as BadNet, InputAware, and LC. However, when the poisoning rate decreases to 1%, $E_{TSC} = 3$ is required to defend against all attacks. TSC demonstrates similar behavior on the ImageNet100 dataset. These findings further indicate that conservative hyperparameter settings are reasonable in supervised learning scenarios.

Tables 9 to 11 summarize the defense results of TSC and other baseline defenses on CIFAR10, ImageNet100, and GTSRB,

respectively. Tables 9 to 11 summarize the defense results of TSC and other baseline defenses on CIFAR10, ImageNet100, and GTSRB, respectively.

The experimental results reveal that while lower poisoning rates can increase the robustness of attack, excessively low poisoning rates sometimes result in diminished ASRs on the original model. For instance, with a 1% poisoning rate, WaNet attack achieves only 12.63% initial ASR on CIFAR10; with a 0.5% poisoning rate, LC attack yields merely 0.61% ASR on ImageNet100. Although these attacks are not considered successful backdoor attacks, we include them in our evaluation for completeness.

Notably, TSC successfully reduces the attack success rate of all attacks to below 15%. As mentioned in our main paper, ANP (Wu & Wang, 2021), FT-SAM (Zhu et al., 2023), and I-BAU (Zeng et al., 2022) perform well on smaller datasets like CIFAR10 and GTSRB, particularly with higher poisoning rates. However, their effectiveness is limited on the ImageNet100 dataset. While SAU (Wei et al., 2023) demonstrates good defense capabilities, it sometimes reduces clean accuracy (ACC) to suboptimal levels. We attribute this phenomenon to the lack of theoretical convergence guarantees for the unlearning loss function employed by SAU.

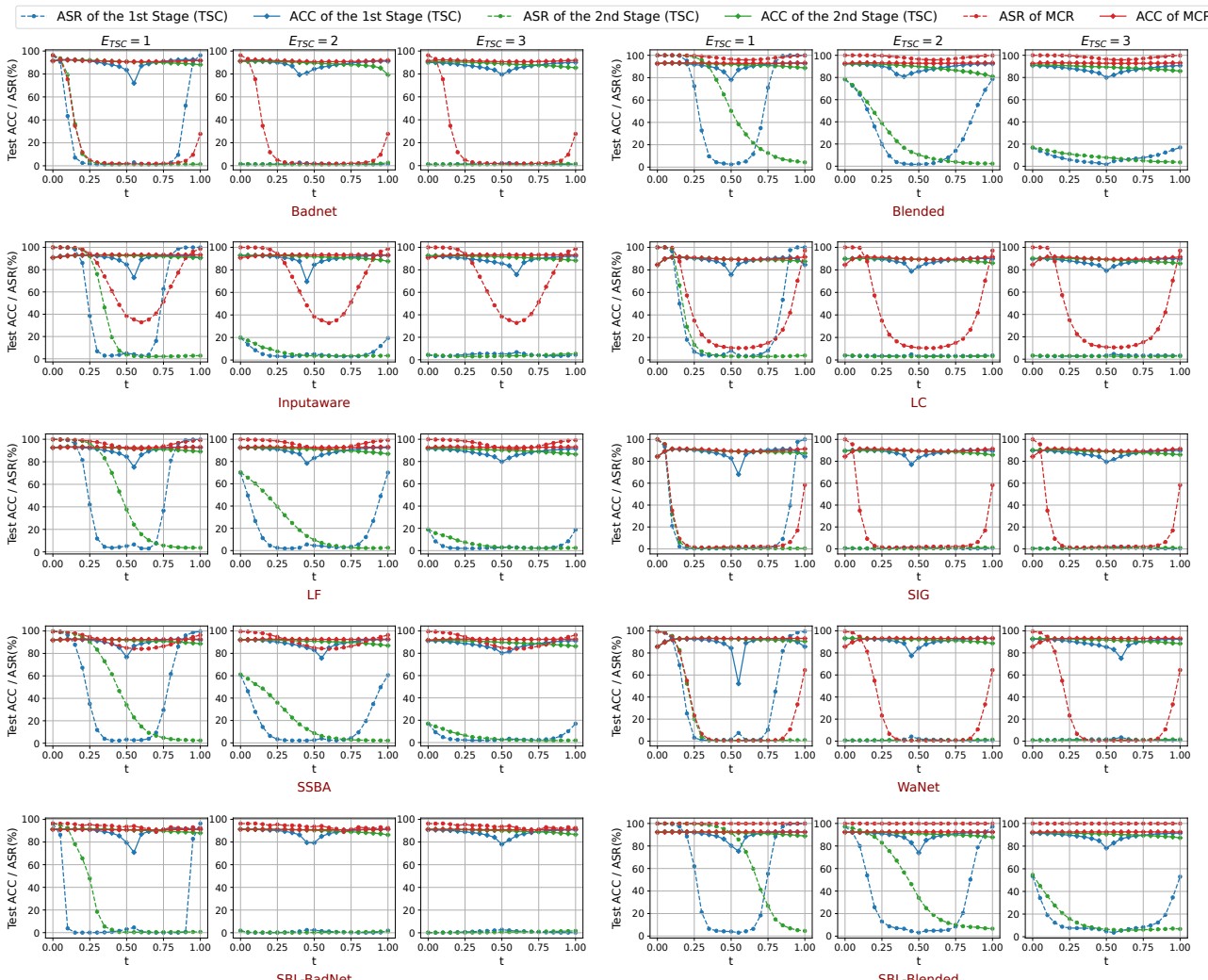

*Figure 7.* **(Supervised Learning)** Performance of TSC and MCR on CIFAR-10 with 10% poisoning rate using PreAct-ResNet18. Test accuracy (ACC) on benign samples and the attack success rate (ASR) are evaluated as functions of the points along the quadratic Bézier curve found by MCR and TSC. We select model points along the curve at $t = 0.4$ for each stage and round.

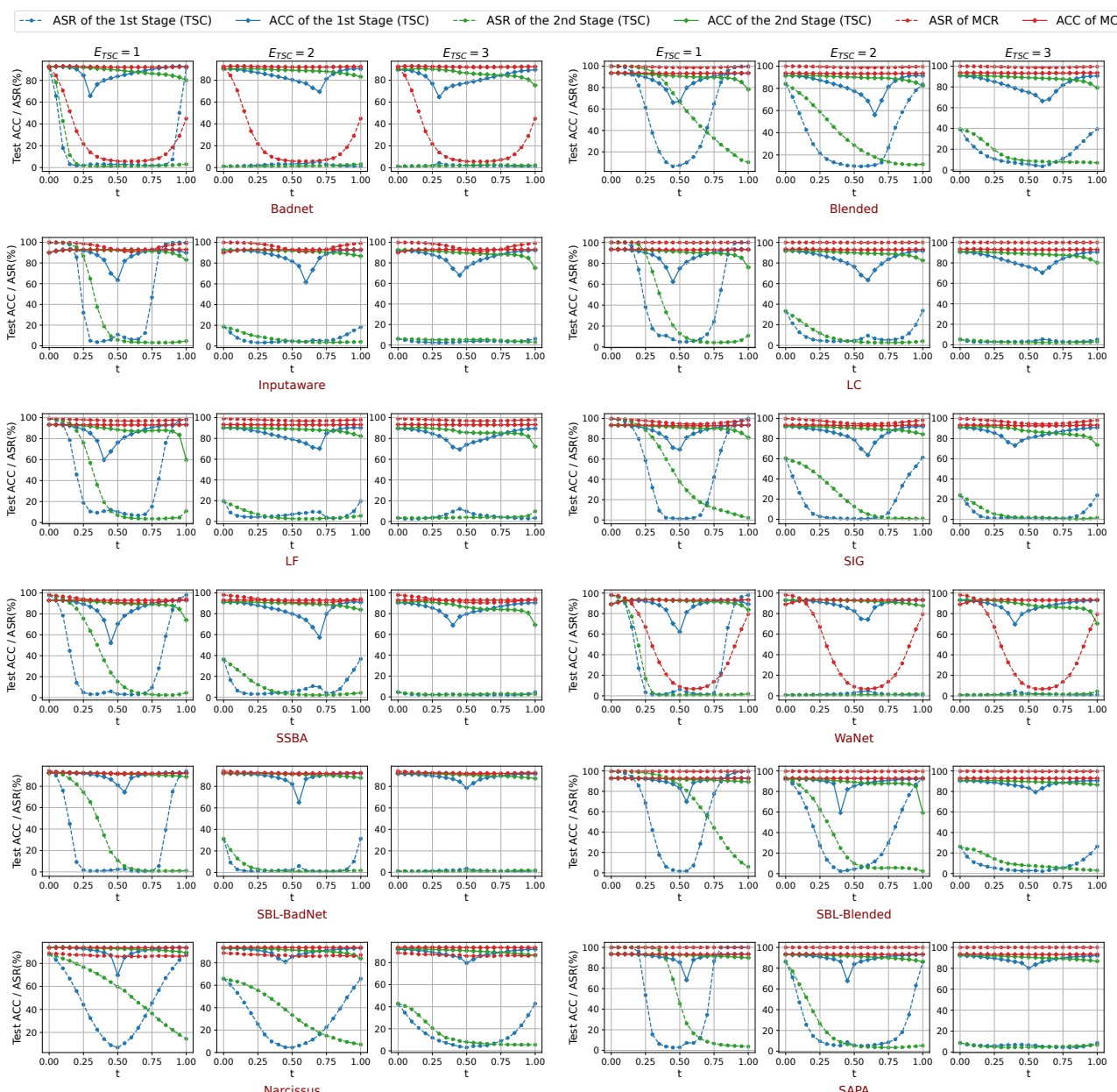

*Figure 8.* (**Supervised Learning**) Performance of TSC and MCR on CIFAR-10 with 5% poisoning rate using PreAct-ResNet18. We select model points along the curve at $t = 0.4$ for each stage and round.

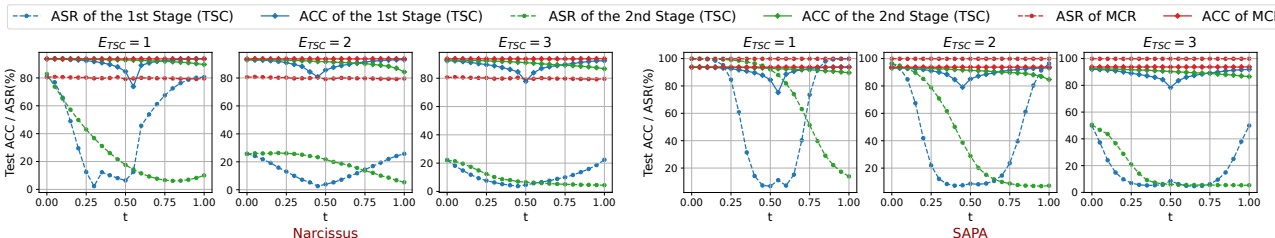

*Figure 9.* (**Supervised Learning**) Performance of TSC and MCR on CIFAR-10 with 1% poisoning rate using PreAct-ResNet18. We select model points along the curve at $t = 0.4$ for each stage and round.

*Figure 10.* (**Supervised Learning**) Performance of TSC and MCR on CIFAR-10 with 0.5% poisoning rate using PreAct-ResNet18 against Narcissus and SAPA attacks. We select model points along the curve at $t = 0.4$ for each stage and round.

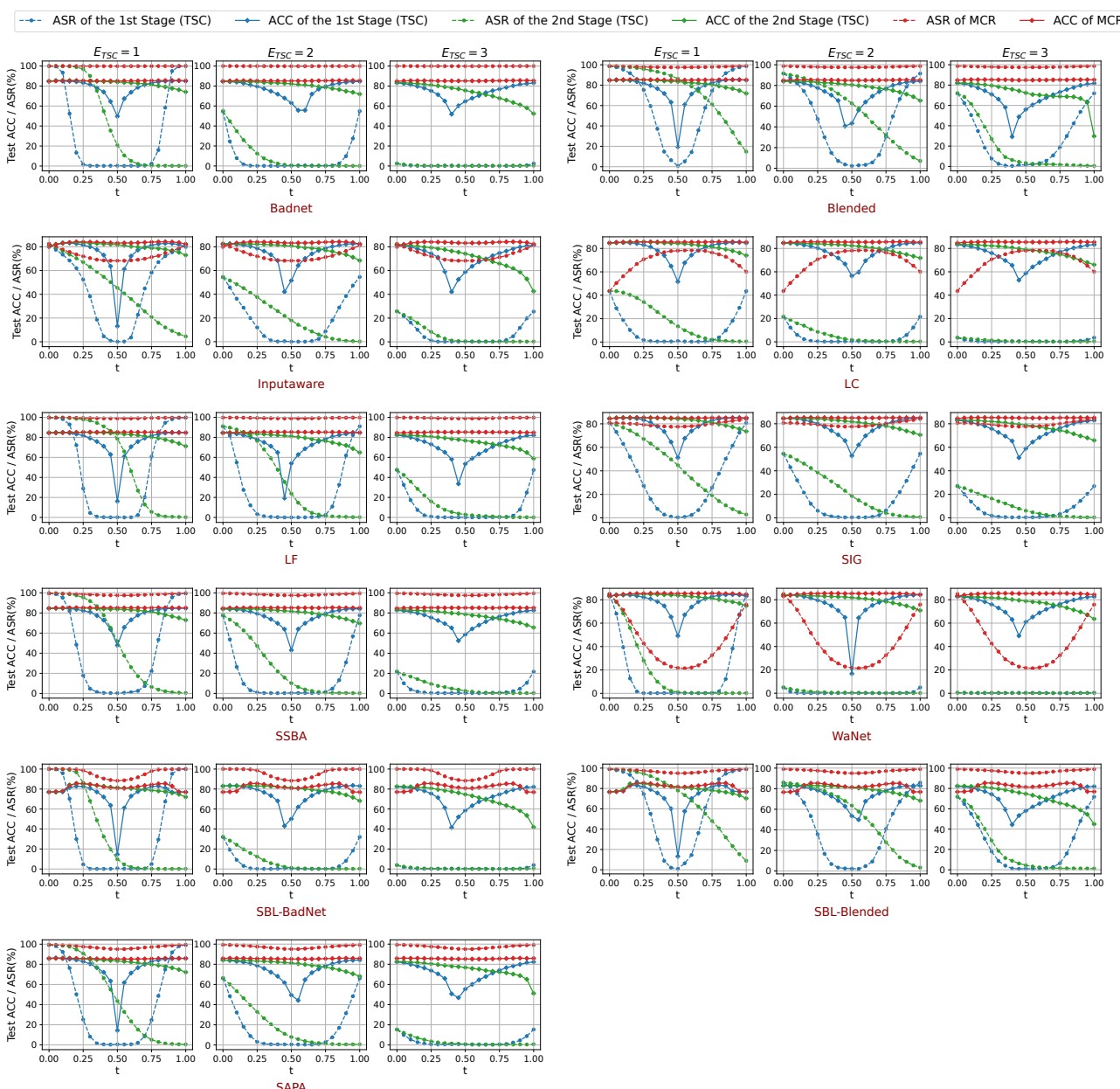

*Figure 11.* **(Supervised Learning)** Performance of TSC and MCR on ImageNet100 with 1% poisoning rate using ResNet50. We select model points along the curve at $t = 0.4$ for each stage and round.

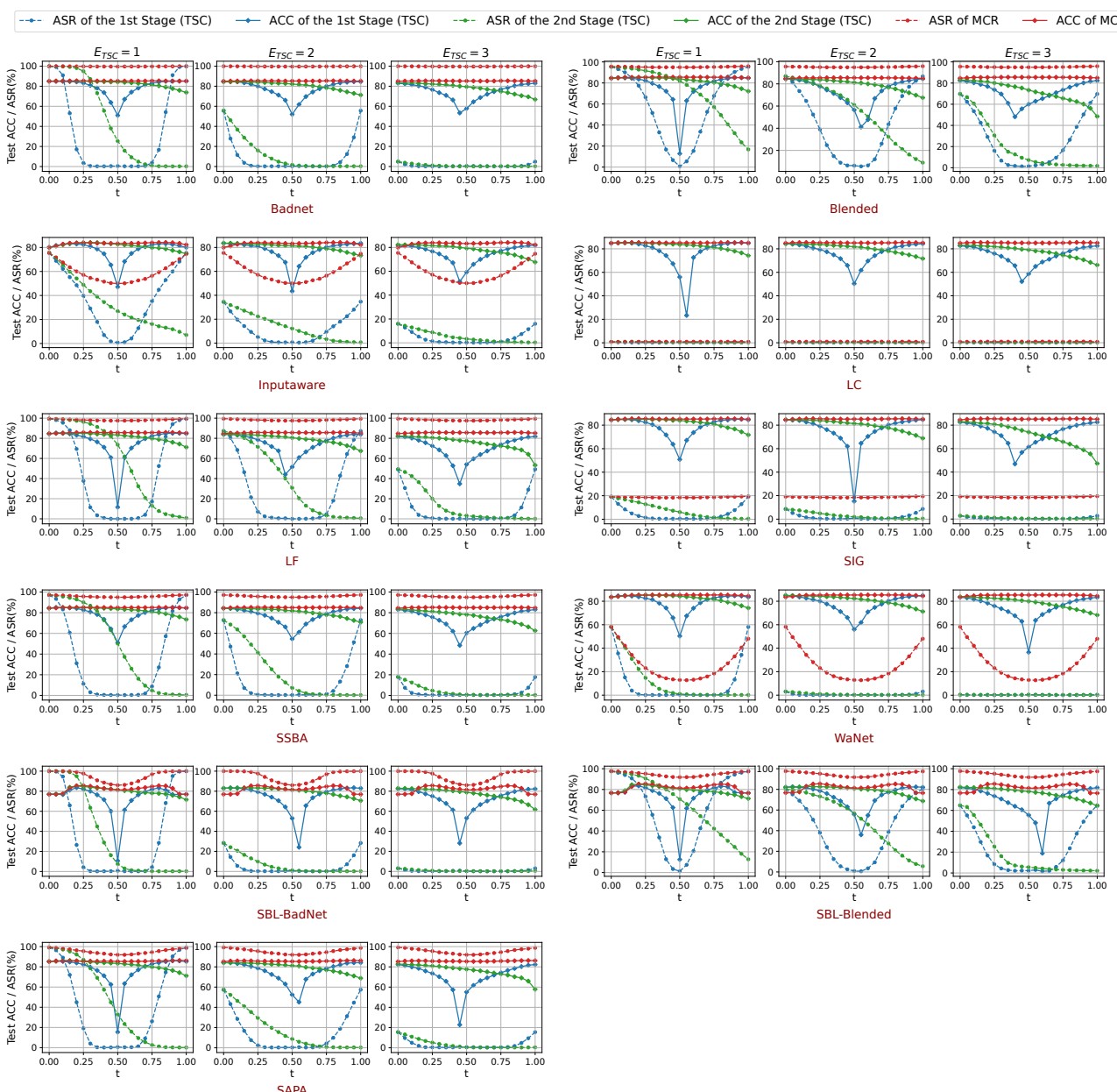

*Figure 12.* (**Supervised Learning**) Performance of TSC and MCR on ImageNet100 with 0.5% poisoning rate using ResNet50. We select model points along the curve at $t = 0.4$ for each stage and round.

*Table 9.* Results on CIFAR10 under **supervised learning** scenarios. Attack Success Rates (ASRs) below 15% are highlighted in blue to indicate a successful defense, while ASRs above 15% are denoted in red as failed defenses.

| | Attacks | Poison Rate | No Defense | | FP | | NC | | MCR | | ANP | | FT-SAM | | I-BAU | | SAU | | TSC (ours) | |
|---|---|---|---|---|---|---|---|---|---|---|---|---|---|---|---|---|---|---|---|---|
| | | | ACC(↑) | ASR(↓) | ACC(↑) | ASR(↓) | ACC(↑) | ASR(↓) | ACC(↑) | ASR(↓) | ACC(↑) | ASR(↓) | ACC(↑) | ASR(↓) | ACC(↑) | ASR(↓) | ACC(↑) | ASR(↓) | ACC(↑) | ASR(↓) |
| CIFAR10 | BadNet | 10% | 91.63 | 93.88 | 91.88 | 0.82 | 90.47 | 1.08 | 90.98 | 2.01 | 84.03 | 0.00 | 91.84 | 1.63 | 88.45 | 2.40 | 90.74 | 1.08 | 90.09 | 1.16 |
| | | 5% | 92.64 | 88.74 | 92.26 | 1.17 | 90.53 | 1.01 | 92.17 | 7.62 | 86.45 | 0.02 | 92.19 | 3.50 | 88.66 | 0.92 | 89.32 | 1.74 | 89.19 | 1.90 |
| | | 1% | 93.14 | 74.73 | 92.59 | 2.29 | 92.07 | 0.77 | 92.90 | 18.06 | 85.82 | 0.04 | 92.39 | 1.57 | 87.80 | 2.29 | 65.38 | 2.06 | 90.71 | 1.26 |
| | Blended | 10% | 93.46 | 99.78 | 91.97 | 18.42 | 90.75 | 2.44 | 92.93 | 97.87 | 84.91 | 6.14 | 92.48 | 11.38 | 88.86 | 9.52 | 90.39 | 9.57 | 90.33 | 8.79 |
| | | 5% | 93.66 | 99.61 | 92.70 | 49.47 | 93.67 | 99.61 | 93.23 | 99.01 | 88.95 | 18.76 | 93.00 | 29.59 | 88.07 | 34.86 | 90.69 | 7.74 | 90.14 | 10.53 |
| | | 1% | 93.76 | 94.88 | 92.92 | 69.74 | 93.76 | 94.88 | 93.62 | 93.10 | 89.69 | 60.52 | 93.00 | 49.36 | 89.62 | 25.74 | 90.02 | 36.16 | 91.12 | 12.46 |
| | Inputaware | 10% | 91.54 | 88.34 | 93.29 | 15.36 | 92.77 | 5.57 | 93.25 | 60.84 | 87.30 | 0.20 | 93.32 | 2.66 | 90.62 | 0.78 | 91.94 | 1.40 | 92.05 | 3.11 |
| | | 5% | 91.51 | 90.20 | 93.25 | 35.21 | 91.52 | 90.20 | 92.94 | 95.49 | 88.75 | 0.22 | 93.32 | 2.88 | 91.31 | 8.43 | 91.62 | 1.67 | 90.40 | 5.07 |
| | | 1% | 91.74 | 79.18 | 93.16 | 8.58 | 91.74 | 79.19 | 93.09 | 79.62 | 83.95 | 1.32 | 93.83 | 10.42 | 90.98 | 6.36 | 91.60 | 2.53 | 92.04 | 9.52 |
| | LF | 10% | 93.19 | 99.28 | 92.37 | 42.14 | 91.43 | 2.50 | 92.73 | 94.66 | 87.60 | 0.74 | 92.68 | 7.10 | 86.45 | 28.03 | 85.40 | 2.01 | 90.66 | 3.90 |
| | | 5% | 93.36 | 98.03 | 92.84 | 59.12 | 90.98 | 2.43 | 93.07 | 97.32 | 84.20 | 2.46 | 92.89 | 7.44 | 88.64 | 45.66 | 90.60 | 1.71 | 88.50 | 3.78 |
| | | 1% | 93.56 | 86.44 | 92.45 | 65.80 | 93.56 | 86.46 | 93.09 | 84.11 | 86.27 | 11.28 | 93.47 | 11.71 | 90.53 | 69.28 | 91.58 | 18.12 | 90.68 | 11.67 |
| | SSBA | 10% | 92.88 | 97.07 | 92.15 | 19.20 | 92.88 | 97.07 | 92.49 | 88.49 | 84.86 | 0.03 | 92.18 | 4.07 | 88.75 | 3.84 | 90.54 | 1.80 | 90.54 | 3.67 |
| | | 5% | 93.27 | 94.91 | 92.55 | 16.27 | 93.27 | 94.91 | 92.94 | 92.06 | 88.72 | 0.13 | 92.71 | 2.87 | 89.65 | 1.54 | 91.30 | 2.06 | 89.43 | 2.18 |
| | | 1% | 93.43 | 73.44 | 93.01 | 7.68 | 91.60 | 0.46 | 93.33 | 65.88 | 85.33 | 0.31 | 93.02 | 1.49 | 89.56 | 4.87 | 91.38 | 0.99 | 91.18 | 2.18 |
| | WaNet | 10% | 90.56 | 96.92 | 93.18 | 0.81 | 90.56 | 96.92 | 93.10 | 0.71 | 89.11 | 0.42 | 93.73 | 0.74 | 91.94 | 13.44 | 91.73 | 0.80 | 92.00 | 0.98 |
| | | 5% | 91.76 | 85.50 | 93.66 | 7.51 | 91.76 | 85.50 | 93.25 | 20.83 | 87.64 | 0.72 | 93.85 | 1.00 | 90.66 | 4.43 | 91.70 | 1.98 | 90.46 | 1.34 |
| | | 1% | 90.65 | 12.63 | 93.47 | 0.51 | 92.55 | 0.64 | 93.48 | 0.77 | 83.25 | 0.12 | 93.96 | 0.72 | 91.87 | 1.32 | 91.90 | 1.20 | 91.40 | 0.87 |
| | SBL-BadNet | 10% | 91.30 | 95.11 | 92.00 | 91.07 | 91.63 | 0.34 | 91.37 | 70.99 | 90.72 | 0.00 | 91.41 | 89.13 | 88.99 | 17.09 | 90.63 | 1.52 | 90.36 | 0.21 |
| | | 5% | 90.79 | 93.48 | 92.59 | 1.13 | 92.22 | 0.59 | 92.26 | 91.82 | 82.82 | 51.63 | 92.16 | 60.03 | 90.67 | 27.06 | 91.31 | 0.60 | 91.02 | 1.12 |
| | | 1% | 91.71 | 88.64 | 93.10 | 31.77 | 91.82 | 0.72 | 93.23 | 86.11 | 82.71 | 81.48 | 92.77 | 59.58 | 90.63 | 2.00 | 92.32 | 1.01 | 91.54 | 1.93 |
| | SBL-Blended | 10% | 90.46 | 94.12 | 92.49 | 29.61 | 90.46 | 88.12 | 92.51 | 99.91 | 86.32 | 52.96 | 92.40 | 74.02 | 91.44 | 22.79 | 88.11 | 9.09 | 90.98 | 8.27 |
| | | 5% | 91.70 | 97.67 | 92.97 | 79.74 | 91.70 | 97.67 | 92.75 | 99.61 | 85.13 | 20.48 | 92.50 | 77.90 | 89.65 | 57.41 | 91.43 | 11.53 | 90.47 | 8.94 |
| | | 1% | 92.07 | 91.84 | 93.43 | 83.80 | 92.07 | 91.84 | 93.37 | 95.02 | 85.30 | 58.19 | 93.31 | 82.64 | 90.67 | 64.08 | 92.34 | 16.31 | 90.11 | 6.70 |
| | SAPA | 5% | 93.57 | 100.00 | 92.56 | 41.88 | 92.76 | 2.51 | 93.25 | 100.00 | 84.83 | 1.14 | 92.80 | 8.40 | 88.51 | 1.44 | 91.39 | 3.30 | 91.13 | 4.51 |
| | | 1% | 94.01 | 99.97 | 92.34 | 92.22 | 92.80 | 2.14 | 93.83 | 100.00 | 86.06 | 92.68 | 93.06 | 79.80 | 86.69 | 15.17 | 91.83 | 1.96 | 90.37 | 7.41 |
| | | 0.5% | 93.77 | 84.80 | 88.82 | 82.76 | 92.74 | 1.52 | 93.78 | 80.83 | 87.99 | 81.52 | 93.23 | 82.02 | 90.16 | 26.48 | 91.75 | 0.68 | 90.98 | 7.32 |
| | LC | 10% | 84.49 | 99.68 | 89.90 | 2.82 | 90.17 | 2.02 | 89.95 | 12.82 | 84.08 | 90.01 | 91.32 | 3.09 | 86.28 | 3.80 | 88.78 | 0.36 | 89.95 | 2.74 |
| | | 5% | 93.31 | 98.33 | 92.19 | 72.99 | 92.32 | 0.64 | 92.94 | 99.94 | 88.15 | 13.83 | 92.59 | 57.18 | 90.15 | 1.99 | 91.53 | 1.50 | 90.04 | 2.38 |
| | | 1% | 93.79 | 75.93 | 92.86 | 29.86 | 92.31 | 0.68 | 93.67 | 82.54 | 86.58 | 31.46 | 92.83 | 39.40 | 89.78 | 0.71 | 92.16 | 3.77 | 90.08 | 5.78 |
| | SIG | 10% | 84.48 | 97.43 | 89.95 | 0.69 | 84.48 | 97.43 | 90.06 | 1.33 | 80.94 | 0.01 | 91.61 | 0.24 | 86.39 | 3.61 | 90.43 | 2.87 | 89.67 | 0.64 |
| | | 5% | 93.29 | 95.06 | 92.81 | 43.02 | 93.28 | 95.06 | 92.96 | 95.82 | 86.51 | 4.76 | 92.62 | 0.94 | 87.98 | 7.54 | 90.63 | 0.40 | 90.39 | 2.19 |
| | | 1% | 93.82 | 83.40 | 93.00 | 70.51 | 93.82 | 83.40 | 93.56 | 86.31 | 91.49 | 52.69 | 93.20 | 37.74 | 88.18 | 38.01 | 91.07 | 33.71 | 91.67 | 10.97 |
| | Narcissus | 5% | 93.72 | 90.91 | 91.93 | 68.61 | 93.72 | 80.91 | 93.63 | 86.64 | 87.87 | 49.27 | 93.19 | 27.92 | 87.82 | 73.79 | 90.72 | 1.57 | 91.35 | 14.48 |
| | | 1% | 93.68 | 82.87 | 92.29 | 44.88 | 93.68 | 47.87 | 93.61 | 49.79 | 92.01 | 27.01 | 93.05 | 26.80 | 90.21 | 18.67 | 91.36 | 3.24 | 90.65 | 7.88 |
| | | 0.5% | 93.68 | 80.58 | 92.94 | 29.59 | 93.67 | 32.57 | 93.69 | 32.96 | 89.35 | 16.78 | 93.06 | 14.08 | 89.16 | 21.09 | 91.74 | 5.81 | 91.71 | 8.02 |

*Table 10.* Results on ImageNet100 under **supervised learning** scenarios.

| | Attacks | Poison Rate | No Defense | | FP | | NC | | MCR | | ANP | | FT-SAM | | I-BAU | | SAU | | TSC (ours) | |
|---|---|---|---|---|---|---|---|---|---|---|---|---|---|---|---|---|---|---|---|---|
| | | | ACC(↑) | ASR(↓) | ACC(↑) | ASR(↓) | ACC(↑) | ASR(↓) | ACC(↑) | ASR(↓) | ACC(↑) | ASR(↓) | ACC(↑) | ASR(↓) | ACC(↑) | ASR(↓) | ACC(↑) | ASR(↓) | ACC(↑) | ASR(↓) |
| ImageNet100 | BadNet | 0.5% | 84.30 | 99.78 | 83.36 | 9.80 | 81.92 | 0.52 | 85.24 | 99.66 | 78.44 | 94.18 | 83.70 | 9.45 | 73.70 | 8.34 | 73.86 | 0.28 | 80.20 | 0.22 |
| | | 1% | 84.56 | 99.86 | 83.10 | 9.58 | 81.92 | 0.49 | 85.08 | 99.86 | 79.48 | 93.64 | 83.88 | 24.14 | 71.46 | 43.66 | 72.84 | 0.26 | 78.06 | 0.14 |
| | Blended | 0.5% | 84.44 | 94.32 | 82.80 | 63.25 | 84.44 | 94.32 | 85.58 | 94.97 | 84.56 | 93.27 | 83.40 | 75.43 | 74.22 | 62.34 | 73.84 | 3.72 | 76.58 | 12.63 |
| | | 1% | 84.90 | 98.04 | 83.36 | 69.21 | 80.21 | 70.21 | 85.04 | 97.58 | 84.54 | 97.70 | 83.86 | 82.00 | 73.10 | 61.25 | 69.24 | 0.53 | 75.88 | 6.35 |
| | Inputaware | 0.5% | 76.62 | 69.94 | 83.56 | 16.57 | 76.62 | 69.94 | 83.56 | 51.52 | 73.82 | 42.20 | 82.96 | 39.23 | 71.80 | 43.11 | 76.18 | 1.27 | 80.10 | 4.63 |
| | | 1% | 77.66 | 65.13 | 83.54 | 25.23 | 72.66 | 43.62 | 83.54 | 68.55 | 70.56 | 56.24 | 82.60 | 54.46 | 73.64 | 56.53 | 70.86 | 24.63 | 76.46 | 1.09 |
| | LF | 0.5% | 84.24 | 98.87 | 83.10 | 50.26 | 84.24 | 98.87 | 85.70 | 97.70 | 81.32 | 86.20 | 83.80 | 70.48 | 74.36 | 74.97 | 75.22 | 0.18 | 78.78 | 5.39 |
| | | 1% | 83.92 | 99.56 | 83.00 | 35.82 | 76.76 | 49.87 | 85.30 | 99.03 | 81.10 | 88.53 | 83.40 | 70.69 | 71.06 | 22.32 | 67.38 | 2.93 | 78.58 | 5.41 |
| | SSBA | 0.5% | 84.30 | 95.31 | 83.34 | 46.75 | 84.30 | 95.31 | 85.04 | 95.13 | 76.96 | 6.18 | 83.16 | 15.70 | 71.52 | 1.19 | 76.12 | 0.89 | 79.56 | 1.45 |
| | | 1% | 84.02 | 99.43 | 83.34 | 59.68 | 78.47 | 70.78 | 85.14 | 97.72 | 80.22 | 22.77 | 83.50 | 20.30 | 72.58 | 7.45 | 73.94 | 0.36 | 79.88 | 4.91 |
| | WaNet | 0.5% | 83.34 | 53.92 | 84.04 | 0.95 | 77.34 | 22.66 | 85.30 | 14.34 | 79.66 | 0.00 | 83.90 | 0.12 | 72.46 | 0.00 | 76.18 | 0.18 | 80.70 | 0.28 |
| | | 1% | 79.62 | 90.69 | 84.90 | 4.87 | 78.34 | 14.12 | 85.44 | 25.52 | 78.56 | 0.04 | 83.28 | 0.26 | 71.24 | 1.56 | 76.42 | 1.23 | 80.20 | 0.32 |
| | SBL-BadNet | 0.5% | 72.50 | 100.00 | 85.10 | 67.07 | 70.15 | 3.62 | 82.76 | 88.61 | 65.40 | 89.49 | 83.98 | 40.71 | 72.34 | 3.52 | 78.54 | 2.57 | 80.32 | 0.34 |
| | | 1% | 72.64 | 100.00 | 85.36 | 79.82 | 69.97 | 1.95 | 82.24 | 90.44 | 67.36 | 89.49 | 83.68 | 64.97 | 74.02 | 19.33 | 78.14 | 0.36 | 77.48 | 0.22 |
| | SBL-Blended | 0.5% | 72.52 | 97.56 | 85.10 | 89.29 | 72.52 | 97.56 | 82.54 | 92.63 | 68.28 | 97.05 | 83.84 | 89.07 | 70.78 | 37.35 | 73.66 | 14.87 | 79.42 | 7.18 |
| | | 1% | 72.68 | 99.17 | 83.41 | 68.24 | 71.42 | 70.14 | 82.82 | 95.78 | 72.72 | 99.15 | 83.92 | 92.69 | 73.52 | 20.30 | 76.78 | 39.29 | 77.16 | 8.87 |
| | SAPA | 0.5% | 85.04 | 98.83 | 83.44 | 20.53 | 78.57 | 9.20 | 85.60 | 93.05 | 80.42 | 96.59 | 83.82 | 30.04 | 69.32 | 18.34 | 73.34 | 1.07 | 79.00 | 1.74 |
| | | 1% | 85.50 | 98.79 | 83.34 | 27.86 | 77.42 | 3.52 | 85.42 | 95.80 | 83.12 | 93.76 | 83.96 | 45.88 | 69.12 | 41.64 | 75.42 | 1.23 | 78.14 | 1.41 |
| | LC | 0.5% | 84.22 | 0.61 | 83.34 | 0.20 | 84.22 | 0.61 | 85.04 | 0.93 | 84.48 | 0.57 | 83.86 | 0.24 | 69.72 | 0.36 | 76.20 | 0.16 | 80.36 | 0.22 |
| | | 1% | 84.10 | 32.97 | 83.48 | 4.28 | 81.13 | 0.42 | 85.48 | 76.75 | 84.22 | 32.42 | 83.58 | 8.06 | 73.52 | 1.76 | 70.50 | 0.79 | 80.18 | 0.57 |
| | SIG | 0.5% | 84.20 | 16.22 | 83.26 | 2.22 | 78.48 | 0.43 | 85.18 | 18.34 | 84.00 | 15.86 | 83.88 | 4.51 | 70.76 | 3.43 | 75.22 | 0.12 | 77.20 | 0.55 |
| | | 1% | 84.16 | 70.08 | 83.40 | 20.48 | 79.58 | 0.89 | 85.02 | 77.84 | 80.40 | 65.68 | 83.36 | 44.81 | 70.98 | 19.98 | 73.76 | 0.30 | 80.22 | 9.98 |

*Table 11.* Results on GTSRB under **supervised learning** scenarios.

| | Attacks | Poison Rate | No Defense | | FP | | NC | | MCR | | ANP | | FT-SAM | | I-BAU | | SAU | | TSC (ours) | |
|---|---|---|---|---|---|---|---|---|---|---|---|---|---|---|---|---|---|---|---|---|
| | | | ACC(↑) | ASR(↓) | ACC(↑) | ASR(↓) | ACC(↑) | ASR(↓) | ACC(↑) | ASR(↓) | ACC(↑) | ASR(↓) | ACC(↑) | ASR(↓) | ACC(↑) | ASR(↓) | ACC(↑) | ASR(↓) | ACC(↑) | ASR(↓) |
| GTSRB | BadNet | 10% | 97.62 | 95.48 | 98.21 | 0.09 | 97.48 | 0.01 | 98.68 | 4.18 | 95.86 | 0.00 | 98.82 | 0.31 | 96.47 | 0.02 | 97.75 | 0.02 | 98.05 | 0.01 |
| | | 5% | 97.89 | 93.00 | 97.86 | 2.11 | 97.09 | 0.00 | 97.99 | 4.02 | 92.72 | 0.00 | 98.96 | 0.09 | 14.43 | 0.00 | 94.65 | 0.00 | 98.15 | 0.00 |
| | | 1% | 98.39 | 79.23 | 98.48 | 0.01 | 97.20 | 0.00 | 98.32 | 3.33 | 93.99 | 0.00 | 98.73 | 0.00 | 93.80 | 0.02 | 10.30 | 0.00 | 98.20 | 0.01 |
| | Blended | 10% | 98.62 | 100.00 | 98.38 | 100.00 | 97.76 | 8.03 | 98.62 | 100.00 | 95.85 | 42.80 | 98.38 | 49.82 | 92.35 | 86.35 | 96.15 | 6.51 | 97.67 | 11.95 |
| | | 5% | 98.86 | 99.96 | 98.75 | 99.93 | 96.64 | 3.14 | 98.79 | 99.87 | 93.92 | 68.09 | 98.48 | 25.18 | 84.63 | 0.00 | 92.80 | 4.53 | 97.18 | 12.59 |
| | | 1% | 98.80 | 96.95 | 98.71 | 95.87 | 97.00 | 10.88 | 98.75 | 96.00 | 92.78 | 79.88 | 98.57 | 38.01 | 93.98 | 27.85 | 95.32 | 2.55 | 97.11 | 12.98 |
| | Inputaware | 10% | 98.76 | 95.93 | 98.91 | 4.46 | 98.76 | 95.92 | 99.00 | 13.88 | 98.19 | 0.00 | 99.45 | 0.07 | 97.09 | 0.52 | 98.37 | 0.08 | 99.11 | 0.02 |
| | | 5% | 98.26 | 92.84 | 98.93 | 4.07 | 98.26 | 92.84 | 99.21 | 17.58 | 98.82 | 0.00 | 99.53 | 0.01 | 98.61 | 0.07 | 98.64 | 0.00 | 99.37 | 0.01 |
| | | 1% | 98.75 | 7.05 | 99.45 | 0.03 | 99.12 | 0.61 | 99.15 | 11.13 | 98.44 | 0.03 | 99.62 | 0.14 | 97.87 | 0.00 | 98.43 | 0.37 | 99.40 | 0.02 |
| | LF | 10% | 97.89 | 99.36 | 98.28 | 82.28 | 97.23 | 0.27 | 97.93 | 98.81 | 96.07 | 0.00 | 98.03 | 3.93 | 95.78 | 16.15 | 96.45 | 0.76 | 98.33 | 3.91 |
| | | 5% | 98.16 | 98.81 | 97.89 | 98.17 | 97.55 | 0.02 | 97.95 | 98.90 | 89.10 | 2.16 | 98.47 | 0.99 | 87.54 | 1.40 | 95.88 | 0.01 | 98.31 | 2.05 |
| | | 1% | 98.17 | 96.11 | 97.63 | 71.56 | 97.55 | 0.49 | 98.11 | 95.86 | 88.60 | 15.49 | 98.34 | 1.28 | 96.70 | 68.50 | 94.74 | 0.01 | 97.52 | 2.62 |
| | SSBA | 10% | 97.90 | 99.47 | 97.75 | 99.46 | 97.72 | 0.29 | 97.95 | 99.33 | 88.47 | 0.00 | 98.32 | 34.71 | 96.14 | 1.88 | 96.55 | 0.32 | 97.63 | 6.23 |
| | | 5% | 98.01 | 99.22 | 98.12 | 99.12 | 97.03 | 0.33 | 97.97 | 98.98 | 90.29 | 6.09 | 97.66 | 4.93 | 6.12 | 0.00 | 6.32 | 0.00 | 98.03 | 3.12 |
| | | 1% | 98.84 | 94.51 | 98.78 | 91.38 | 97.51 | 0.12 | 98.78 | 91.65 | 89.87 | 0.02 | 98.40 | 6.55 | 86.37 | 0.96 | 93.94 | 1.96 | 97.48 | 0.64 |
| | WaNet | 10% | 97.74 | 94.25 | 97.62 | 88.07 | 98.25 | 0.00 | 98.52 | 1.94 | 97.08 | 0.00 | 0.00 | 0.00 | 96.72 | 0.00 | 98.91 | 0.04 | 98.71 | 0.00 |
| | | 5% | 97.42 | 92.85 | 98.00 | 12.24 | 97.42 | 92.85 | 98.67 | 5.80 | 98.67 | 0.00 | 98.97 | 0.00 | 97.93 | 0.25 | 98.19 | 0.05 | 98.86 | 0.00 |
| | | 1% | 97.08 | 62.24 | 98.08 | 1.14 | 98.88 | 9.67 | 98.51 | 53.94 | 97.66 | 0.00 | 99.04 | 0.35 | 96.27 | 0.00 | 97.73 | 0.06 | 98.50 | 0.00 |
| | SBL-BadNet | 10% | 88.84 | 95.47 | 97.50 | 0.13 | 96.44 | 0.04 | 96.95 | 0.00 | 98.52 | 0.00 | 92.06 | 0.00 | 97.13 | 0.00 | 97.13 | 0.00 | 97.56 | 0.03 |
| | | 5% | 89.21 | 95.10 | 97.78 | 0.18 | 97.27 | 0.00 | 97.67 | 1.36 | 98.42 | 51.52 | 92.84 | 0.00 | 93.73 | 0.00 | 93.73 | 0.00 | 98.09 | 0.00 |
| | | 1% | 88.93 | 89.72 | 97.76 | 10.56 | 97.03 | 0.00 | 97.47 | 21.06 | 98.43 | 8.64 | 92.29 | 0.02 | 96.86 | 0.03 | 96.86 | 0.03 | 98.05 | 0.00 |
| | SAPA | 5% | 98.31 | 100.00 | 98.00 | 97.31 | 96.18 | 0.83 | 98.24 | 100.00 | 98.03 | 0.02 | 96.12 | 0.00 | 96.42 | 0.06 | 96.42 | 0.06 | 98.18 | 0.10 |
| | | 1% | 98.28 | 100.00 | 98.16 | 89.42 | 98.00 | 0.25 | 98.27 | 99.96 | 98.06 | 0.02 | 90.67 | 0.00 | 96.22 | 0.00 | 96.22 | 0.00 | 97.56 | 0.64 |
| | | 0.5% | 98.44 | 100.00 | 98.43 | 55.10 | 97.95 | 0.09 | 98.46 | 0.00 | 98.63 | 0.00 | 93.15 | 0.00 | 68.65 | 0.27 | 68.65 | 0.27 | 97.64 | 0.02 |
| | LC | 0.5% | 98.02 | 0.00 | 97.76 | 0.01 | 98.02 | 0.00 | 98.03 | 0.01 | 98.02 | 0.00 | 97.93 | 0.00 | 94.66 | 0.00 | 68.80 | 0.43 | 98.01 | 0.00 |
| | | 0.1% | 98.46 | 0.01 | 98.37 | 0.00 | 98.46 | 0.01 | 98.53 | 0.01 | 98.46 | 0.01 | 98.27 | 0.00 | 93.29 | 0.03 | 95.34 | 0.02 | 98.10 | 0.00 |
| | SIG | 0.5% | 98.52 | 71.33 | 98.38 | 71.26 | 98.52 | 71.30 | 98.57 | 68.54 | 94.24 | 58.47 | 97.98 | 0.63 | 68.45 | 0.00 | 94.39 | 0.79 | 98.03 | 3.33 |
| | | 0.1% | 98.69 | 58.09 | 98.84 | 56.11 | 98.69 | 58.09 | 98.70 | 57.91 | 91.21 | 33.44 | 98.38 | 0.91 | 94.07 | 0.57 | 89.16 | 0.36 | 98.22 | 0.08 |

# J. Additional Results for Self-supervised Learning

## J.1. Self-supervised Learning with SimCLR

Figure 13 illustrates the performance of TSC and MCR against BadEncoder (Jia et al., 2022) on CIFAR10 and ImageNet using SimCLR (Chen et al., 2020). Table 12 presents the corresponding ultimate defense results of TSC and other baseline defenses. Table 12 shows the defense results against CTRL (Li et al., 2023) on CIFAR10 and ImageNet100.

Notably, for the BadEncoder attack, we utilize publicly available backdoored model checkpoints [3] as the original model, which was trained on ImageNet containing 1,000 classes with image dimensions of $224 \times 224$. As the original model for the CTRL attack is not publicly available, we follow the settings described in the original paper and train the corresponding encoder on ImageNet-100. This version of ImageNet-100 contains 100 classes. During training, we scale the image dimensions to $64 \times 64$. Furthermore, we adopt the evaluation methodology from the BadEncoder paper, which employs linear probe evaluation on downstream tasks. For the CTRL attack, we also follow the evaluation methods from the original paper: in addition to using linear probe, we employ the K-Nearest Neighbor (KNN) method to evaluate ACC and ASR on the pre-training dataset.

In defending against BadEncoder attacks, we observe that MCR, SSL-Cleanse (Zheng et al., 2024), and TSC all successfully reduce the ASR values. However, MCR fails to counter backdoor attacks targeting SVHN downstream task. SSL-Cleanse and TSC show similar defense effectiveness, successfully defending against BadEncoder attacks while maintaining high ACC values on downstream tasks. For CTRL attacks, MCR is only effective against STL10 dataset. In contrast, both SSL-Cleanse and TSC consistently maintain strong defense performance.

## J.2. Self-supervised Learning with CLIP

Table 14 presents the defense results of TSC and MCR against BadEncoder under CLIP (Radford et al., 2021). Since the unlearning algorithm used in SSL-Cleanse defense is based on the simCLR training method, and the original paper of SSL-Cleanse did not conduct experiments in the CLIP learning scenario, we do not test the defense performance of SSL-Cleanse against BadEncoder in this work.

We test on downstream datasets (STL10, GTSRB, CIFAR10, Food101 (Bossard et al., 2014), and Pascal VOC 2007 (Everingham et al.)) and report ACC and ASR using zero-shot a nd linear probe methods. TSC maintains strong defense performance against BadEncoder, though it shows lower ACC on GTSRB and CIFAR10. In contrast, it outperforms the backdoored model on Pascal VOC 2007. This may be due to the use of MS-COCO (Lin et al., 2014) for post-training. MS-COCO shares more similarity with Pascal VOC 2007, while its images differ from those in GTSRB and CIFAR10, leading the CLIP model to "forget" features for the latter datasets.

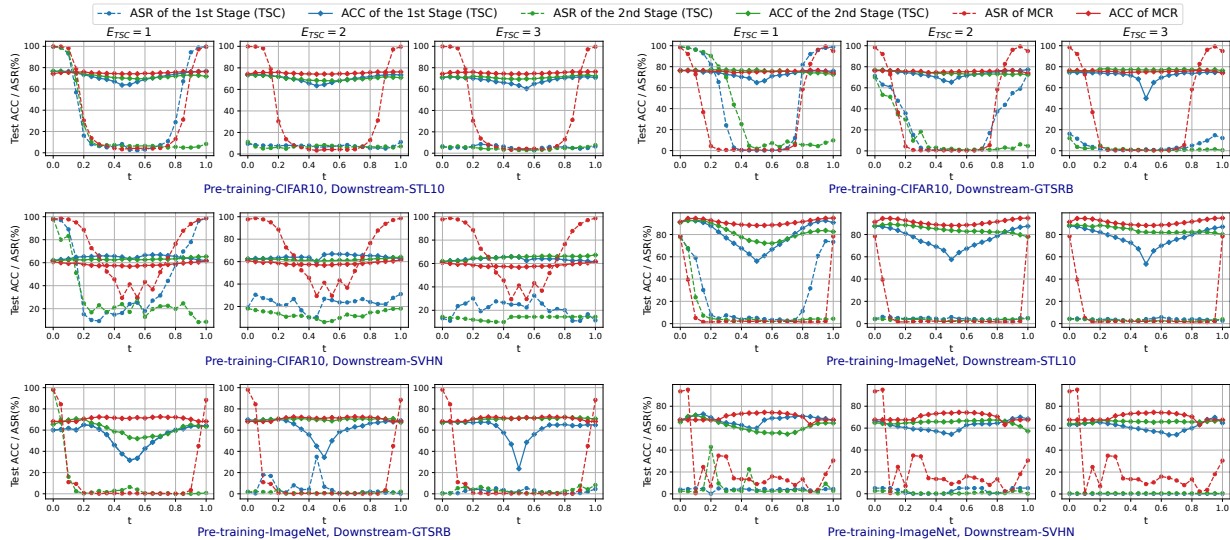

*Figure 13.* (**Self-supervised Learning, SimCLR**) Performance comparison of TSC and MCR against BadEncoder (Jia et al., 2022) attacks using CIFAR10 and ImageNet as pre-training datasets. Model checkpoints are selected at $t = 0.25$ for each stage and round.

---

[3]https://github.com/jinyuan-jia/BadEncoder

*Table 12.* Defense results under **self-supervised learning (SimCLR)** settings. We evaluate MCR (Zhao et al., 2020), SSL-Cleanse (Zheng et al., 2024), and TSC against the **BadEncoder** attack (Jia et al., 2022).

| Pre-training Dataset | Downstream Dataset | No Defense | | MCR | | SSL-Cleanse | | TSC (ours) | |
|---|---|---|---|---|---|---|---|---|---|
| | | ACC(↑) | ASR(↓) | ACC(↑) | ASR(↓) | ACC(↑) | ASR(↓) | ACC(↑) | ASR(↓) |
| CIFAR10 | STL10 | 76.74 | 99.65 | 74.93 | 7.92 | 70.51 | 2.44 | 71.11 | 4.44 |
| | GTSRB | 81.12 | 98.79 | 75.51 | 0.54 | 75.50 | 1.23 | 77.57 | 1.68 |
| | SVHN | 63.12 | 98.71 | 57.35 | 65.58 | 61.01 | 7.95 | 64.13 | 10.26 |
| ImageNet | STL10 | 94.93 | 98.99 | 90.20 | 2.08 | 88.72 | 1.69 | 86.99 | 3.11 |
| | GTSRB | 75.94 | 99.76 | 72.38 | 0.13 | 67.55 | 1.81 | 69.47 | 6.47 |
| | SVHN | 72.64 | 99.21 | 71.27 | 34.15 | 67.96 | 8.00 | 66.44 | 3.64 |

*Table 13.* Defense results under **self-supervised learning (SimCLR)** settings. We evaluate MCR (Zhao et al., 2020), SSL-Cleanse (Zheng et al., 2024), and TSC against the **CTRL** attack (Li et al., 2023). Following CTRL's evaluation methodology, we also assess ACC and ASR on the pre-training dataset using K-Nearest Neighbor (KNN) classification.

| Pre-training Dataset | Evaluation Dataset | No Defense | | MCR | | SSL-Cleanse | | TSC (ours) | |
|---|---|---|---|---|---|---|---|---|---|
| | | ACC(↑) | ASR(↓) | ACC(↑) | ASR(↓) | ACC(↑) | ASR(↓) | ACC(↑) | ASR(↓) |
| CIFAR10 (Poisoning rate-5%) | CIFAR10 (KNN) | 84.66 | 93.84 | 81.45 | 44.64 | 80.05 | 2.95 | 82.25 | 3.95 |
| | STL10 (linear probe) | 74.69 | 25.60 | 70.48 | 13.72 | 70.55 | 1.28 | 71.29 | 2.45 |
| | GTSRB (linear probe) | 72.65 | 41.80 | 71.05 | 24.54 | 69.55 | 4.16 | 70.95 | 4.25 |
| | SVHN (linear probe) | 60.42 | 62.64 | 57.35 | 32.58 | 59.22 | 0.24 | 58.91 | 7.42 |
| ImageNet100 (Poisoning rate-5%) | ImageNet100 (KNN) | 43.66 | 42.53 | 42.39 | 34.24 | 41.20 | 1.86 | 41.80 | 0.24 |
| | STL10 (linear probe) | 74.96 | 17.27 | 70.24 | 12.33 | 69.47 | 1.43 | 72.05 | 3.37 |
| | GTSRB (linear probe) | 63.58 | 68.20 | 60.85 | 50.41 | 61.85 | 5.41 | 60.73 | 1.62 |
| | SVHN (linear probe) | 56.73 | 90.77 | 56.27 | 52.77 | 53.70 | 10.04 | 54.39 | 1.21 |

*Table 14.* Defense results under **self-supervised learning (CLIP)** settings. We evaluate MCR (Zhao et al., 2020) and and TSC against the **BadEncoder** attack (Jia et al., 2022).

| Pre-training Dataset | Downstream Dataset | No Defense | | MCR | | TSC (ours) | |
|---|---|---|---|---|---|---|---|
| | | ACC(↑) | ASR(↓) | ACC(↑) | ASR(↓) | ACC(↑) | ASR(↓) |
| CLIP (linear probe) | STL10 | 97.07 | 99.33 | 96.43 | 99.86 | 94.15 | 0.67 |
| | GTSRB | 82.36 | 99.40 | 78.22 | 99.21 | 72.42 | 5.32 |
| | CIFAR10 | 86.36 | 99.98 | 84.36 | 99.45 | 79.31 | 1.32 |
| | Food101 | 72.58 | 97.91 | 72.36 | 96.62 | 69.33 | 1.04 |
| | Pascal VOC 2007 | 76.07 | 99.83 | 75.47 | 99.92 | 78.42 | 0.34 |
| CLIP (zero-shot) | STL10 | 94.06 | 99.86 | 91.51 | 99.85 | 90.25 | 0.88 |
| | GTSRB | 29.94 | 99.44 | 24.66 | 99.34 | 17.24 | 2.01 |
| | CIFAR10 | 69.95 | 99.39 | 62.51 | 99.10 | 41.59 | 1.28 |
| | Food101 | 67.72 | 99.96 | 66.51 | 99.56 | 61.69 | 0.28 |
| | Pascal VOC 2007 | 71.22 | 99.92 | 70.09 | 99.12 | 75.08 | 1.45 |

## J.3. Evaluation of I-BAU and SAU in Self-supervised Learning

Previously, we have presented the defense results of TSC against BadEncoder and CTRL attacks. Our defense, as shown in Figure 1, specifically targets self-supervised learning (SSL) scenarios by directly purifying the encoder. The other SSL defenses we evaluated, such as MCR and SSL-Cleanse, also follow this workflow. This design enables TSC to be effectively applied to zero-shot scenarios, such as CLIP, where neither a linear classifier nor fine-tuning is required.

It might be noted that other defenses designed for supervised learning (SL) settings, such as I-BAU and SAU, could also be applied to the combined encoder and linear classifier after fine-tuning. Initially, we excluded those defenses from SSL comparisons to maintain fairness and methodological consistency. To address this, we conducted additional experiments

to evaluate the performance of I-BAU and SAU against BadEncoder with 5% clean downstream training data. The other defense settings are consistent with the SL scenarios involving models trained on CIFAR10 and ImageNet100. Table 15 presents the corresponding results.

It's clear that while I-BAU and SAU reduce the ASR, they significantly degrade benign accuracy (ACC). For instance, on CIFAR10-STL10 settings, the ACC dropped from 76.73% to 30.13% with I-BAU and further to 21.52% with SAU. We suspect this decline occurs because I-BAU and SAU employ post-training methods analogous to adversarial training in SL, potentially harming the representation extraction capability of encoders trained via SSL methods like SimCLR. As these specific findings extend beyond the primary scope of this paper, we reserve a more extensive exploration for future work.

Table 15. Defense results of I-BAU (Zeng et al., 2022) and SAU (Wei et al., 2023) under SimCLR training scenario, where linear probing is used to evaluate the downstream tasks.

| Pre-training Dataset | Downstream Dataset | No Defense | | I-BAU | | SAU | |
|---|---|---|---|---|---|---|---|
| | | ACC(↑) | ASR(↓) | ACC(↑) | ASR(↓) | ACC(↑) | ASR(↓) |
| CIFAR10 | STL10 | 76.74 | 99.65 | 30.13 | 12.42 | 21.52 | 7.22 |
| | GTSRB | 81.12 | 98.79 | 22.36 | 15.10 | 47.45 | 5.52 |
| | SVHN | 63.12 | 98.71 | 43.44 | 17.11 | 24.07 | 7.25 |
| ImageNet100 | STL10 | 94.93 | 98.99 | 74.62 | 10.32 | 81.24 | 11.83 |
| | GTSRB | 75.94 | 99.76 | 39.85 | 7.34 | 10.75 | 0.80 |
| | SVHN | 72.64 | 99.21 | 25.27 | 13.70 | 24.37 | 4.10 |

## K. Clean Accuracy Drops for Non-backdoored Models

To illustrate the impact of TSC on non-backdoored models, in this section, we provide additional results on the clean accuracy (ACC) drops for no poison scenarios when applying TSC and other baseline defenses. Tables 16 and 17 present the clean ACC drops of non-backdoored models under supervised learning and self-supervised learning (SimCLR), respectively.

Except for settings specific to the attacks, the experimental configurations are consistent with those detailed in previous tables. While the ACC drops for TSC are not the lowest among all defenses, they are considered acceptable. Furthermore, the reason why NC always achieves the same ACC as the original model is that NC would check if the model is backdoored before applying it's corresponding unlearning method. The intermediate results of Table 16 show that NC successfully classifies the model as non-backdoored, thus it does not apply any unlearning method.

Table 16. (**Supervised Learning**) Clean Accuracy (ACC) drop results of various defenses on CIFAR10, GTSRB and ImageNet100 with poisoning rate of 0 (no poison).

| Dataset | No Defense | FP | NC | MCR | ANP | FT-SAM | I-BAU | SAU | TSC (ours) |
|---|---|---|---|---|---|---|---|---|---|
| CIFAR10 | 93.12 | 92.10 | **93.12** | 90.98 | 83.09 | 91.32 | 87.47 | 89.97 | 91.12 |
| GTSRB | 99.20 | 99.11 | **99.20** | 98.14 | 95.43 | 98.04 | 95.40 | 96.84 | 98.31 |
| ImageNet100 | 84.32 | 82.22 | **84.32** | 81.46 | 77.78 | 82.90 | 77.10 | 76.93 | 81.44 |

Table 17. (**Self-supervised Learning**) Clean Accuracy (ACC) drop results for SimCLR of various defenses on CIFAR10 and ImageNet100 with poisoning rate of 0 (no poison).

| Pre-training | Downstream | No Defense | MCR | SSL-Cleanse | TSC (ours) |
|---|---|---|---|---|---|
| CIFAR10 | STL10 | 79.50 | **77.63** | 73.01 | 74.60 |
| | GTSRB | 83.68 | 78.02 | **78.30** | 80.39 |
| | SVHN | 66.57 | 60.19 | **63.55** | 63.41 |
| ImageNet100 | STL10 | 95.62 | **90.92** | 89.27 | 89.81 |
| | GTSRB | 77.58 | **74.41** | 69.90 | 71.74 |
| | SVHN | 74.98 | **72.98** | 70.14 | 68.10 |

# L. Experimental Setup

Our deep learning training algorithm is implemented using PyTorch. All experiments were run on one Ubuntu 18.04 server equipped with four NVIDIA RTX V100 GPUs. Our implementation for supervised learning is mainly based on BackdoorBench [4] (Wu et al., 2022). For self-supervised learning, we use the official implementation of BadEncoder [5] (Jia et al., 2022) and CTRL [6] (Li et al., 2023) to conduct attacks and defenses experiments.

## L.1. Attack Settings

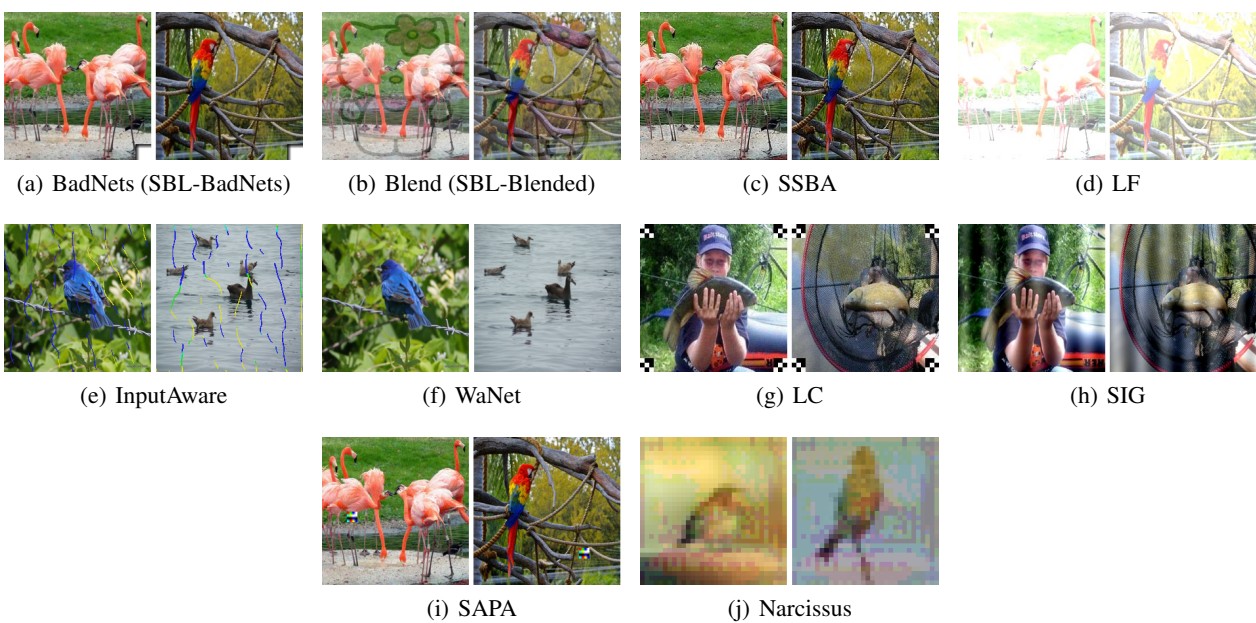

*Figure 14.* Examples of 12 supervised learning backdoor trigger patterns on ImageNet100. The triggers of SBL-BadNets and SBL-Blended are as same as those of BadNets and Blend, respectively.

(1) **Supervised Learning.** We employed the stochastic gradient descent (SGD) optimization method with a batch size of 256. We set the initial learning rate to 0.01 and decayed it using the cosine annealing strategy (Loshchilov & Hutter, 2016). For the CIFAR-10 and GTSRB datasets, we trained for a total of 100 epochs. In the case of the ImageNet100 dataset, we trained for 200 epochs.

Generally, on the CIFAR10, ImageNet100 and GTSRB datasets, the target label for all backdoor attacks is class 0, corresponding to the specific class names "airplane", "tench" and "Speed limit 20km/h". The CIFAR10 and GTSRB images are resized to $32 \times 32$, while the ImageNet100 images are resized to $224 \times 224$.

The trigger patterns for BadNet (Gu et al., 2017), Blended (Chen et al., 2017), SSBA (Li et al., 2021c), LF (Zeng et al., 2021), WaNet (Nguyen & Tran, 2021), Inputaware (Nguyen & Tran, 2020), LC (Turner et al., 2019), SIG (Barni et al., 2019), SBL (Pham et al., 2024), Narcissus (Zeng et al., 2023) and SAPA (He et al., 2024) are shown in Figure 14. BadNet, Blended, SSBA, LF, WaNet, Inputaware, SBL and SAPA are label-flipping attacks, which turns the original label of the poisoned data to the target label. LC, SIG and Narcissus are clean label attacks, which utilize the data beloning to the target class to generate poisoned samples. The implementation details for each backdoor attack are as follows:

- BadNets attack (Gu et al., 2017) employs a white square placed at the bottom-right corner as the trigger pattern.

- Blended attack (Chen et al., 2017) poisons the data by introducing a Hello Kitty image trigger. We implement the blended injection strategy, denoted as $\alpha t + (1 - \alpha)x$, to incorporate the trigger $t$ into the benign sample $x$ with a value

---

of $\alpha = 0.2$.

- SSBA (Li et al., 2021c) utilizes the StegaStemp algorithm (Tancik et al., 2020) to generate specific triggers for poison samples across various classes.

- LF (Zeng et al., 2021) employs frequency domain analysis and optimization algorithms to create poison samples.

- InputAware (Nguyen & Tran, 2020) attack requires the attacker to control the entire training process. During training, the attacker not only trains the model but also trains a generator to produce unique triggers for different samples. The generator continuously optimizes the trigger design while minimizing its size, ensuring a high attack success rate.

- WaNet (Nguyen & Tran, 2021) uses image embedding to generate invisible triggers for poisoned samples. To enhance attack robustness, WaNet introduces Gaussian noise to poisoned samples with a certain probability during training and restores the original labels of these poisoned samples.

- LC attack (Turner et al., 2019) utilizes a checkboard pattern positioned in the four corners as the trigger. To establish a link between the trigger and the target label, LC attacks initially employ the Projection Gradient Descent (PGD) method to introduce adversarial perturbations to the images before incorporating the trigger.

- SIG attack (Barni et al., 2019) utilizes a sinusoidal signal that is seamlessly integrated into the image as the trigger.

- SBL-BadNet and SBL-Blended attacks (Pham et al., 2024) employ continual learning algorithm to fine-tune poisoned models to generate backdoors that are resilient against previous fine-tuning defenses. We used EWC (Kirkpatrick et al., 2017) as the continual learning algorithm, with BadNet and Blended as base attacks.

- Narcissus attack (Zeng et al., 2023) optimizes a universal trigger pattern based a the surrogate (target) model. For fair comparison, we used the open-source trigger for CIFAR10 from Zeng et al. (2023), with target label "bird".

- SAPA attack (He et al., 2024) combines sharpness-aware minimization (Foret et al., 2021) with the Sleeper-Agent (Souri et al., 2022) backdoor attack to smooth the poison loss landscape. We utilized the colorful patch from (Souri et al., 2022) as the trigger. For CIFAR10 and GTSRB, we used an $8 \times 8$ pixel patch; for ImageNet100, a $16 \times 16$ pixel patch was employed. Other recommended parameters for generating the poisoned samples were adopted from the SAPA and Sleeper-Agent. These included: sharpness sigma = 0.01, number of source samples = 1000, $R = 250$ optimization steps, $T = 4$ retraining periods, and an $L_\infty$-norm perturbation bound of $16/255$.

(2) **Self-Supervised Learning.**

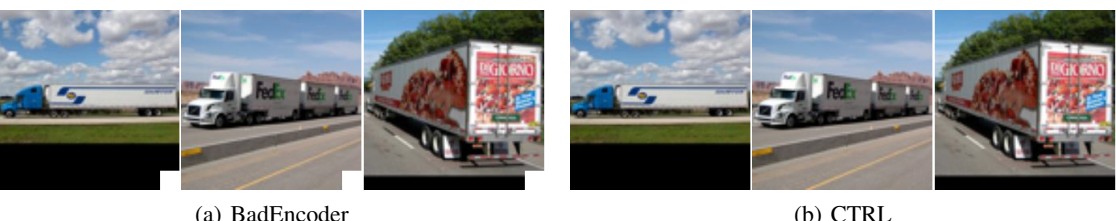

(a) BadEncoder        (b) CTRL

*Figure 15.* Examples of 2 self-supervised learning backdoor trigger patterns on STL10.

We use publicly available backdoored ResNet18 and ResNet50 models as encoders with SimCLR in the BadEncoder attack (Jia et al., 2022). For CLIP, following the BadEncoder approach, we fine-tune a pre-trained CLIP ResNet50 on ImageNet100 using SimCLR for 200 epochs to inject backdoors. For the CTRL (Li et al., 2023) attack with SimCLR, we apply the cosine annealing strategy for the learning rate. For the CIFAR10 dataset under CTRL, we set the initial learning rate to 0.06 and run for 800 epochs. For the ImageNet100 dataset under CTRL, the initial learning rate is set to 0.02 with 1000 training epochs.

Tables 18 and 19 show the dataset and target class settings for BadEncoder and CTRL in different learning scenarios. The post-training dataset refers to the fine-tuning dataset used by each defense method. When the downstream dataset lacks the target class, we add the corresponding class and include target class data from the pre-training dataset to evaluate ACC and ASR. To align with the CTRL paper's settings, we use a $64 \times 64$ ImageNet100 dataset as the pre-training dataset.

The implementation details for BadEncoder and CTRL attacks are as follows:

*Table 18.* Dataset and target class settings for BadEncoder attack. If the "Requiring extra data" column is marked as '✓', it indicates that the downstream dataset lacks the target class and we manually add the target class and data to the dataset for evaluating ACC and ASR.

| Method | Image Size | Pre-training | Downstream | Post-training | Target class | Requiring extra data |
|--------|-----------|--------------|------------|---------------|--------------|---------------------|
| simCLR | 32×32 | CIFAR10 | STL10 | CIFAR10 | truck | ✗ |
| simCLR | 32×32 | CIFAR10 | GTSRB | CIFAR10 | priority road sign | ✗ |
| simCLR | 32×32 | CIFAR10 | SVHN | CIFAR10 | 1 | ✗ |
| simCLR | 224×224 | ImageNet | STL10 | ImageNet100 | truck | ✗ |
| simCLR | 224×224 | ImageNet | GTSRB | ImageNet100 | priority road sign | ✗ |
| simCLR | 224×224 | ImageNet | SVHN | ImageNet100 | 1 | ✗ |
| CLIP | 224×224 | CLIP | STL10 | MS-COCO | truck | ✗ |
| CLIP | 224×224 | CLIP | GTSRB | MS-COCO | stop sign | ✗ |
| CLIP | 224×224 | CLIP | SVHN | MS-COCO | 0 | ✗ |
| CLIP | 224×224 | CLIP | Food101, VOC 2007 | MS-COCO | truck | ✓ |

*Table 19.* Dataset and target class settings for CTRL attack.

| Method | Image Size | Pre-training | Downstream | Post-training | Target class | Requiring extra data |
|--------|-----------|--------------|------------|---------------|--------------|---------------------|
| simCLR | 32×32 | CIFAR10 | STL10 | CIFAR10 | airplane | ✗ |
| simCLR | 32×32 | CIFAR10 | GTSRB | CIFAR10 | airplane | ✓ |
| simCLR | 32×32 | CIFAR10 | SVHN | CIFAR10 | airplane | ✓ |
| simCLR | 64×64 | ImageNet100 | STL10 | ImageNet100 | tench | ✗ |
| simCLR | 64×64 | ImageNet100 | GTSRB | ImageNet100 | tench | ✓ |
| simCLR | 64×64 | ImageNet100 | SVHN | ImageNet100 | tench | ✓ |

- BadEncoder attack (Jia et al., 2022) requires the attacker to control the training process. During training, the attacker uses the SimCLR loss function to optimize the similarity between poisoned samples and target class images. Similar to BadNets, we use a white square in the bottom-right corner as the trigger. The loss function assigns equal weight to clean and poisoned samples, i.e., $\lambda_1 = \lambda_2 = 1$.

- CTRL attack (Li et al., 2023) uses a invisible frequency trigger to generate poisoned samples. Following the recommended settings, we set the poisoning rate to 0.5% and the trigger window size to 32. For CIFAR10, the trigger magnitude is set to 100. For ImageNet100, we use a magnitude of 100 for the first 700 epochs and a magnitude of 200 for the remaining 300 epochs.

### L.2. Defense Settings

We allocate 5% of the clean training samples from each dataset to the defender. For all other settings not specified below, we follow the default settings outlined in their publications or public implementation.

Since we use the original mathematical symbols from each publication, please note that some symbols and terms may conflict with each other. All symbols and terms below are specific to the parameters in their respective papers.

(1) **Supervised Learning.**

- FP (Liu et al., 2018): In the experiment, we use SGD as the base fine-tuning method with a learning rate of 0.01. The pruning rate for all models is set to 1%, and the number of fine-tuning epochs is 100.

- NC (Wang et al., 2019): We set the threshold of the Anomaly Index at 2 for all the datasets. For models with an Anomaly Index higher than 2 (marked as attacked), we conduct the unlearning procedure for 40 epochs, utilizing 5% of the training data and applying the reversed trigger to 20% of these samples.

- ANP (Wu & Wang, 2021): We set the learning rate to 0.2 for optimizing the neuron mask with SGD. After optimization, neurons with a mask value smaller than the threshold of 0.2 are pruned. Additionally, we set the tradeoff coefficient $\alpha$ to 0.2 and the perturbation budget $\epsilon$ to 0.4 for a total of 2000 iterations.

- FT-SAM (Zhu et al., 2023): SGD is used as the base fine-tuning method with a learning rate of 0.01. The label smoothing rate is set to 0.1, and the SAM (Sharpness-aware Minimization) (Foret et al., 2021) optimization method uses a Neighborhood size of $\rho = 0.2$. The number of fine-tuning epochs is 100.

- I-BAU (Zeng et al., 2022): We follow the default settings provided in the public implementation of I-BAU. We use Adam with a learning rate of $1e^{-3}$ to optimize the outer loop. And, we set the maximum number of unlearning rounds to 5 and the maximum number of fixed-point iterations to 5.

- SAU (Wei et al., 2023): We iterate 5 times to compute the adversarial perturbation, using an $\ell_\infty$ norm with perturbation bound 0.2 and adversarial learning rate 0.2. The total epochs for CIFAR10 and GTSRB are 100, and for ImageNet100, 50 epochs. We set the unlearning weights as $\lambda_1 = \lambda_2 = \lambda_4 = 1$ and $\lambda_3 = 0.1$.

- TSC (ours): We set the global number of epochs $E_{TSC} = 3$, curve index $t = 0.4$, and curve training epochs $e = 200$. The initial learning rate is set to 0.02, with cosine annealing used for post-training. After every round of TSC, we slightly fine-tune the model extracted from the second stage for 5 epochs with a learning rate of $1e^{-4}$.

- MCR (Zhao et al., 2020): For a fair comparison with TSC, we set the curve index $t = 0.4$ and curve training epochs $e = 200$. The initial learning rate is 0.02, and cosine annealing is used for post-training.

(2) **Self-Supervised Learning.**

- SSL-Cleanse (Zheng et al., 2024): This method is specifically designed for defending against backdoors in self-supervised learning. Similar to NC, it first attempts to restore the trigger pattern and then applies an unlearning method to remove the backdoor. We set the initial value of $\lambda$ to 0.01 and use a learning rate of 0.01.

- TSC (ours): We set the global number of epochs $E_{TSC} = 2$ and the curve index $t = 0.25$. All learning rates use a cosine annealing strategy for iteration. (1) For SimCLR, we set the batch size to 256, the curve training epochs $e = 200$, and use Adam with an initial learning rate of $5e^{-3}$ for CIFAR10 and $2e^{-3}$ for ImageNet100. After every round of TSC, we slightly fine-tune the model extracted from the second stage for 5 epochs with a learning rate of $5e^{-5}$. (2) For CLIP, we follow the training method proposed by Radford et al. (2021), setting the batch size to 32768 and using Adam with an initial learning rate of $1e^{-4}$ ($\beta_1 = 0.9$, $\beta_2 = 0.999$). For such a large batch size, we use gradient accumulation to update the model, with each stage training for only 2 epochs. After every round of TSC, we do not fine-tune the model extracted from the second stage.

- MCR (Zhao et al., 2020): We set the curve index $t = 0.25$. All other parameters are the same as those used in TSC.

