# OpenReview forum: "Circumventing Backdoor Space via Weight Symmetry"
_ICML.cc/2025/Conference — ICML 2025 poster_

### Official Review · Reviewer_AW4n · 2025-03-07

**Overall Recommendation:** 4

**Summary:**

The paper highlights the vulnerability of deep neural networks to backdoor attacks, which can compromise model integrity and lead to unauthorized access or malfunction. The proposed method TSC leverages the concept of weight symmetry to purify models. It trains a quadratic Bezier curve in the parameter space, connecting two endpoint models, and selects a point along this curve as the final model to mitigate backdoor attacks.

**Claims And Evidence:**

Yes

**Essential References Not Discussed:**

/NA

**Experimental Designs Or Analyses:**

Yes

**Methods And Evaluation Criteria:**

Yes

**Other Comments Or Suggestions:**

/NA

**Other Strengths And Weaknesses:**

## Strengths

The paper emphasizes the potential societal benefits of enhancing model security by eliminating backdoor behavior across various machine learning scenarios.

The proposed method is evaluated against various backdoor attacks under both supervised and self-supervised learning settings. Results demonstrate TSC's robustness across different datasets and model architectures.

## Weaknesses

The details on how the adaptive attacks were designed and implemented are sparse.

The evaluation is primarily conducted on CIFAR-10, ImageNet100, and GTSRB datasets. While these datasets are commonly used in the field, it would be beneficial to see the performance of TSC on a wider range of datasets, particularly in real-world scenarios where backdoor attacks may be more sophisticated.

**Questions For Authors:**

/NA

**Relation To Broader Scientific Literature:**

/NA

**Theoretical Claims:**

Yes

---

> ### Author Rebuttal · Authors · 2025-04-01
>
> We sincerely appreciate your insightful comments! We address your concerns as below.
>
> **W1: Adaptive Attack Design**
>
> Due to space limitations, the initial version includes implementation details and the design of the adaptive attack in **Appendix F**, with the attack process outlined in **Algorithm 3**, which provides a detailed intuition and workflow of our designed adaptive attack.
> Moreover, for Reviewer uSaT, the adaptive attack is considered strong.
>
> During the rebuttal period, we further analyzed the attack and found that, via Eq. (8), TSC consistently identifies an endpoint in a different loss basin from both $\theta_{adv}$ and $\theta_{adv}'$ in Algorithm 3. To better illustrate this, we will include additional loss landscape visualizations (similar to Fig. 2) in the revised version to demonstrate how TSC locates this endpoint.
>
> **If you have specific questions about our adaptive attack, we welcome further discussion.**
>
>
> **W2: Additional Datasets**
>
> In addition to CIFAR-10, ImageNet100, and GTSRB, in the initial version,
> a variety of other datasets in the self-supervised learning scenario were employed in our experiments. Detailed information on these datasets can be found in **Tables 15 and 16 of Appendix J**.
>
> For instance, in the case of the CLIP model, we fine-tuned the backdoored CLIP model on the **MS-COCO** dataset and conducted downstream task experiments on **five datasets: STL10, GTSRB, SVHN, Food101, and VOC 2007** .
>
> We believe the current results on these datasets sufficiently demonstrate the generalizability of our algorithm.
>
> **W2: Additional Advanced Attacks**
>
> Following your feedback and Reviewer uSaT's comments, we are also considering implementing attacks based on the loss landscape of poisoned samples, including SPSA [1], SBL [2], and Narcissus [3], in the supervised learning setting. These modern adaptive backdoor attacks optimize flatter loss landscapes or entangle benign and backdoor features.
>
> Due to time constraints, we included experiments against SBL [2] and Narcissus [3]. We are actively working on incorporating SPAP [1] and will provide results in a future discussion.
>
> For the experiments with the PreActResNet18 on CIFAR-10, we present the following comparative results (we will also include similar experiments for ImageNet100 and GTSRB).
> - For the SBL attack [2], we used EWC as the continual learning algorithm, with BadNet and Blended as base attacks, and experimented with poison rates of 10%, 5%, and 1%.
> - For the Narcissus attack [3], we used the open-source triggers from [3], with poison rates of 5%, 1%, and 0.5%. Other experimental settings were consistent with the original paper.
>
> The results show that TSC effectively defends against these attacks. The Narcissus attack (poison rate = 5%) exhibits the strongest attack effect on our defense, but its ASR remains below 20%.
>
> We plan to include the corresponding results in the revised version, along with ASR/ACC plots for these attacks as functions of t, similar to Figure 3.
>
>
> |Attack|Poison Rate|No Defense|||FP|||NC||||MCR|||ANP|||FT-SAM|||I-BAU|||SAU|||TSC (ours)||
> |---|---|---|---|---|---|---|---|---|---|---|---|---|---|---|---|---|---|---|---|---|---|---|---|---|---|---|---|---|
> |||ACC|ASR||ACC|ASR||ACC|ASR|||ACC|ASR||ACC|ASR||ACC|ASR||ACC|ASR||ACC|ASR||ACC|ASR|
> |**SBL-BadNet**[2]|10%|91.30|95.11||92.00|91.07||91.63|0.34|||91.37|70.99||90.72|0.00||91.41|89.13||88.99|17.09||90.63|1.52||**90.36**|**0.21**|
> ||5%|90.79|93.48||92.59|1.13||92.22|0.59|||92.26|91.82||82.82|51.63||92.16|60.03||90.67|27.06||91.31|0.60||**91.02**|**1.12**|
> ||1%|91.71|88.64||93.10|31.77||91.82|0.72|||93.23|86.11||82.71|81.48||92.77|59.58||90.63|2.00||92.32|1.01||**91.54**|**1.93**|
> |**SBL-Blended**[2]|10%|90.46|94.12||92.49|29.61||90.46|88.12|||92.51|99.91||86.32|52.96||92.40|74.02||91.44|22.79||88.11|9.09||**90.98**|**8.27**|
> ||5%|91.70|97.67||92.97|79.74||91.70|97.67|||92.75|99.61||85.13|20.48||92.50|77.90||89.65|57.41||91.43|11.53||**90.47**|**8.94**|
> ||1%|92.07|91.84||93.43|83.80||92.07|91.84|||93.37|95.02||85.30|58.19||93.31|82.64||90.67|64.08||92.34|16.31||**90.11**|**6.70**|
> |**Narcissus**[3]|5%|93.72|90.91||91.93|68.61||93.72|80.91|||93.63|86.64||87.87|49.27||93.19|27.92||87.82|73.79||90.72|1.57||**91.35**|**14.48**|
> ||1%|93.68|82.87||92.29|44.88||93.68|47.87|||93.61|49.79||92.01|27.01||93.05|26.80||90.21|18.67||91.36|3.24||**90.65**|**7.88**|
> ||0.5%|93.68|80.58||92.94|29.59||93.67|32.57|||93.69|32.96||89.35|16.78||93.06|14.08||89.16|21.09||91.74|5.81||**91.71**|**8.02**|
>
> [1]: He et al. Sharpness-Aware Data Poisoning Attack. ICLR 2024.
>
> [2]: Pham et al. Flatness-Aware Sequential Learning Generates Resilient Backdoors. ECCV 2024.
>
> [3]: Zeng et al. Narcissus: A practical clean-label backdoor attack with limited information. CCS 2023.

---

### Official Review · Reviewer_uSaT · 2025-03-09

**Overall Recommendation:** 3

**Summary:**

This paper introduces a backdoor purification method called Two-stage Symmetry Connectivity (TSC). The approach is devided into two stages, aiming to use permutation invariance and mode connectivity to circumvent backdoor spacenwhile maintaining clean accuracy. The method is designed to be applicable beyond supervised learning, including self-supervised learning frameworks such as SimCLR and CLIP. Theoretical analysis and empirical results is provided to support the approach.

## update after rebuttal
The authors have addressed most of my concerns. So I decide to raise my score to weak accept.

**Claims And Evidence:**

Yes. The core claims (i.e., TSC can effectively remove backdoors across different learning paradigms by utilizing mode connectivity and weight symmetry) are backed by empirical results in various settings. However, some theoretical claims (or assumptions) are not rigorously justified. Specifically, the assumption that the first-stage unalignment always increases backdoor loss and the second stage re-alignment would only decrease clean loss is not thoroughly explained, i.e.,

- The claim that Eq. (8) optimization increases poisoned sample loss, despite being trained only on benign samples, is not well-supported.
- Corollary 4.3 states that re-aligning the model with clean samples leads to selective clean accuracy recovery, but it lacks concrete justification.

**Essential References Not Discussed:**

Yes. I recommend the authors to evaluate the method's effectiveness on modern adaptive backdoor attacks that optimize flatter loss landscapes or entangle benign & backdoor features, which may violate the assumption of the paper:

[1]: He et al. Sharpness-Aware Data Poisoning Attack. ICLR 2024.

[2]: Pham et al. Flatness-Aware Sequential Learning Generates Resilient Backdoors. ECCV 2024.

[3]: Zeng et al. Narcissus: A practical clean-label backdoor attack with limited information. CCS 2023.

**Experimental Designs Or Analyses:**

Yes. The experimental setup appears rigorous, covering a variety of datasets, attack strategies, and learning settings. The authors also evaluated their approach against potential adaptive attacks. The results provide good empirical support for the method's effectiveness. However:

- The paper does not fully explain why unaligning the model in the first stage effectively increases the backdoor loss. It assumes that benign and poisoned samples lie in easy-to-separate loss basins, but this may not always hold.
- The paper proposes an adaptive attack but does not provide sufficient analysis of why it fails against TSC. Further exploration would strengthen the paper.

**Methods And Evaluation Criteria:**

Yes. The paper uses Attack Success Rate (ASR) and Clean Accuracy (ACC) as primary evaluation metrics, which are widely used in backdoor defense research. The experimental setup appears rigorous, covering a variety of datasets, attack strategies, and learning settings. The authors also evaluated their approach against potential adaptive attacks.

**Other Comments Or Suggestions:**

Minor Issues:

Line 297: "similary" → "similar"

Line 324: "origianl" → "original"

**Other Strengths And Weaknesses:**

## Strengths:
- Backdoor attacks are a major security risk in machine learning and defending against them is thus very important.
- Unlike many existing defenses, TSC is applicable to learning paradigms beyond supervised learning (e.g., SSL), broadening its impact.
- The experiments demonstrate that TSC effectively reduces ASR while maintaining competitive ACC.
- Discovering defenses from the perspective of loss landscape and mode connectivity is quite interesting and valuable as it provides some level of interpretability.


## Weaknesses:
- The paper does not clearly explain why the first stage's unalignment effectively increases backdoor loss. It appears to rely on the assumption that benign and backdoor samples occupy significantly different loss landscape basins, making the interpolation likely to land in a high-loss region for backdoor samples. However, this assumption may not hold for all backdoor attack methods, particularly those that already tightly couple benign and trigger-related features or train models with flatter backdoor loss landscapes. Since the paper does not experiment with such methods and only provides a loose theoretical bound on the upper loss relationship, I suggest adding more experiments to validate the generalizability of the approach on more advanced backdoor attacks [1-3].
- The authors claim that optimizing according to Eq. (8) can increase the loss of poisoned samples. However, in practice, the training process only involves benign samples. How does this ensure that poisoned samples experience a loss increase? Additionally, why does the method require an argmax operation? Would this not cause a significant degradation in benign sample performance for the adversarially updated model $θ'_{adv}$?
- The theoretical contributions of the paper appear somewhat incremental. Theorem 4.2 mainly states that feature-aligned networks exhibit mode connectivity curves with smaller loss expectations, which has already been explored in prior work (Tatro et al.). The paper extends this idea but does not provide fundamentally new insights. Furthermore, Eq. (2) and Eq. (9) appear different in formulation—why is the latter necessary, and are they truly equivalent? Additionally, Corollary 4.3 claims that the proposed optimization increases loss along the mode connectivity curve for both clean and poisoned samples. However, in the second stage, when re-aligning to the original model’s basin, the authors assume that only the clean sample loss will significantly decrease. This claim lacks justification.
- The notation is somewhat complex, making the paper difficult to follow to some extent. For example, Ml(θA, θB ; D) is introduced early (Line 170) but used much later, creating confusion. The PERMUTELAYERS function is also recommended to be presented as pseudocode for clarity.
- The adaptive attack introduced in the paper appears to be a strong method against TSC. However, it fails against the proposed defense. Why does this happen? Additional analysis would improve the comprehensiveness of the paper's understanding of the method’s limitations.


The reviewer will actively participate in the rebuttal and would like to increase the score if the aforementioned concerns are alleviated properly.

**Questions For Authors:**

- Why does the first stage’s unalignment procedure work effectively to increase backdoor loss? Could this assumption fail for attacks with highly entangled benign and backdoor features?
- How does optimizing Eq. (8) ensure that poisoned samples receive a higher loss, given that training only involves benign samples?
- Why is the formulation in Eq. (9) necessary, and is it equivalent to Eq. (2)?
- Why does the second stage of the method selectively lower clean sample loss while maintaining high loss for poisoned samples?
- The adaptive attack appears strong, yet it fails against TSC. Why is this the case? A deeper analysis would be beneficial.

**Relation To Broader Scientific Literature:**

This paper is relevant to the field of backdoor defenses and aligns with recent work on mode connectivity and weight permutation strategies. However, it does not sufficiently compare with backdoor defenses that actively disentangle benign and poisoned features or strengthen their resistance through finding flat backdoor minima — which might be critical in evaluating TSC’s generalizability.

**Theoretical Claims:**

Yes, I briefly looked through the proofs. Overall, the claims align with intuition and the conclusions of previous studies. I skimmed the detailed proofs in the Appendix and did not notice any obvious issues, but I did not follow each step of the derivation in depth due to time constraints.

---

> ### Author Rebuttal · Authors · 2025-04-01
>
> We sincerely appreciate your insightful comments! We address your concerns as below.
>
> **W1.Assumption of TSC and Advanced attacks**
>
> The assumption that "benign and backdoor samples occupy significantly different loss landscape basins" does not fully correspond to our method. Instead, the underlying mechanism we rely on is based on research in the field of mode connectivity (Frankle et al., Tatro et al.). In these studies, it is observed that the curve connecting two aligned models (i.e., models lying in the same loss basin) typically exhibits lower loss compared to the curve connecting misaligned models. In Stage 1 of TSC, we leverage this mechanism to amplify the loss of poisoned samples.
>
> Due to time constraints, we included experiments against SBL [2] and Narcissus [3], and the results are discussed in **W2: Additional Advanced Attacks for Reviewer AW4n**. We are actively working with SPAP[1] and will include the results in a future discussion.
>
> **W1, W2, Q1, Q2 & Q4: Mechanism of TSC**
>
> Due to space limitations, some explanations were deferred to Appendices A-C. We explain the mechanism of TSC in the following workflow:
> 1. Unalignment
>    - Eq.8 is used to compute the permutation $P_l'$, which projects the backdoor model $\theta_{adv}$ into a different loss basin. As noted in Sec.3.2, the permutation operation does not change the original function output for the same input **(i.e., $f(x, \theta_{adv}) = f(x, \theta_{adv}')$). Thus, solving the optimization problem in Eq.8 does not increase the loss of poisoned or benign samples for $\theta_{adv}'$**.
>    - Since solving Eq.6 (argmin) aligns two models, we aim to unalign $\theta_{adv}$ and $\theta_{adv}'$ by solving the opposite problem (argmax, Eq.8) in this step.
>
> 2. Training the Curve $\gamma$:
>    - We then train the curve $\gamma$, connecting $\theta_{adv}$ and $\theta_{adv}'$, and pick a midpoint along $\theta_t$ as the core step to increase the loss for poisoned samples.
>    -  As mentioned in rebuttal section *W1.Assumption of TSC and Advanced attacks*, unaligning the models increases the loss along the connecting curve.
>    - However, once trained, the curve (e.g., Bezier curve in our method) can find a low-loss path even between unaligned models (Garipov et al.). **As we only train $\gamma$ with benign samples, the loss of benign samples remians low while the adversarial loss can be amplified along the curve.** As shown in the middle of Fig.2, adversarial loss is notably high along $\gamma$. Meanwhile, the benign ACC of $\theta_t$ we picked degrades as the endpoints are unaligned. Thus, we need Stage 2 to retain the clean accuracy.
>
> 3. Second Stage: Re-aligning to Retain Clean Accuracy
>    - Inspired by model fusion literature (Tatro et al., Ainsworth et al.), we recover clean accuracy by re-aligning $\theta_t$ with $\theta_{adv}$.
>    - Given that $\theta_t$ and $\theta_{adv}$ may not lie in the same loss basin for benign samples, we re-align them using Eq.6. After alignment, the benign loss along the curve connecting $\theta_t*$ and $\theta_{adv}$ is expected to be low, which helps recover accuracy.
>    - Again, aligning does not alter the model's output for a fixed input. **Hence, after being aligned with $\theta_{adv}$, $\theta_t$ still exhibits high loss for poisoned samples. Since $\theta_{adv}$ is the original poisoned model, the distribution difference between $\theta_{adv}$ and $\theta_t$ leads to an increase in adversarial loss as the point approaches the aligned $\theta_t$.**
>
> **W4: Complex Notations**
>
> We agree that some notations may be difficult to follow. A more detailed response can be found in the rebuttal section **W1: Difficulty with Notation and Writing Clarity for Reviewer XfkX**.
>
> **W3, Q3 & Q4: Contribution and Concerns of Theorem**
>
> As mentioned in Line 297, Tatro et al.'s work covers linear mode connectivity, but their result doesn't directly extend to quadratic mode connectivity, such as with Bezier curves. Our Theorem 4.2 addresses this gap by providing the proper analysis and incorporating $M$ for models with varying feature distances.
>
> Eq.2 and Eq.9 are equivalent; we use Eq.9 for clarity, and the derivation is provided in Eq.14.
>
> As mentioned above and to the right of Line 299, the distribution differences at the endpoints of Stage 2 account for these results. We will include the above discussion in the final manuscript.
>
> **W5 & Q5: Evaluation of Adaptive Attacks**
>
> The adaptive attack is indeed designed to learn a backdoored model that maintains a low backdoor loss along the defensive curve identified by Stage 1 of TSC. However, we find that, via Eq.8, TSC always finds an endpoint in a different loss basin from the both $\theta_{adv}$ and $\theta_{adv}'$ in Algorithm 3. We will include additional loss landscape visualizations (similar to Fig.2) in the revised version to illustrate how TSC locates a endpoint.
>
> **Typos**: We will correct these typos in the revised version.

---

> > ### Comment · Reviewer_uSaT · 2025-04-04
> >
> > Dear authors, thanks for your rebuttal. I think it addresses some of my main concerns and so I would like to raise my score to 3. Thank you.

---

> > > ### Author Response · Authors · 2025-04-09
> > >
> > > Dear Reviewer,
> > >
> > > Thank you for taking the time to consider the additional clarifications we provided and the score raise!
> > >
> > > We have conducted additional experiments to assess the robustness of our method against the **SAPA attack** [1], which was not included in our previous response.
> > >
> > > Specifically, we followed the procedure outlined in [1] and combined sharpness-aware minimization [5] with the Sleeper-Agent [4] backdoor attack to perform SAPA attack. The results presented below are for the PreActResNet18 model on the CIFAR-10 dataset. We used the recommended parameters from SAPA and Sleeper-Agent to generate the poisoned samples:
> > >
> > >  - sharpness sigma = 0.01, number of source data = 1000, optimizing
> > >    steps R = 250, retraining periods T = 4, and $l_\infty$-norm
> > >    bounded by 16/255.
> > >  - The colorful patch from [4] was used as the trigger, with the target
> > >    label set to 0 and the source data coming from class 1.
> > >  - To ensure fairness and complementarity, we evaluated SAPA under three
> > >    poison rates: 5%, 1%, and 0.5% (1% poison rate is the main setting
> > >    used in [1] and [4]).
> > >
> > > It's clear that SAU and TSC are the most effective defenses against SAPA attack. Moreover, we observed that SAPA attack, when using smaller poison rates (1% and 0.5%), is more robust to certain defenses, such as ANP, FT-SAM, and I-BAU, than when using a higher poison rate (5%).
> > >
> > > As noted in [1], the sharpness-aware minimization [5] in SAPA is employed to find the ***Worst-case Poisoned Model***, which has the worst poisoning effect. While SAPA does help smooth the loss landscape (as shown by He et al. [1]), it mainly focuses on improving poisoning effect under various re-training uncertainty (such as differences in training algorithms, model initialization, and model architectures compared to the settings used by the attacker to generate poison samples).
> > >
> > > We will include these results and discussion in the revised version.
> > >
> > > | Attack Method | Poison Rate |  | No Defense |  |  | FP |  |  | NC |  |  | MCR |  |  | ANP |  |  | FT-SAM |  |  | I-BAU |  |  | SAU |  |  | TSC (ours) |  |
> > > |---|---|---|---|---|---|---|---|---|---|---|---|---|---|---|---|---|---|---|---|---|---|---|---|---|---|---|---|---|
> > > |  |  |  | ACC | ASR |  | ACC | ASR |  | ACC | ASR |  | ACC | ASR |  | ACC | ASR |  | ACC | ASR |  | ACC | ASR |  | ACC | ASR |  | ACC | ASR |
> > > | SASP [1] | 5% |  | 93.57 | 100.00 |  | 92.56 | 41.88 |  | 92.76 | 2.51 |  | 93.25 | 100.00 |  | 84.83 | 1.14 |  | 92.80 | 8.40 |  | 88.51 | 1.44 |  | 91.39 | 3.30 |  | **91.13** | **4.51** |
> > > |  | 1% |  | 94.01 | 99.97 |  | 92.34 | 92.22 |  | 92.80 | 2.14 |  | 93.83 | 100.00 |  | 86.06 | 92.68 |  | 93.06 | 79.80 |  | 86.69 | 15.17 |  | 91.83 | 1.96 |  | **90.37** | **7.41** |
> > > |  | 0.5% |  | 93.77 | 84.80 |  | 88.82 | 82.76 |  | 92.74 | 1.52 |  | 93.78 | 80.83 |  | 87.99 | 81.52 |  | 93.23 | 82.02 |  | 90.16 | 26.48 |  | 91.75 | 0.68 |  | **90.98** | **7.32** |
> > >
> > > [1]: He et al. Sharpness-Aware Data Poisoning Attack. ICLR 2024.
> > >
> > > [4]: Souri et al. Sleeper Agent: Scalable Hidden Trigger Backdoors for Neural Networks Trained from Scratch. NeurIPS 2022.
> > >
> > > [5]: Foret et al. Sharpness-aware minimization for efficiently improving generalization ICLR 2021.

---

### Official Review · Reviewer_XfkX · 2025-03-09

**Overall Recommendation:** 4

**Summary:**

The authors proposed an extension to Mode Connectivity Repair (MCR) [Zhao et al. (2020)]. The task is to purify a backdoored model using a small number of clean samples. MCR uses the poisoned model and a fine-tuned model to find an intermediate model that can lower the attack effectiveness.

The proposed model introduces an extra stage to induce a more misaligned alternative model. This involves two stages:

Stage 1. Instead of using a fine-tuned model directly, permute the latent nodes to introduce a loss barrier between the new model and the original model (as shown in Figure 2b). Construct a curve through the barrier, and then along the curve, select an intermediate model that is more misaligned than a fine-tuned model.
Stage 2. Use the model from Stage 1, then construct another curve that allows us to find a different model with a low loss for clean data but likely a high loss for poisoned data. This stage is similar to MCR.

**Claims And Evidence:**

Overall, there is a broad set of empirical evaluation, with reasonable results supporting the claims on performance gains.

**Essential References Not Discussed:**

N/A

**Experimental Designs Or Analyses:**

The experimental designs seem reasonable with my cursory check.

**Methods And Evaluation Criteria:**

yes.

**Other Comments Or Suggestions:**

N/A

**Other Strengths And Weaknesses:**

Strengths:
- Novel combination of permutation and mode connectivity

Weaknesses:
- I find the writing difficult to follow, with confusing notation choices and sometimes missing important information.

**Questions For Authors:**

1. Can you include the ACC drop results for Poison Rate=0 (no poison)?
2. In Algorithm 1, why use the same curve index t for both stages?

**Relation To Broader Scientific Literature:**

The work is related to purifying poisoned models through purification. Prior work uses mode connectivity to induce a high-quality fine-tuned model. This work uses permutation invariance to create a loss barrier targetting the poisoned samples. This overcomes the challenge of insufficient difference in a continuously fine-tuned model. The idea is novel and interesting.

**Theoretical Claims:**

I have not verified the proofs, although the Lipschitz condition seems unrealistic.

---

> ### Author Rebuttal · Authors · 2025-03-31
>
> We sincerely appreciate your insightful comments! We address your concerns as below.
>
> **Theoretical Claims: Lipschitz Condition**
>
> The Lipschitz condition required in Theorem 4.2 and Corollary 4.3 specifically applies to the loss and activation functions, which is realistic and commonly satisfied by standard deep learning practices. For instance:
>
> - Softmax Cross-Entropy Loss: Although the original cross-entropy loss alone is not Lipschitz continuous, the softmax normalization—commonly used in classification tasks—renders the resulting softmax cross-entropy loss Lipschitz continuous with constant $\sqrt{2}$ under the $\ell_2$-norm. This condition is sufficient to derive Eq.34 (Line 830) in our proof.
> - Activation Functions:
>   - ReLU: 1-Lipschitz
>   - Sigmoid: $\frac{1}{4}$-Lipschitz
>   - Tanh: 1-Lipschitz
>
> Thus, the Lipschitz condition utilized in our analysis is practically relevant and aligned with typical neural network implementations.
>
> **W1: Difficulty with Notation and Writing Clarity**
>
> Thank you for your valuable feedback.
> We agree that some notations may have been unclear or inadequately explained.
> To address this, we plan to revise the manuscript as follows:
>
> - **Clarifying Notation**: For notations defined in the preliminaries, such as $M_l(\theta_A, \theta_B;D)$, we will revisit their basic meaning when they are specifically employed in the analysis.
>
> - **Simplifying Complex Notations**: For notations involving complex symbols, such as $W_2(\mathbb{P}\_{x_l^{adv}}, \mathbb{P}\_{x_l^{adv*}}; D_{adv})$, we will simplify them by replacing the inner parts with more straightforward and clear symbols.
> For example, we will denote the distribution $\mathbb{P}\_{x_l^{A}}$ as $\mathbb{P}\_l ^A$ and rewrite $W_2(\mathbb{P}\_{x_l^{ adv}}, \mathbb{P}\_{x_l^{adv*}}; D_{adv})$ as $W_2(\mathbb{P}\_l^{\\; adv}, \mathbb{P}\_l^{\\; adv*}; D_{adv})$ for better clarity.
>
> Due to space limitations in the initial version, certain details about mode connectivity and permutation invariance were placed in Appendices A-C. Additionally, some intuition and mechanisms behind our method were left out of the main text for brevity.
>
> **Given the additional page allowance in the final version, we will move relevant content into the main text to improve readability and ensure all essential details are more accessible.**
> The concrete pseudocode for the function ***PermuteLayers*** will also be included in Appendix E.
>
> **Q1: ACC Drop Results for Poison Rate=0**
>
> We have included the ACC drop results for Poison Rate=0 (i.e., clean model) in the following tables:
>
> - Supervised Learning (SL): The first table shows the results for CIFAR-10, GTSRB, and ImageNet100 in supervised learning settings.
>
> - Self-Supervised Learning (SSL-SimCLR): The second table presents results for two pre-training datasets, CIFAR-10 and ImageNet100, along with their corresponding downstream datasets.
>
> The training settings are consistent with those in Tabs.1 and 2, except for the attack settings.
>
> It's clear that ACC drops for TSC are small at Poison Rate=0. We will include these results in the revised manuscript.
>
> |Clean Dataset (SL)|No Defense|FP|NC|MCR|ANP|FT-SAM|I-BAU|SAU|TSC (ours)|
> |:---:|:---:|:---:|:---:|:---:|:---:|:---:|:---:|:---:|:---:|
> |CIFAR10|93.12|92.10|93.12|90.98|83.09|91.32|87.47|89.97|91.12|
> |GTSRB|99.20|99.11|99.20|98.14|95.43|98.04|95.40|96.84|98.31|
> |ImageNet100|84.32|82.22|84.32|81.46|77.78|82.90|77.10|76.93|81.44|
>
> |Pre-training (SSL-SimCLR)|Downstream|No Defense|MCR|SSL-Cleanse|TSC (ours)|
> |:---:|:---:|:---:|:---:|:---:|:---:|
> |CIFAR10 (Clean)|STL10|79.50|77.63|73.01|74.60|
> ||GTSRB|83.68|78.02|78.30|80.39|
> ||SVHN|66.57|60.19|63.55|63.41|
> |ImageNet100 (Clean)|STL10|95.62|90.92|89.27|89.81|
> ||GTSRB|77.58|74.41|69.90|71.74|
> ||SVHN|74.98|72.98|70.14|68.10|
>
>
> **Q2: Use of Same $t$ for Both Stages in Algorithm 1**
>
> In our original design, the parameter $t$ is used in both stages, with its values ranging from [0, 1]. We initially chose to use the same $t$ for both stages to maintain a simpler parameter design. Introducing different values of $t$ for each stage would increase the complexity of the design and lead to numerous potential parameter combinations, which could complicate the overall algorithm.
>
> Moreover, as shown in Figs.3 and 6–10, the ACC/ASR values in Stage 1 exhibit a roughly symmetric pattern with respect to $t$, whereas in Stage 2, they decrease as $t$ increases. Although the trends differ across the full range $t \in [0,1]$, they remain consistent within $t \in [0,0.5]$. Notably, in Stage 2, ASR decreases while ACC remains high around $t = 0.5$. Considering this and the symmetry in Stage 1, we select $t$ within $[0,0.5]$ for both stages. Within this range, both stages show a decreasing trend in ACC/ASR, making it reasonable to use similar $t$ values to balance high ACC and low ASR.
>
>
> We hope this explanation clarifies our design decision. We will include this discussion in the final version.

---

### Official Review · Reviewer_nnZL · 2025-03-12

**Overall Recommendation:** 3

**Summary:**

This paper proposes a new method for removing backdoor attacks from trained models post-training. Specifically, it maps the network to a different basin resulting in an functionally equivalent model and then shows that the bezier curve that cnnects the original and new model greatly reduces the adversarial samples loss. However, performing a single bezier curve also hurts the performance on clean samples, and another bezier curve from the original model to a point on the first curve is needed to fix that. This process is repeated a few times to get the clean model.

**Claims And Evidence:**

I believe the claims presented in the paper are convincingly supported by evidence.

**Essential References Not Discussed:**

-

**Experimental Designs Or Analyses:**

The experimental design seems valid to me.

**Methods And Evaluation Criteria:**

The evaluation criteria make sense to me for this problem.

**Other Comments Or Suggestions:**

- Figure 3 is very difficult to understand and could benefit from making the colors more clear / presenting less scenarios.

**Other Strengths And Weaknesses:**

Strengths:

  - The paper is well written and easy to follow.
  - The analysis (Figure 2) of the two loss landscapes over the curve between connecting the points from different basins is interesting. Moreover, the idea of transferring to a point on the connecting curve from another basin to remove the overfitting for adversarial examples is clever.
  - The approach is applicable to both supervised and self-supervised settings.

Weaknesses:
  - The motivation for this problem is not very clear to me. On the one hand, the training procedure is known, but on the other hand I dont haver control over it. It seems like a niche scenario. Can the authors give a few examples for when is this setting applicable?
  - In the supervised scenario the SAU baselines reaches comparable performance but it is not evaluated for self-supervised scenarios. However, since the self-supervised scenarios are evaluated by training a linear probing classifier on the backbone, it seems that SAU can be implemented over that linear probing classifier (i.e., in this scenario the purification happends after the fine-tuning of the model). As it shows strong performance in the supervised setting it would be helpful to add it to Tab. 2 and 3 as well.
  - Different t values are chosen for different experiments, however, in practice this hyper-parameter tuning is not possible as the adversarial samples are not given.

**Questions For Authors:**

- Can the authors provide more details on the experiment performed in Figure 2? E.g., what dataset was used? which backdoor attacks?

**Relation To Broader Scientific Literature:**

This paper aims to purify corrupted models which is important as neural network models are used for decision making, therefore we do not want to allow for backdoor attacks on them.

**Theoretical Claims:**

The theoretical analysis seems valid to me.

---

> ### Author Rebuttal · Authors · 2025-03-29
>
> We sincerely appreciate your insightful comments! We address your concerns as below.
>
> **W1: Applicability of the Proposed Setting (Threat Model)**
>
> As clarified in Section 3.4, we specifically consider two practical scenarios where defenders either face partial data poisoning or do not control the initial training procedure:
>
> (1) Data Poisoning Scenario: When an adversary poisons only a portion of the training data, defenders typically retain control over training. Consequently, they possess complete knowledge of the training process. This allows them to effectively apply post-training purification defenses like our proposed method.
>
> (2) Adversarial Training Control Scenario: We agree this scenario warrants additional clarification. Here are two real-world examples:
>  - **Public Pre-trained Models**: Public repositories or research papers release pre-trained models that may contain backdoors. Since these sources typically provide detailed descriptions of the model-training procedure, defenders can leverage this information to apply TSC effectively. Using public large-scale image encoders for downstream tasks is increasingly common, making our setting practically relevant. Advanced zero-shot deployment models (e.g., CLIP) further exemplify this applicability.
>  - **Internal Adversary in Organizations**: Consider an internal adversary scenario within an organization where malicious attackers backdoor a model without others' awareness. Typically, benign team members possess knowledge of the basic training process but lack insight into the malicious manipulations. In this context, defenders within the organization can deploy TSC to purify the model without taking retraining from scratch.
>
> **W2: Evaluation of SAU in Self-Supervised Scenarios**
>
> Indeed, supervised learning (SL) defenses such as SAU can be applied to the combination of encoder and linear classifier after fine-tuning. **However, as shown in Fig.1, our TSC approach specifically targets self-supervised learning (SSL) scenarios by directly purifying the encoder rather than the combined model. The other SSL defenses we evaluated, such as MCR and SSL-Cleanse, also follow this workflow**. This design enables TSC to be effectively applied to zero-shot scenarios, such as CLIP, where neither a linear classifier nor fine-tuning is required.
>
> Initially, we excluded SAU from SSL comparisons to maintain fairness and methodological consistency. Nevertheless, following your suggestion, we conducted additional experiments applying I-BAU and SAU to the combined model (SimCLR) using 5% downstream labeled data. The experiments were conducted under identical conditions as those in Appendix J.2, using the same backdoored models in Tab.2.
>
> The results below indicate that while I-BAU and SAU reduce the ASR, they significantly degrade benign accuracy (ACC). For instance, on CIFAR10-STL10, the ACC dropped from 76.73% to 30.13% with I-BAU and further to 21.52% with SAU. We suspect this decline occurs because I-BAU and SAU employ post-training methods analogous to adversarial training in SL, potentially harming the representation extraction capability of encoders trained via SSL methods like SimCLR.
>
> We will include these findings and discussions in the revised manuscript.
> |Pre-training|Downstream|No Defense||I-BAU||SAU||
> |:---:|:---:|:---:|:---:|:---:|:---:|:---:|:---:|
> |||ACC|ASR|ACC|ASR|ACC|ASR|
> |CIFAR10|STL10|**76.74**|99.65|**30.13**|12.42|**21.52**|7.22|
> ||GTSRB|**81.12**|98.79|**22.36**|15.10|**47.45**|5.52|
> ||SVHN|**63.12**|98.71|**43.44**|17.11|**24.07**|7.25|
> |ImageNet100|STL10|**94.93**|98.99|**74.62**|10.32|**81.24**|11.83|
> ||GTSRB|**75.94**|99.76|**39.85**|7.34|**10.75**|0.80|
> ||SVHN|**72.64**|99.21|**25.27**|13.70|**24.37**|4.10|
>
> **W3: Selection of t**
>
> Actually, the curve index t used in our method was **consistently fixed** across experiments.
>  - For experiments in SL (Tabs. 1, 4, 5, and 7-11), we set t=0.4.
>  - For experiments in SSL (Tabs. 2, 3, 6, and 12-14), we set t=0.2.
>
> The rationale behind these selections is detailed in Appendix G.1. We guess your confusion may have arisen from Fig.3, where our intention was to show the trend in accuracy across various t for analytical purposes. In practice, no experiment-specific tuning of t was performed.
>
> **C1: Clarity of Fig.3**
>
> We agree that Fig.3 currently appears complex due to the multiple scenarios presented. In the revised manuscript, we plan to simplify Fig.3 by focusing on the results against SSBA, aligning with Fig.2 for consistency. Additionally, we will split the visualizations of ASR and ACC trends into two separate subfigures to enhance readability.
>
> **Q1:Details of Experiment in Fig.2**
>
> Currently, detailed information regarding Fig.2 is provided in the caption text.
> The experiment involves a PreAct-ResNet18 model trained on the CIFAR-10 dataset, with the SSBA backdoor attack (5% posion rate). Since the information is buried in the text, we will explicitly include these details within Fig.2 itself.

---

> > ### Comment · Reviewer_nnZL · 2025-04-04
> >
> > Thank you for your response. While the motivation for this setting still seems a bit niche, I believe the additional explanations help. Additionally, the added results clarify some of my other concerns. For that, I choose to keep my score.

---

> > > ### Author Response · Authors · 2025-04-08
> > >
> > > Dear Reviewer,
> > >
> > > Thank you for your feedback and for taking the time to consider the additional clarifications we provided!
> > >
> > > We understand your concerns regarding the niche nature of the motivation for our approach. However, we believe that the scenarios we've outlined—**particularly the increasing use of public pre-trained models**—are becoming more prevalent, especially as modern deep learning algorithms require large-scale datasets and substantial computational resource.
> > >
> > > Moreover, beyond traditional learning scenarios, our method has potential extensions to other settings. For example, in federated learning, models are trained collaboratively across many distributed devices, with participants computing and sending local gradients for global aggregation. Malicious participants could inject poisoned updates into the system, introducing backdoors, even without direct access to the full training procedure. As the basic training method is shared across clients and the server, defenders in such setting can apply our method to remove backdoors with only a small amount of data.
> > >
> > > **To better illustrate the applicability of our method, we will include a detailed discussion in a separate section in the revised version**.

---

### Decision · Program_Chairs · 2025-05-01

**Decision:**

Accept (poster)

**Comment:**

The paper received mixed scores pre-rebuttal. The rebuttal addressed the reviewers' concerns well, making both Reviewers uSaT and AW4n increase their scores. The final scores include 2 Weak Accept and 2 Accept.

The ACs checked and agreed to accept the paper. The authors should include the discussions and results in the rebuttal in the camera-ready version.